# CONVOLUTIONS THROUGH THE LENS OF TENSOR NETWORKS

## ABSTRACT

Despite their simple intuition, convolutions are more tedious to analyze than dense layers, which complicates the transfer of theoretical and algorithmic ideas. We provide a simplifying perspective onto convolutions through tensor networks (TNs) which allow reasoning about the underlying tensor multiplications by drawing diagrams, and manipulating them to perform function transformations and sub-tensor access. We demonstrate this expressive power by deriving the diagrams of various autodiff operations and popular approximations of second-order information with full hyper-parameter support, batching, channel groups, and generalization to arbitrary convolution dimensions. Further, we provide convolution-specific transformations based on the connectivity pattern which allow to re-wire and simplify diagrams before evaluation. Finally, we probe computational performance, relying on established machinery for efficient TN contraction. Our TN implementation speeds up a recently-proposed KFAC variant up to 4.5 x and enables new hardware-efficient tensor dropout for approximate backpropagation.

## 1 INTRODUCTION

Convolutional neural networks (CNNs) (LeCun et al., 1989) mark a milestone in the development of deep learning architectures as their 'sliding window' approach represents an important inductive bias for vision tasks. Their intuition is simple to explain with graphical illustrations such as in Dumoulin & Visin (2016). Yet, convolutions are more challenging to analyze than fully-connected layers in multi-layer perceptrons (MLPs) or transformers (Vaswani et al., 2017). One reason is that they are hard to express in matrix notation and—even when switching to index notation—compact expressions that are convenient to work with only exist for special hyper-parameter choices (e.g. Grosse & Martens, 2016; Arora et al., 2019). Many hyper-parameters (stride, padding, . . . ) and additional features like channel groups (Krizhevsky et al., 2012) introduce additional complexity. And related objects like (higher-order) derivatives and related routines for autodiff inherit this complexity. Between CNNs and MLPs, we observe a delay of analytical and algorithmic developments, e.g.

|  | for MLPs | for CNNs |
|---|---|---|
| Approximate Hessian diagonal | 1989 | 2023 |
| Kronecker-factored curvature (KFAC, KFRA, KFLR) | 2015, 2017, 2017 | 2016, 2020b, 2020b |
| Kronecker-factored quasi-Newton methods (KBFGS) | 2021 | 2022 |
| Neural tangent kernel (NTK) | 2018 | 2019 |
| Hessian rank | 2021 | 2023 |

Here, we seek to reduce this complexity gap by providing a new perspective onto convolutions through tensor networks (TNs, e.g. Penrose, 1971; Biamonte & Bergholm, 2017; Bridgeman & Chubb, 2017), which express the underlying tensor multiplications as diagrams. Those simplify reading off structure like factorization, and can seamlessly be (i) manipulated to take derivatives or add batching, (ii) merged with other diagrams, and (iii) sliced to extract sub-tensors.

TNs are not only convenient for analytic investigations. Techniques to efficiently evaluate TNs have been developed by the quantum simulation community (e.g. Smith & Gray, 2018; Gray & Kourtis, 2021; Zhang, 2020; cuQuantum development team, 2023) and are accessible through a simple interface (`einsum`) with automated under-the-hood-optimizations like finding a high-quality contraction order, or distributing computations. In summary:

```python
from einconv import conv_index_pattern; from einops import einsum; from torch import nn, rand

I, K, D, P, S = ...   # convolution hyper-parameters (2-tuples)
X = rand(C_in, I[0], I[1])  # input to convolution
kfc_shape = (C_in * K[0] * K[1], C_in * K[0] * K[1])

def kfc_im2col():
  """Compute Kronecker factor via im2col."""
  X_unf = nn.functional.unfold(X, K, D, P, S)
  return einsum(X_unf, X_unf, "i out, j out -> i j")

def kfc_tn():
  """Compute Kronecker factor via its tensor network."""
  Pi1 = conv_index_pattern(I[0], K[0], S[0], P[0], D[0])
  Pi2 = conv_index_pattern(I[1], K[1], S[1], P[1], D[1])
  return einsum(X, Pi1, Pi2, X, Pi1, Pi2, "c_in i1 i2, "
    + "i1 o1 k1, i2 o2 k2, c_in_ i1_ i2_, i1_ o1_ k1_, "
    + "i2_ o2_ k2_ -> c_in k1 k2 c_in_ k1_ k2_").reshape(kfc_shape)

def kfc_simplified_tn(): # for dense convolutions
  """Compute Kronecker factor via its simplified tensor network."""
  X_re = X.reshape(C_in, I[0] // K[0], K[0], I[1] // K[1], K[1])
  return einsum(X_re, X_re, "c_in o1 k1 o2 k2, c_in_ o1_ k1_ o2_ k2_ ",
    + "-> c_in k1 k2 c_in_ k1_ k2_").reshape(kfc_shape)
```

Figure 1: Convolutions and related operations can be expressed as TNs and evaluated with einsum. We illustrate this for the input-based factor of the Kronecker-factorized Fisher approximation (KFC Grosse & Martens, 2016), whose standard implementation (*top*) requires unfolding the input (large memory). By replacing the unfolded input with its TN (*middle*), a contraction path optimizer like opt_einsum (Smith & Gray, 2018) can automatically optimize run time and/or memory inside einsum. (*Bottom*) For many convolutions, the TN further simplifies due to structures in the index pattern, which reduces cost. In practise, the TN versions need not be implemented; our framework automatically generates their einsum expressions.

- We use the TN format of convolution from Hayashi et al. (2019) to derive diagrams for various autodiff routines and structural approximations of second-order information with support for all hyper-parameters, batching, channel groups, and arbitrary dimensions (Table 1).

- We present transformations based on the convolution's connectivity pattern to re-wire and symbolically simplify TNs before evaluation (see Figure 1 for a full example).

- We compare the performance of default and TN implementation, demonstrating speed-ups up to 4.5 x for a recent KFAC variant, and use its increased flexibility to impose hardware-efficient dropout that reduces the cost of randomized backpropagation.

## 2 PRELIMINARIES

We briefly review 2d convolution (§2.1), describe tensor multiplication and the einsum interface (§2.2), introduce the graphical TN notation, and apply it to convolution (§2.3). Bold lower-case ($\boldsymbol{a}$), upper-case ($\boldsymbol{A}$), and upper-case sans-serif ($\mathsf{A}$) symbols indicate vectors, matrices, and tensors. Entries follow the same convention but use regular font weight and $[\cdot]$ denotes slicing ($[\boldsymbol{A}]_{i,j} = A_{i,j}$). Parenthesized indices mean reshapes, e.g. $[\boldsymbol{a}]_{(i,j)} = [\boldsymbol{A}]_{i,j}$ where $\boldsymbol{a}$ is the flattened matrix $\boldsymbol{A}$.

### 2.1 CONVOLUTION

2d convolutions process channels of two-dimensional signals $\mathsf{X} \in \mathbb{R}^{C_{\text{in}} \times I_1 \times I_2}$ with $C_{\text{in}}$ channels of spatial dimensions[1] $I_1, I_2$ by sliding a collection of $C_{\text{out}}$ filter banks, arranged in a kernel $\mathsf{W} \in \mathbb{R}^{C_{\text{out}} \times C_{\text{in}} \times K_1 \times K_2}$ with kernel size $K_1, K_2$, over the input. The sliding operation depends on various hyper-parameters (padding, stride, dilation, see Dumoulin & Visin, 2016). At each step, the filters are contracted with the overlapping area, yielding the channel values of a pixel in the output $\mathsf{Y} \in \mathbb{R}^{C_{\text{out}} \times O_1 \times O_2}$ with spatial dimensions $O_1, O_2$. Optionally, a bias from $\boldsymbol{b} \in \mathbb{R}^{C_{\text{out}}}$ is added per channel.

One way to implement convolution is via matrix multiplication (Chellapilla et al., 2006), similar to fully-connected layers. First, one extracts the overlapping patches from the input for each output, then flattens and column-stacks them into a matrix $[\![\mathsf{X}]\!] \in \mathbb{R}^{C_{\text{in}} K_1 K_2 \times O_1 O_2}$, called the *unfolded input* (also

---

[1] We prefer $I_1, I_2$ over the more common choice $H, W$ to simplify generalization to higher dimensions.

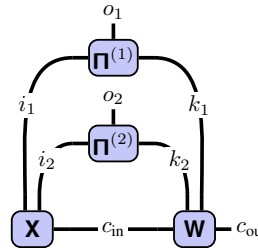

$$= \quad \text{(diagram) } - c_{\text{out}} \qquad Y_{c_{\text{out}},o_1,o_2} = \sum_{i_1,i_2,c_{\text{in}},k_1,k_2} X_{c_{\text{in}},i_1,i_2} \Pi^{(1)}_{i_1,o_1,k_1} \Pi^{(2)}_{i_2,o_2,k_2} W_{c_{\text{out}},c_{\text{in}},k_1,k_2}$$

where

Figure 2: TN representation of a 2d convolution with input **X** and kernel **W** (no batch dimension and no channel groups). The connectivity pattern along each dimension is made explicit via a tensor **Π**. Evaluating the indicated tensor multiplications (closed connections mean summation) yields the convolution's result **Y**.

called im2col). Multiplying a matrix view $W \in \mathbb{R}^{C_{\text{out}} \times C_{\text{in}} K_1 K_2}$ of the kernel onto the unfolded input then yields a matrix view $Y$ of **Y** (the vector of ones, $\mathbf{1}_{O_1 O_2}$, copies the bias for each channel),

$$Y = W[\![\mathbf{X}]\!] + b\, \mathbf{1}_{O_1 O_2}^{\top} \in \mathbb{R}^{C_{\text{out}} \times O_1 O_2} . \tag{1}$$

Alternatively, convolution can be seen as an affine map of the flattened input $x \in \mathbb{R}^{C_{\text{in}} I_1 I_2}$ into a vector view $y$ of **Y** with a Toeplitz-structured matrix $A(\mathbf{W}) \in \mathbb{R}^{C_{\text{out}} O_1 O_2 \times C_{\text{in}} I_1 I_2}$,

$$y = A(\mathbf{W})x + b \otimes \mathbf{1}_{O_1 O_2} \in \mathbb{R}^{C_{\text{out}} O_1 O_2} . \tag{2}$$

This perspective of unfolding the kernel is uncommon in implementations, but used in theoretical works (e.g. Singh et al., 2023) as it highlights the similarity between convolutions and dense layers.

## 2.2 Tensor Multiplication

Tensor multiplication unifies inner, element-wise (Hadamard), and outer (Kronecker) multiplication and relies on the input-output index relation to infer the multiplication type. We start with the binary case, then generalize to more inputs: Consider **A**, **B**, **C** whose index names are described by the index tuples $S_1, S_2, S_3$ where $S_3 \subseteq (S_1 \cup S_2)$ (converting tuples to sets if needed). Any multiplication of **A** and **B** can be described by the tensor multiplication operator $*_{(S_1,S_2,S_3)}$ with

$$\mathbf{C} = *_{(S_1,S_2,S_3)}(\mathbf{A},\mathbf{B}) \Leftrightarrow [\mathbf{C}]_{S_3} = \sum_{(S_1 \cup S_2) \setminus S_3} [\mathbf{A}]_{S_1}[\mathbf{B}]_{S_2} , \tag{3}$$

summing over indices that are not present in the output. E.g., for two matrices $A, B$, their product is $AB = *_{((i,j),(j,k),(i,k))}(A,B)$ (see §H.2), their Hadamard product $A \odot B = *_{((i,j),(i,j),(i,j))}(A,B)$, and their Kronecker product $A \otimes B = *_{((i,j),(k,l),((i,k),(j,l)))}(A,B)$. Libraries support this functionality via einsum, which takes a string encoding of $S_1, S_2, S_3$, followed by **A**, **B**. It also accepts longer sequences $\mathbf{A}_1, \ldots, \mathbf{A}_N$ with index tuples $S_1, S_2, \ldots, S_N$ and output index tuple $S_{N+1}$,

$$\mathbf{A}_{N+1} = *_{(S_1,\ldots,S_N,S_{N+1})}(\mathbf{A}_1,\ldots,\mathbf{A}_N) \Leftrightarrow [\mathbf{A}_{N+1}]_{S_{N+1}} = \sum_{(\bigcup_{n=1}^N S_n) \setminus S_{N+1}} \left(\prod_{n=1}^N [\mathbf{A}_n]_{S_n}\right). \tag{4}$$

Binary and $N$-ary tensor multiplication are commutative: Simultaneously permuting operands and their index tuples does not change the result, $*_{(S_1,S_2,S_3)}(\mathbf{A},\mathbf{B}) = *_{(S_2,S_1,S_3)}(\mathbf{B},\mathbf{A})$ and $*_{(\ldots,S_i,\ldots,S_j,\ldots)}(\ldots,\mathbf{A}_i,\ldots,\mathbf{A}_j\ldots) = *_{(\ldots,S_j,\ldots,S_i,\ldots)}(\ldots,\mathbf{A}_j,\ldots,\mathbf{A}_i,\ldots)$. They are also associative, i.e. we can multiply operands in any order. However, the notation becomes involved as it requires additional set arithmetic to detect when an index can be summed (see §H.1 for an example).

## 2.3 Tensor Networks & Convolution

A simpler way to understand tensor multiplications is via diagrams developed by e.g. Penrose (1971). Rank-$K$ tensors are represented by nodes with $K$ legs labelled by the index's name[2]. For instance, $\boxed{a}-i$ denotes a vector $a$, $i-\boxed{B}-j$ a matrix $B$, and $i-\boxed{c}-k$ (with $j$) a rank-3 tensor **C**. The 2d Kronecker delta $[\boldsymbol{\delta}]_{i,j} = \delta_{i,j}$ is simply a line, $j-\boxed{\delta}-i = j-\boxed{I}-i = j{-\!\!-\!\!-}i$. Multiplications are indicated by connections between legs. For inner multiplication, we join the legs of the involved indices, e.g. the matrix multiplication diagram is $i-\boxed{AB}-k = i-\boxed{A}-j-\boxed{B}-k$. Element-wise multiplication is similar, but with a leg sticking out. For example, the Hadamard and Kronecker product diagrams are

$$i-\boxed{A \odot B}-j = i-\left[\begin{array}{c}\boxed{A}\\\boxed{B}\end{array}\right]-j, \qquad (i,k)-\boxed{A \otimes B}-(j,l) = (i,k)\blacktriangleleft\begin{array}{c}i-\boxed{A}-j\\k-\boxed{B}-l\end{array}\blacktriangleright(j,l). \tag{5}$$

---

[2]We use identical shapes for all tensors. Leg orientation does not assign properties like co-/contra-variance.

Table 1: Contraction expressions of operations related to 2d convolution. They include batching and channel groups, which are standard features in implementations. We describe each operation by a tuple of input tensors and a contraction string that uses the `einops` library's syntax (Rogozhnikov, 2022) which can express index (un-)grouping. Some quantities are only correct up to a scalar factor which is suppressed for brevity. See §B for visualizations and Table B2 for more operations.

| Operation | Operands | Contraction string (`einops` (Rogozhnikov, 2022) convention) |
|---|---|---|
| Convolution (no bias) | $\mathbf{X}, \mathbf{\Pi}^{(1)}, \mathbf{\Pi}^{(2)}, \mathbf{W}$ | `"n (g c_in) i1 i2, i1 o1 k1, i2 o2 k2, (g c_out) c_in k1 k2 -> n (g c_out) o1 o2"` |
| Unfolded input (im2col) | $\mathbf{X}, \mathbf{\Pi}^{(1)}, \mathbf{\Pi}^{(2)}$ | `"n c_in i1 i2, i1 o1 k1, i2 o2 k2 -> n (c_in k1 k2) (o1 o2)"` |
| Unfolded kernel (Toeplitz) | $\mathbf{\Pi}^{(1)}, \mathbf{\Pi}^{(2)}, \mathbf{W}$ | `"i1 o1 k1, i2 o2 k2, c_out c_in k1 k2 -> (c_out o1 o2) (c_in i1 i2)"` |
| Weight VJP | $\mathbf{X}, \mathbf{\Pi}^{(1)}, \mathbf{\Pi}^{(2)}, \mathbf{V}^{(\mathbf{Y})}$ | `"n (g c_in) i1 i2, i1 o1 k1, i2 o2 k2, n (g c_out) o1 o2 -> (g c_in) c_in k1 k2"` |
| Input VJP (transpose conv.) | $\mathbf{W}, \mathbf{\Pi}^{(1)}, \mathbf{\Pi}^{(2)}, \mathbf{V}^{(\mathbf{Y})}$ | `"(g c_out) c_in k1 k2, i1 o1 k1, i2 o2 k2, n (g c_out) o1 o2 -> n (g c_in)"` |
| KFC/KFAC-expand | $\mathbf{X}, \mathbf{\Pi}^{(1)}, \mathbf{\Pi}^{(2)}, \mathbf{X}, \mathbf{\Pi}^{(1)}, \mathbf{\Pi}^{(2)}$ | `"n (g c_in) i1 i2, i1 o1 k1, i2 o2 k2, n (g c_in) i1 i2, i1 o1 k1, i2 o2 k2 -> g (c_in k1 k2) (c_in k1 k2)"` |
| KFAC-reduce | $\mathbf{X}, \mathbf{\Pi}^{(1)}, \mathbf{\Pi}^{(2)}, \mathbf{X}, \mathbf{\Pi}^{(1)}, \mathbf{\Pi}^{(2)}$ | `"n (g c_in) i1 i2, i1 o1 k1, i2 o2 k2, n (g c_in) i1 i2, i1 o1_ k1_, i2 o2_ k2_ -> g (c k1 k2) (c_ k1_ k2_)"` |

Note that the outer tensor product yields a rank-4 tensor which needs to be reshaped (indicated by black triangles[3]) to obtain a matrix. This syntax allows for extracting and embedding tensors along diagonals; e.g. taking a matrix diagonal, $\boxed{\text{diag}(A)}\text{-}i = \llcorner\boxed{A}\lrcorner\text{-}i$, or forming a diagonal matrix, $i\text{-}\boxed{\text{diag}(a)}\text{-}i = i\llcorner\boxed{a}\lrcorner\text{-}i$; and generalizes to larger diagonal blocks (§B). In the following, we stick to the simplest case to avoid the more advanced syntax. However, it shows the expressive power of TNs and is required to support common features of convolutions like channel groups (known as separable convolutions).

**Application to Convolution:** We define a binary tensor $\mathbf{P} \in \{0, 1\}^{I_1 \times O_1 \times K_1 \times I_2 \times O_2 \times K_2}$ which represents the connectivity pattern between input, output, and kernel. $P_{i_1, o_1, k_1, i_2, o_2, k_2}$ is 1 if input locations $(i_1, i_2)$ overlap with kernel positions $(k_1, k_2)$ when computing output locations $(o_1, o_2)$ and 0 otherwise. The spatial couplings are independent along each dimension, hence $\mathbf{P}$ decomposes into $P_{i_1, o_1, k_1, i_2, o_2, k_2} = \Pi^{(1)}_{i_1, o_1, k_1} \Pi^{(2)}_{i_2, o_2, k_2}$ where the index pattern tensor $\mathbf{\Pi}^{(j)} \in \{0, 1\}^{I_j \times O_j \times K_j}$ encodes the connectivity along dimension $j$. With that, one obtains

$$Y_{c_{\text{out}}, o_1, o_2} = b_{c_{\text{out}}} + \sum_{c_{\text{in}}=1}^{C_{\text{in}}} \sum_{i_1=1}^{I_1} \sum_{i_2=1}^{I_2} \sum_{k_1=1}^{K_1} \sum_{k_2=1}^{K_2} X_{c_{\text{in}}, i_1, i_2} \Pi^{(1)}_{i_1, o_1, k_1} \Pi^{(2)}_{i_2, o_2, k_2} W_{c_{\text{out}}, c_{\text{in}}, k_1, k_2} \,,$$

which translates into the TN diagram shown in Figure 2 if neglecting the bias.

## 3 TENSOR NETWORKS FOR CONVOLUTION OPERATIONS

We now demonstrate the elegance of TNs for computing derivatives (§3.1), autodiff operations (§3.2), and approximate second-order information (§3.3) by graphical manipulation. For simplicity, we exclude batching (`vmap`-ing like in JAX (Bradbury et al., 2018)) and channel groups, and provide the diagrams with full support in §B. Table 1 summarizes our derivations (with batching and groups).

As a warm-up, we identify the unfolded input and kernel from the matrix-multiplication view from Equations (1) and (2). They follow by contracting the index patterns with either the input or kernel,

$$[[\mathbf{X}]]_{(c_{\text{in}}, k_1, k_2),(o_1, o_1)} = \sum_{i_1=1}^{I_1} \sum_{i_2=1}^{I_2} X_{c_{\text{in}}, i_1, i_2} \Pi^{(1)}_{i_1, o_1, k_1} \Pi^{(2)}_{i_2, o_2, k_2} \,,$$

$$[A(\mathbf{W})]_{(c_{\text{out}}, o_1, o_2),(c_{\text{in}}, i_1, i_2)} = \sum_{k_1=1}^{K_1} \sum_{k_2=1}^{K_2} \Pi^{(1)}_{i_1, o_1, k_1} \Pi^{(2)}_{i_2, o_2, k_2} W_{c_{\text{out}}, c_{\text{in}}, k_1, k_2} \,,$$

or, in diagram notation,

(6)

### 3.1 TENSOR NETWORK DIFFERENTIATION

Derivatives play a crucial role in theoretical and practical ML. First, we show that differentiating a TN diagram amounts to a simple graphical manipulation. Then, we derive the Jacobians of convolution.

---

[3]Reshape can be seen as tensor multiplication with a one-hot tensor, but we decided to use a separate symbol to emphasize that it merely serves for re-interpreting the tensor and does not cause expensive computations.

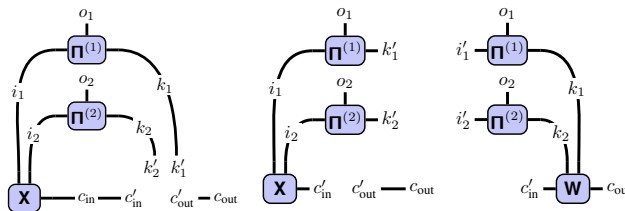

(a) Argument cutting     (b) Weight Jacobian     (c) Input Jacobian

Figure 3: TN differentiation as graphical manipulation. (a) Differentiating a 2d convolution w.r.t. $\mathbf{W}$ requires cutting it out of the diagram, introducing indices for open legs. (b) Weight Jacobian after simplifying the new legs connected to other tensors. (c) Same procedure applied to the Jacobian w.r.t. $\mathbf{X}$.

Consider an arbitrary TN represented by the tensor multiplication from Equation (4). The Jacobian tensor $[\mathbf{J}_{\mathbf{A}_j}\mathbf{A}_{N+1}]_{S_{N+1},S'_j} = \partial[\mathbf{A}_{N+1}]_{S_{N+1}}/\partial[\mathbf{A}_j]_{S'_j}$ w.r.t. an input $\mathbf{A}_j$ collects all partial derivatives and is addressed through indices $S_{n+1} \times S'_j$ with $S'_j$ an independent copy of $S_j$. Assume that $\mathbf{A}_j$ only enters once in the tensor multiplication. Then, taking the derivative of Equation (4) w.r.t. $[\mathbf{A}_j]_{S'_j}$ simply replaces the tensor by a Kronecker delta $\delta_{S_j,S'_j}$,

$$\frac{\partial[\mathbf{A}_{N+1}]_{S_{N+1}}}{\partial[\mathbf{A}_j]_{S'_j}} = \sum_{\left(\bigcup_{n=1}^{N} S_n\right)\setminus S_{n+1}} [\mathbf{A}_1]_{S_1}\cdots[\mathbf{A}_{j-1}]_{S_{j-1}}\left(\prod_{i\in S_j}\delta_{i,i'}\right)[\mathbf{A}_{j+1}]_{S_{j+1}}\cdots[\mathbf{A}_N]_{S_N}. \quad (7)$$

If an index $i \in S_j$ is summed, $i \notin S_{n+1}$, we can sum the Kronecker delta $\delta_{i,i'}$, effectively replacing all occurrences of $i$ by $i'$. If instead $i$ is part of the output index, $i \in S_{n+1}$, the Kronecker delta remains part of the Jacobian and imposes structure. Figures 3a and 3b illustrate this process in diagrams for differentiating a convolution w.r.t. its kernel. Equation (7) amounts to cutting out the argument of differentiation and assigning new indices to the resulting open legs (Figure 3a). Then, we can simplify the new legs connected to other tensors (Figure 3b). For the weight Jacobian $\mathbf{J}_{\mathbf{W}}\mathbf{Y}$, this introduces structure: If we re-interpret the two disjoint diagrams in Figure 3b as matrices, compare with the Kronecker diagram from Equation (5) and use Equation (6), we find $[\![\mathbf{X}]\!]^\top \otimes \boldsymbol{I}_{C_{\text{out}}}$ for the Jacobian's matrix view (e.g. Dangel et al., 2020a). Figure 3c shows the input Jacobian $\mathbf{J}_{\mathbf{X}}\mathbf{Y}$ which is given by a tensor view of $\boldsymbol{A}(\mathbf{W})$, as expected from the matrix-vector perspective of Equation (2).

Differentiating a TN is more convenient than using matrix calculus (Magnus & Neudecker, 1999) as it amounts to a simple graphical manipulation and does not rely on a flattening convention and therefore preserves the full index structure. The resulting TN can still be translated back to matrix language, if desired. It also simplifies the computation of higher-order derivatives (e.g. $\partial^2\mathbf{Y}/\partial\mathbf{w}\partial\mathbf{x}$), since differentiation yields another TN and can thus be repeated. If a tensor occurs more than once in a TN, the product rule applies and the derivative is a sum of TNs with one occurrence removed.

## 3.2 AUTOMATIC DIFFERENTIATION & CONNECTIONS TO TRANSPOSE CONVOLUTION

Although Jacobians can sometimes be useful, crucial routines for integration with autodiff are vector-Jacobian and Jacobian-vector products (VJPs, JVPs). Both are simple to realize with TNs due to access to full Jacobians. VJPs are used in backpropagation to pull back a tensor $\mathbf{V}^{(\mathbf{Y})} \in \mathbb{R}^{C_{\text{out}}\times O_1\times O_2}$ from the output space. The VJP results $\mathbf{V}^{(\mathbf{X})} \in \mathbb{R}^{C_{\text{in}}\times I_1\times I_2}$ and $\mathbf{V}^{(\mathbf{W})} \in \mathbb{R}^{C_{\text{out}}\times C_{\text{in}},K_1,K_2}$ are

$$V^{(\mathbf{X})}_{c'_{\text{in}},i'_1,i'_2} = \sum_{c_{\text{out}},o_1,o_2} V^{(\mathbf{Y})}_{c_{\text{out}},o_1,o_2}\frac{\partial Y_{c_{\text{out}},o_1,o_2}}{\partial X_{c'_{\text{in}},i'_1,i'_2}}, \quad V^{(\mathbf{W})}_{c'_{\text{out}},c'_{\text{in}},k'_1,k'_2} = \sum_{c_{\text{out}},o_1,o_2} V^{(\mathbf{Y})}_{c_{\text{out}},o_1,o_2}\frac{\partial Y_{c_{\text{out}},o_1,o_2}}{\partial W_{c'_{\text{out}},c'_{\text{in}},k'_1,k'_2}}.$$

Both are simply new TNs constructed from contracting the vector with the respective Jacobian, see Figure 4 (VJPs are analogous). The input VJP is often used to define transpose convolution (Dumoulin & Visin, 2016). In the matrix-multiplication perspective (Equation (2)), this operation is defined relative to a convolution with kernel $\mathbf{W}$ by multiplication with $\boldsymbol{A}(\mathbf{W})^\top$, i.e. using the same connectivity pattern but mapping from the convolution's output space to its input space. The TN makes this pattern sharing explicit as the same $\boldsymbol{\Pi}$s are used, and provides a clean definition of transpose convolution.[4]

## 3.3 KRONECKER-FACTORED APPROXIMATE CURVATURE (KFAC)

The Jacobian TN diagrams allow to construct the TNs of second-order information like the Fisher/generalized Gauss-Newton (GGN) matrix and sub-tensors like its diagonal (see §C) Here, we focus

---

[4]In standalone implementations of transpose convolution, one must supply an additional parameter to unambiguously reconstruct the convolution's input dimension (see §D for how to compute $\boldsymbol{\Pi}$ in this case).

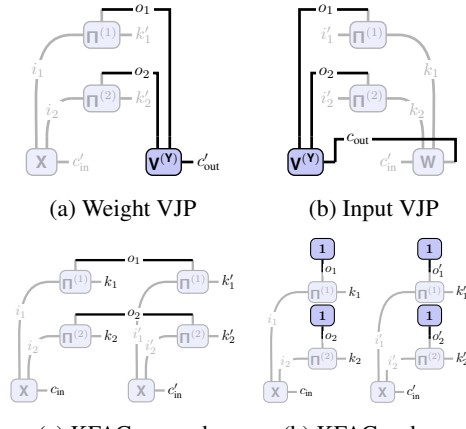

(a) Weight VJP      (b) Input VJP

(a) KFAC-expand      (b) KFAC-reduce

Figure 4: TNs of 2d convolution VJPs for backpropagation. Jacobians from Figure 3 are shaded, only their contraction with the vector $\mathbf{V}^{(\mathbf{Y})}$ is highlighted. (a) VJP for the weight and (b) input Jacobian (transpose convolution). JVPs are similar, but contract vectors $\mathbf{V}^{(\mathbf{X})} \in \mathbb{R}^{C_{\text{in}} \times I_1 \times I_2}$, $\mathbf{V}^{(\mathbf{W})} \in \mathbb{R}^{C_{\text{out}} \times C_{\text{in}} \times K_1 \times K_2}$ with the input and kernel indices of $\mathbf{J_X Y}$, $\mathbf{J_W Y}$.

Figure 5: TNs of input-based Kronecker factors for block-diagonal Fisher/GGN approximations (no batching, no channel groups). The unfolded input is shaded, only additional contractions are highlighted. (a) $\mathbf{\Omega}$ (KFC/KFAC-expand) from Grosse & Martens (2016) and (b) $\hat{\mathbf{\Omega}}$ (KFAC-reduce) from Eschenhagen (2022) (the vectors of ones effectively amount to sums).

on the popular Kronecker-factored approximation of the GGN (Martens & Grosse, 2015; Grosse & Martens, 2016; Eschenhagen, 2022; Martens et al., 2018) whose input-based Kronecker factor requires the unfolded input $[\![\mathbf{X}]\!]$ which requires large memory. State-of-the-art libraries that provide access to it (Dangel et al., 2020b; Osawa et al., 2023) rely on this approach via im2col. Using the TN of $[\![\mathbf{X}]\!]$, we can often avoid expanding it explicitly and save memory. Here, we describe the existing Kronecker approximations of the GGN and their TNs (see §5.1 for their run time evaluation).

**KFC (KFAC-expand):** Grosse & Martens (2016) introduce a Kronecker approximation for the kernel's GGN, $G \approx \mathbf{\Omega} \otimes \mathbf{\Gamma}$ where $\mathbf{\Gamma} \in \mathbb{R}^{C_{\text{out}} \times C_{\text{out}}}$ and the input-based factor is $\mathbf{\Omega} = [\![\mathbf{X}]\!]^{\top} [\![\mathbf{X}]\!] \in \mathbb{R}^{C_{\text{in}} K_1 K_2 \times C_{\text{in}} K_1 K_2}$ (Figure 5a), the unfolded input's self-inner product (averaged over a batch).

**KFAC-reduce:** Eschenhagen (2022) generalized KFAC to graph neural networks and transformers based on the concept of weight sharing, also present in convolutions. They identify two approximations—KFAC-expand and KFAC-reduce—the former of which corresponds to KFC (Grosse & Martens, 2016). The latter shows similar performance in downstream tasks, but is cheaper to compute. It relies on the column-averaged unfolded input, i.e. the average over all patches sharing the same weights. KFAC-reduce approximates $G \approx \hat{\mathbf{\Omega}} \otimes \hat{\mathbf{\Gamma}}$ with $\hat{\mathbf{\Gamma}} \in \mathbb{R}^{C_{\text{out}} \times C_{\text{out}}}$ and $\hat{\mathbf{\Omega}} = {}^{1}/(O_1 O_2)^2 (\mathbf{1}_{O_1 O_2}^{\top} [\![\mathbf{X}]\!])^{\top} \mathbf{1}_{O_1 O_2}^{\top} [\![\mathbf{X}]\!] \in \mathbb{R}^{C_{\text{in}} K_1 K_2 \times C_{\text{in}} K_1 K_2}$ (Figure 5b; averaged over a batch).

## 4   TENSOR NETWORK SIMPLIFICATIONS & IMPLEMENTATION ASPECTS

Many convolutions in real-world CNNs use structured connectivity patterns that allow for simplifications which we describe here along with implementation aspects for efficient TN contraction.

### 4.1   CONVOLUTION INDEX PATTERN STRUCTURE & SIMPLIFICATIONS

The index pattern $\mathbf{\Pi}$ encodes the connectivity of a convolution and depends on its hyper-parameters. Along one dimension, $\mathbf{\Pi} = \mathbf{\Pi}(I, K, S, P, D)$ with input size $I$, kernel size $K$, stride $S$, padding $P$, and dilation $D$. We provide pseudo-code for computing $\mathbf{\Pi}$ in §D which is easy to implement efficiently with standard functions from any numerical library (Algorithm D1). Its entries are

$$[\mathbf{\Pi}(I, K, S, P, D)]_{i,o,k} = \delta_{i, 1+(k-1)D+(o-1)S-P}, \quad i = 1, \ldots, I, \ o = 1, \ldots, O, \ k = 1, \ldots, K \quad (8)$$

with spatial output size $O(I, K, S, P, D) = 1 + \lfloor (I + 2P - (K + (K-1)(D-1)))/S \rfloor$. Since $\mathbf{\Pi}$ is binary and has size linear in $I, O, K$, it is cheap to pre-compute and cache.

The index pattern's symmetries allow for re-wiring a TN. For instance, the symmetry of $(k, D)$ and $(o, S)$ in Equation (8) and $O(I, K, S, P, D)$ permits a *kernel-output swap*, exchanging the role of kernel and output dimension (Figure 6a). Rochette et al. (2019) used this to phrase the per-example gradient computation (weight VJP, Figure 4a) as convolution.

For many convolutions of real-world CNNs (see §E for a hyper-parameter study) the index pattern possesses structure that simplifies its contraction with other tensors into either smaller con-

tractions or reshapes: *Dense convolutions* use a shared kernel size and stride, and thus process non-overlapping adjacent tiles of the input. Their index pattern's action can be expressed as a cheap reshape (Figure 6b). Such convolutions are common in DenseNets (Huang et al., 2017), MobileNets (Howard et al., 2017; Sandler et al., 2018), ResNets (He et al., 2016), and ConvNeXts (Liu et al., 2022). InceptionV3 (Szegedy et al., 2016) has 2d *mixed-dense convolutions* that are dense along one dimension. *Down-sampling convolutions* use a larger stride than kernel size, hence only process a sub-set of their input, and are used in ResNet18 (He et al., 2016), ResNext101 (Xie et al., 2017), and WideResNet101 (Zagoruyko & Komodakis, 2016). Their pattern contracts with a tensor $\mathbf{V}$ like that of a dense convolution with a sub-tensor $\check{\mathbf{V}}$ (Figure 6c). §5.1 shows that those simplifications accelerate computations.

(a) Kernel-output swap

(b) Dense convolution

(c) Down-sampling convolution

Figure 6: TN illustrations of index pattern simplifications and transformations. See §D.3 for their mathematical formulation.

## 4.2 PRACTICAL BENEFITS OF THE TN ABSTRACTION & LIMITATIONS FOR CONVOLUTIONS

**Contraction order optimization:** There exist various orders in which to carry out the summations in a TN and their performance can vary by orders of magnitude. One extreme approach is to carry out all summations via nested for-loops. This so-called Feynman path integral algorithm requires little memory, but many FLOPS since it does not re-cycle intermediate results. The other extreme is sequential pair-wise contraction. This builds up intermediate results and can greatly reduce FLOPS. The schedule is represented by a binary tree, but the underlying search is in general at least #P-hard (Damm et al., 2002). Fortunately, there exist heuristics to find high-quality contraction trees for TNs with hundreds of tensors (Huang et al., 2021; Gray & Kourtis, 2021; cuQuantum development team, 2023), implemented in packages like opt_einsum (Smith & Gray, 2018).

**Index slicing:** A common problem with high-quality schedules is that intermediates exceed memory. Dynamic slicing (Huang et al., 2021) (e.g. cotengra (Gray & Kourtis, 2021)) is a simple method to decompose a contraction until it becomes feasible by breaking it up into smaller identical sub-tasks whose aggregation adds a small overhead. This enables peak memory reduction and distribution.

**Sparsity:** $\mathbf{\Pi}$ is sparse as only a small fraction of the input contributes to an output element. For a convolution with stride $S < K$ and otherwise default parameters ($P = 0, D = 1$), for fixed output and kernel indices $k, o$, there is exactly one non-zero entry in $[\mathbf{\Pi}]_{:,o,k}$. Hence $\mathrm{nnz}(\mathbf{\Pi}) = OK$, which corresponds to a sparsity of $1/I$. Padding leads to more kernel elements that do not contribute to an output pixel, and therefore a sparser $\mathbf{\Pi}$. For down-sampling and dense convolutions, we showed how $\mathbf{\Pi}$'s algebraic structure allows to simplify its contraction. However, if that is not possible, $\mathbf{\Pi}$ contains explicit zeros that add unnecessary FLOPS. One way to circumvent this is to match a TN with that of an operation with efficient implementation (like im2col, (transpose) convolution) using transformations like the *kernel-output swap* or by introducing identity tensors to complete a template, as done in Rochette et al. (2019); Dangel (2021) for per-sample gradients and im2col.

**Approximate contraction & structured dropout:** TNs offer a principled approach for stochastic approximation via Monte-Carlo estimation to save memory and run time at the cost of accuracy. The basic idea is best explained on a matrix product $\mathbf{C} := \mathbf{AB} = \sum_{n=1}^{N} [\mathbf{A}]_{:,n} [\mathbf{B}]_{n,:}$ with $\mathbf{A} \in \mathbb{R}^{I \times N}, \mathbf{B} \in \mathbb{R}^{N,O}$. To approximate the sum, we introduce a distribution over $n$'s range, then use column-row-sampling (CRS, Adelman et al., 2021) to form an unbiased Monte-Carlo approximation with sampled indices, which only requires the sub-matrices with active column-row pairs . Bernoulli-CRS samples without replacement by assigning a Bernoulli random variable $\mathrm{Bernoulli}(\pi_n)$ with probability $\pi_n$ for column-row pair $n$ to be included in the contraction. The Bernoulli estimator is $\tilde{\mathbf{C}} := \sum_{n=1}^{N} z_n/\pi_n [\mathbf{A}]_{n,:} [\mathbf{B}]_{n,:}$ with $z_n \sim \mathrm{Bernoulli}(\pi_n)$. With a shared keep probability, $\pi_n := p$, this yields the unbiased estimator $\mathbf{C}' = 1/p \sum_{n=1,\ldots,N} \mathbf{A}'\mathbf{B}'$ where $\mathbf{A}' = \mathbf{AK}$ and $\mathbf{B}' = \mathbf{KB}$ with $\mathbf{K} = \mathrm{diag}(z_1, \ldots, z_N)$ are the sub-matrices of $\mathbf{A}, \mathbf{B}$ containing the active column-row pairs. CRS applies to a single contraction. For TNs with multiple sums, we can apply it individually. Also, we can impose a distribution over the result indices, which leads to computing a (scaled) sub-tensor.

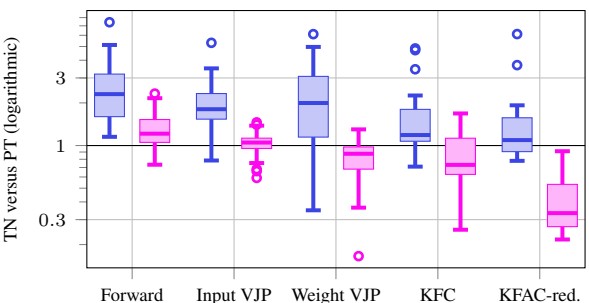

Figure 7: Run time ratios of TN (w/o simplifications) versus standard implementation for dense convolutions of 9 CNNs. With simplifications, convolution and input VJP achieve median ratios slightly above 1, and the TN implementation is faster for weight VJP, KFC & KFAC-reduce. The coloured boxes in Figure 1 correspond to default, TN, and simplified TN implementation for KFC.

## 5 EXPERIMENTS

### 5.1 RUN TIME EVALUATION

We implement the presented TNs' contraction strings and operands[5] in PyTorch (Paszke et al., 2019). The simplifications from §4 can be applied on top and yield a modified einsum expression. To find a contraction schedule, we use opt_einsum (Smith & Gray, 2018) with default settings. We extract the unique convolutions of 9 architectures for ImageNet and smaller data sets, then compare some operations from Table 1 with their standard implementation on an Nvidia Tesla T4 GPU (16 GB); see §F for all details. Due to space constraints, we highlight important insights here and provide references to the corresponding material in the appendix. In general, the performance gap between standard and TN implementation decreases the less common an operation is (Figure F16); from forward pass (inference & training), to VJPs (training), to KFAC (training with a second-order method). This is intuitive as more frequently used routines have been optimized more aggressively.

**Impact of TN simplifications:**   While general convolutions remain unaffected (Figure F17d) when applying the transformations of §4, mixed dense, dense, and down-sampling convolutions consistently enjoy significant run time improvements (Figures F17a to F17c). As an example, we show the performance comparison for dense convolutions in Figure 7: The performance ratio's median between TN and standard forward and input VJP is close to 1, that is both require almost the same time. In the median, the TN even outperforms PyTorch's highly optimized weight VJP, also for down-sampling convolutions (Figure F20). For KFC, the median performance ratios are well below 1 for dense, mixed dense, and sub-sampling convolutions (Figure F21).

**KFAC-reduce:**   For all convolution types, the TN implementation achieves its largest improvements for $\hat{\Omega}$ and consistently outperforms the PyTorch implementation in the median when simplifications are enabled (Figure F22). The standard implementation unfolds the input, takes the row-average, then forms its outer product. The TN does not need to expand $[\![\mathbf{X}]\!]$ in memory and instead averages the index pattern tensors, which reduces peak memory and run time. We observe performance ratios down to 0.22 x (speed-ups up to $\approx 4.5$ x, Table F8) and memory savings up to 3 GiB (§G.1). Hence, our approach can significantly reduce the overhead of a 2nd-order optimizer based on KFAC-reduce like that of Petersen et al. (2023) which only relies on the input-based factor (setting $\mathbf{\Gamma} \propto \mathbf{I}$).

### 5.2 RANDOMIZED AUTODIFF VIA APPROXIMATE TENSOR CONTRACTION

CRS is an alternative to gradient checkpointing (Griewank & Walther, 2008) to lower memory consumption of backpropagation (Oktay et al., 2021; Chen et al., 2023; Adelman et al., 2021). Here, we focus on unbiased gradient approximations by applying the exact forward pass, but CRS when computing the weight VJP, which requires storing a sub-tensor of $\mathbf{X}$. For convolutions, the approaches of existing works are limited by the supported functionality of ML libraries. Adelman et al. (2021) restrict to sampling $\mathbf{X}$ along $c_{\text{in}}$, which eliminates many gradient entries as the index is part of the gradient. The randomized gradient would thus only train a sub-tensor of the kernel per step. Oktay et al. (2021); Chen et al. (2023) apply unstructured dropout to $\mathbf{X}$, store it in sparse form, and restore the sparsified tensor during the backward pass. This reduces memory, but does not reduce computation.

---

[5]einsum does not yet support index un-grouping, so we must reshape manually before and after.

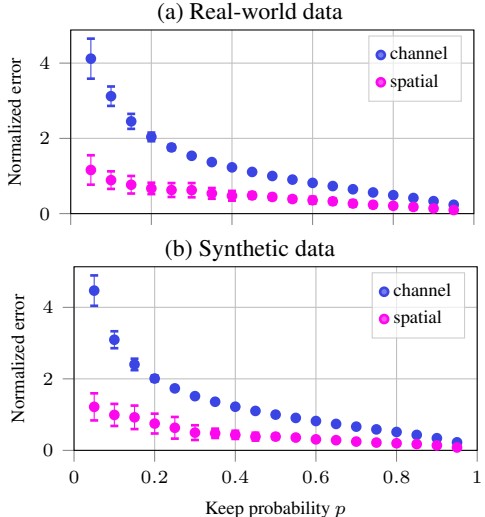

(a) Real-world data

(b) Synthetic data

Figure 8: Sampling spatial axes is more effective than sampling channels on both (a) real-world and (b) synthetic data. We take the untrained All-CNN-C (Springenberg et al., 2015) for CIFAR-100 with cross-entropy loss, disable dropout, and modify the convolutions to use use a fraction $p$ of **X** when computing the weight gradient via Bernoulli-CRS. For mini-batches of size 128, we compute the deterministic gradients for all kernels, then flatten and concatenate them into a vector $\boldsymbol{g}$; likewise for its proxy $\hat{\boldsymbol{g}}$. CRS is described by $(p_{c_{\text{in}}}, p_{i_1}, p_{i_2})$, the keep rates along the channel and spatial dimensions. We compare channel and spatial sampling with same memory reduction, i.e. $(p, 1, 1)$ and $(1, \sqrt{p}, \sqrt{p})$. To measure approximation quality, we use the normalized residual norm $\|\boldsymbol{g} - \hat{\boldsymbol{g}}\|_2 / \|\boldsymbol{g}\|_2$ and report mean and standard deviation of 10 different model and batch initializations.

Our TN implementation is more flexible and can, for example, tackle spatial dimensions with CRS. This reduces memory to the same extent, but also run time due to fewer contractions. Importantly, it does not zero out the gradient for entire filters. In Figure 8 we compare the gradient approximation errors of channel and spatial sub-sampling. For the same memory reduction, spatial sub-sampling yields a smaller approximation error on both real & randomly generated data. E.g., instead of keeping 75 % of channels, we achieve the same approximation quality using only 35 % of pixels.

## 6 RELATED WORK

**Structured convolutions:** We use the TN formulation of convolution from Hayashi et al. (2019) who focus on connecting kernel factorizations to existing (depth-wise separable (Howard et al., 2017; Sandler et al., 2018), factored (Szegedy et al., 2016), bottleneck (He et al., 2016), flattened/CP decomposed, low-rank filter (Smith, 1997; Rigamonti et al., 2013; Tai et al., 2015)) convolutions and explore new factorizations. Our work focuses on operations related to convolutions, diagram manipulations, the index pattern structure, and computational performance/flexibility. Structured convolutions integrate seamlessly with our framework by replacing the kernel with its factorized TN.

**Higher-order autodiff:** ML frameworks prioritize differentiating scalar-valued objectives once. A recent line of works (Laue et al., 2018; 2020; Ma et al., 2020) developed a tensor calculus framework for computing (higher-order) derivatives of matrix/tensor-valued functions, along with compilation techniques based on linear algebra and common sub-expression elimination (CSE). By phrasing convolution as `einsum`, we allow it to be integrated into such frameworks, make it amenable to their optimizations, and complement them with our convolution-specific simplifications.

## 7 CONCLUSION

We proposed using tensor networks (TNs), a diagrammatic representation of tensor multiplications, to analyze convolutions and provide white-box implementations of related routines for autodiff and curvature approximations via simple `einsum` expressions. We derived the diagrams of those operations with full hyper-parameter support, channel groups, batching, and generalization to arbitrary dimensions. This abstraction benefits from automated under-the-hood performance optimizations inside `einsum` (contraction path search, distributing computations). We complemented those by convolution-specific simplifications based on structure in the connectivity pattern and demonstrated their effectiveness to speed up the computation of approximate second-order information (up to 4.5 x).

Our work underlines the elegance and expressiveness of TNs for applying function transformations (differentiation, batching) and partial operand access (diagonal extraction) by simple graphical manipulations. We believe they are a versatile tool to improve the understanding of convolutions and will—due to their simplicity and flexibility—open up new algorithmic possibilities for them.

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
