# CONVOLUTIONS THROUGH THE LENS OF TENSOR NETWORKS (SUPPLEMENTARY MATERIAL)

In this supplementary material we provide additional details of the main text. Overview:

## A  LIMITATIONS

Here we comment on limitations on our approach.

**No common sub-expression elimination (CSE):**  Our implementation relies on `opt_einsum` which focuses on contraction order optimization. This optimization is efficient when all operands are different. However, with multiple occurrences of operands, computing shared sub-expressions might be an advantageous optimization approach which `opt_einsum` does not account for. The second-order quantity TNs from §C and §3.3 contain such sub-expressions, for instance $[\![\mathbf{X}]\!]$ and $1_{O_1 O_2}^\top [\![\mathbf{X}]\!]$ in KFAC-expand and KFAC-reduce, and $\mathbf{S}^{(\mathbf{W})}$ in the GGN quantities from Figure C15. The efficiency of CSE depends on how costly the shared tensor is to compute. For instance, computing $\mathbf{S}^{(\mathbf{W})}$ is expensive and therefore CSE is the more suitable optimization technique. For the input-based Kronecker factors which require the unfolded input, either contraction path optimization or CSE might be better. This is because the optimal contraction order may not correspond to 2x input unfolding and exhibit more parallelism which may lead to faster run times on a GPU. It would be interesting to integrate CSE into the contraction path optimization and develop a heuristic to choose a contraction path, for instance based on a weighted sum of FLOPs and memory.

**No index slicing:**  We mention index slicing as a technique to reduce peak memory of, and distribute, TN contractions. However, our implementation does not use index slicing, although there are packages like `cotengra` Gray & Kourtis (2021) with an interface similar to `opt_einsum`. We did not experiment with index slicing as our benchmark uses a single GPU and did not encounter out-of-memory errors. Still, we mention this technique, as, in combination with CSE, it could automatically reduce peak memory of the GGN quantities from Figure C15 which suffer from high memory requirements.

## B  VISUAL TOUR OF TENSOR NETWORK OPERATIONS FOR CONVOLUTIONS

Here, we extend the presented operations with a batch axis and allow for grouped convolutions.

### B.1  CONVOLUTION & FIRST-ORDER DERIVATIVES

**Adding a batch dimension (`vmap`-ing):**  Adding a batch axis to all presented operations is trivial. We only need to add an additional leg to the batched tensors, and connect these legs via element-wise or inner multiplication, depending on whether the result tensor is batched or not.

**Grouped convolutions:**  Grouped convolutions were originally proposed by Krizhevsky et al. (2012) and allow for parallelizing, distributing, and reducing the parameters of the convolution operation. They split $C_{\text{in}}$ input channels into $G$ groups of size $\tilde{C}_{\text{in}} := {}^{C_{\text{in}}}/G$, then perform independent convolutions per group, each producing $\tilde{C}_{\text{out}} := {}^{C_{\text{out}}}/G$ output channels which are concatenated in the output. Each group uses a kernel $\mathbf{W}_g$ of size $\tilde{C}_{\text{out}} \times \tilde{C}_{\text{in}} \times K_1 \times K_2$. These kernels are stacked into a single tensor $\mathbf{W} \in \mathbb{R}^{C_{\text{out}}, \tilde{C}_{\text{in}}, K_1, K_2}$ such that $[\mathbf{W}]_{(g,:),:,:,:} = \mathbf{W}_g$. To support groups, we thus decompose the channel indices into $c_{\text{in}} := (\tilde{c}_{\text{in}}, g)$ and $c_{\text{out}} := (\tilde{c}_{\text{out}}, g)$. For the forward pass this yields the grouped convolution (without bias)

$$Y_{(g,\tilde{c}_{\text{out}}),o_1,o_2} = \sum_{i_1,i_2,\tilde{c}_{\text{in}},k_1,k_2} X_{(g,\tilde{c}_{\text{in}}),i_1,i_2} \Pi^{(1)}_{i_1,o_1,k_1} \Pi^{(2)}_{i_2,o_2,k_2} W_{(g,\tilde{c}_{\text{out}}),c_{\text{in}},k_1,k_2} . \tag{B9}$$

Figure B9a shows the batched version of Equation (B9) as TN. Applying the differentiation rule from §3 leads to the Jacobians and VJPs shown in the remaining panels of Figure B9.

### B.2  EXACT SECOND-ORDER INFORMATION

In Figure B12 we show the TNs for the GGN diagonal and the GGN Gram matrix (empirical NTK matrix) from Figure C15 extended by channel groups and a batch axis.

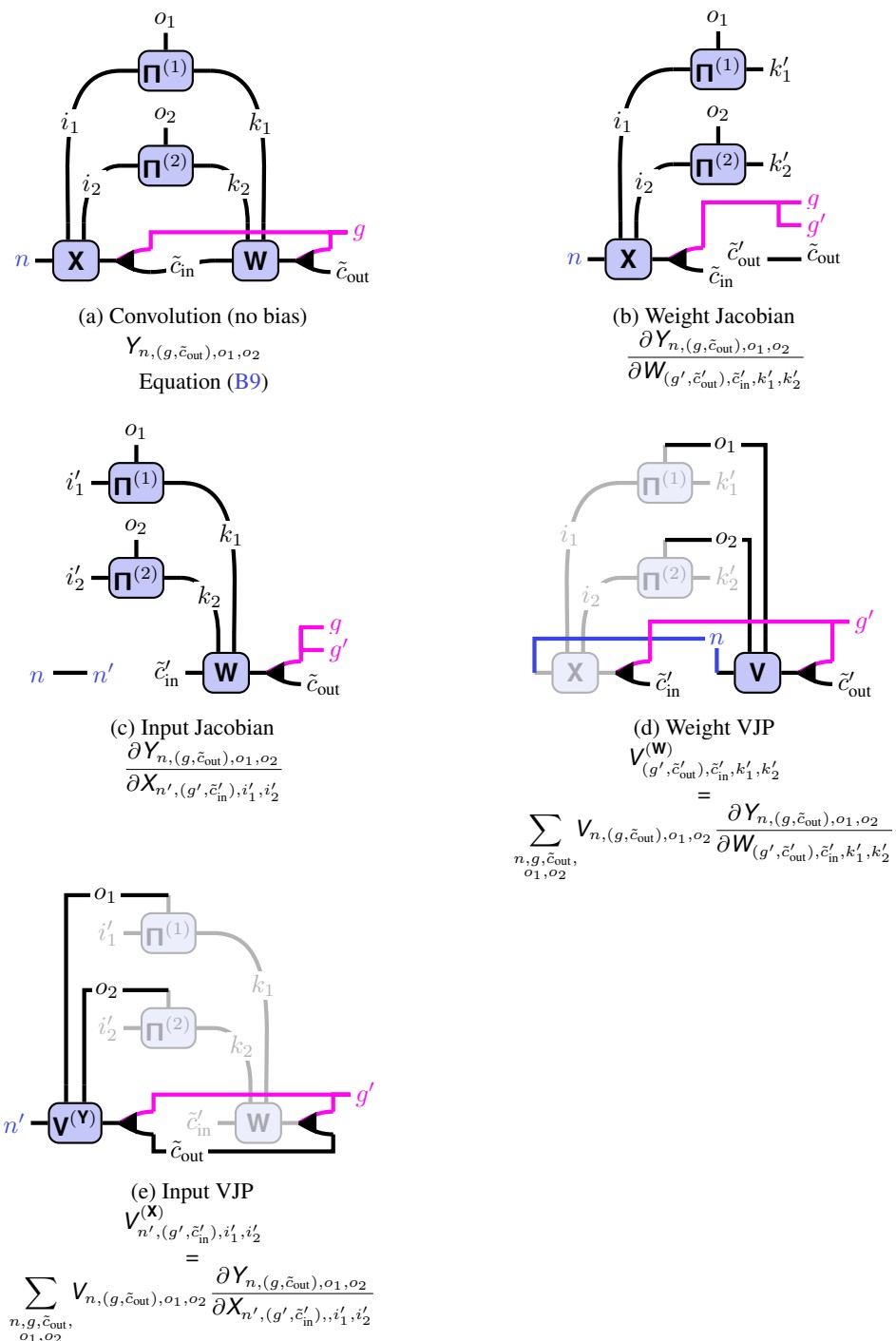

Figure B9: TNs of the (a) forward pass, (b, c) Jacobians, and (d, e) VJPs with batch axis and channel groups. They generalize Figures 2 to 4 from the main text. For the VJPs, the Jacobians are shaded.

**Diagonal block extraction:** Combined with index un-grouping, diagonal extraction generalizes to larger blocks: Let $\boldsymbol{A} \in \mathbb{R}^{KI \times KJ}$ be a matrix of $K$ horizontally and vertically concatenated blocks $\boldsymbol{A}^{(k_1, k_2)} \in \mathbb{R}^{I \times J}, k_i = 1 \ldots, K$. We can extract the diagonal blocks by restoring the sub-structure,

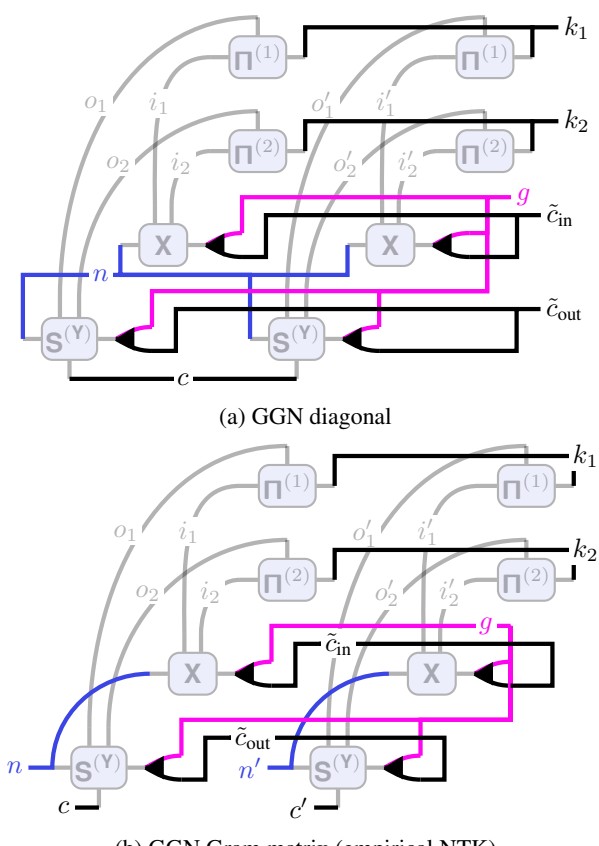

(a) GGN diagonal

(b) GGN Gram matrix (empirical NTK)

Figure B10: TNs of (a) the GGN diagonal and (b) the GGN Gram matrix with batching and channel groups. They extend Figures C15b and C15c from the main text.

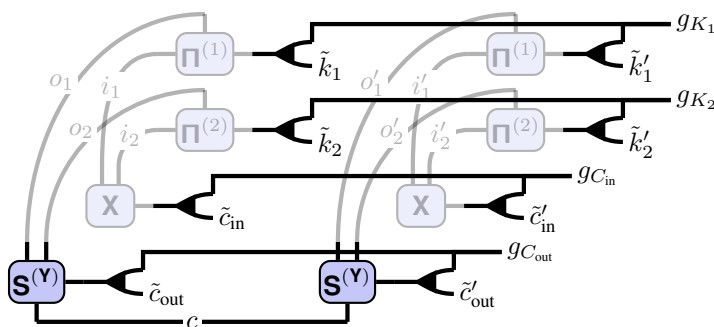

Figure B11: TN of a GGN mini-block diagonal without batching and channel groups.

then taking the diagonal along the $K$-dimensional index,

$$
k\!-\!\{A^{(k,k)}\}\!-\!j \;\; \underset{i}{\overset{k}{=}} \;\; (k,i)\!-\!A\!-\!(k,j)\!\!\overset{}{\underset{}{\longleftarrow}}\!j\,.
$$

We can apply this procedure to the GGN from Figure C15a. Assume we want to divide the output channel, input channel, and spatial dimensions into $G_{C_{\text{out}}}, G_{C_{\text{in}}}, G_{K_1}, G_{K_2}$ groups. A group will thus be indexed with a tuple $(g_{C_{\text{out}}}, g_{C_{\text{in}}}, g_{K_1}, g_{K_2})$ and the corresponding GGN block will be of dimension $C_{\text{out}}/G_{C_{\text{out}}} \times C_{\text{in}}/G_{C_{\text{in}}} \times K_1/G_{K_1} \times K_2/G_{K_2} \times C_{\text{out}}/G_{C_{\text{out}}} \times C_{\text{in}}/G_{C_{\text{in}}} \times K_1/G_{K_1} \times K_2/G_{K_2}$ and correspond to the GGN for $[\mathbf{W}]_{(g_{C_{\text{out}}},:),(g_{C_{\text{in}}},:),(g_{K_1},:),(g_{K_2},:)}$. This process of un-grouping the output dimensions, then taking the diagonal along the group indices, is illustrated in Figure B11. Note that if we choose $G_{C_{\text{out}}} = C_{\text{out}}, G_{C_{\text{in}}} = C_{\text{in}}, G_{K_1} = K_1, G_{K_2} = K_2$, each block will be a single number

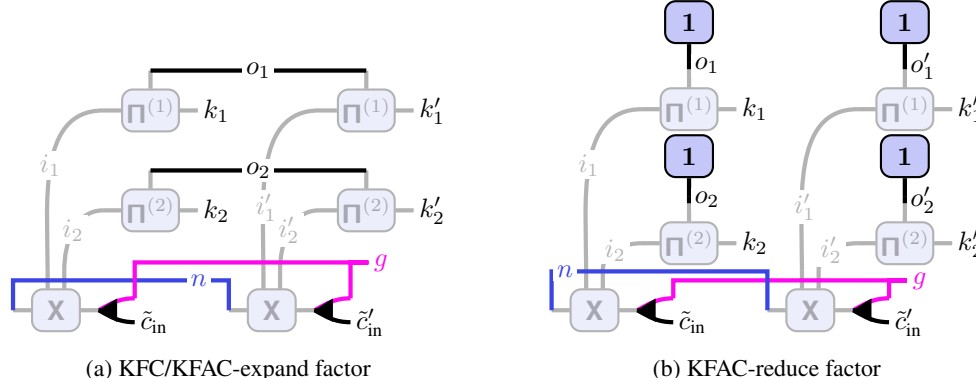

(a) KFC/KFAC-expand factor          (b) KFAC-reduce factor

Figure B12: TN diagrams of input-based factors in Kronecker approximations of the GGN for convolutions with batching and channel groups. They extend Figure 5 from the main text.

and hence we recover the GGN diagonal from Figure C15b. If instead we $G_{C_{\text{out}}} = G_{C_{\text{in}}} G_{K_1} G_{K_2} = 1$, we obtain the full GGN from Figure C15a. The outlined schemes allows to extract mini-blocks of arbitrary size along the diagonal (subject to the total dimension).

## B.3 KRONECKER-FACTORED APPROXIMATE CURVATURE (KFAC)

We were unable to find a definition of KFAC for grouped convolutions. Hence, we derive it here and present the TN diagrams. We use the perspective that grouped convolutions are independent convolutions over channel groups which are then concatenated. For each of those convolutions, we can then apply established the KFAC approximation for convolutions without groups. For a group $g$ we have the kernel $\mathbf{W}_g = [\mathbf{W}]_{(g,:),:,:,:}$ and the unfolded input of its associated input channels, $[\![\mathbf{X}_g]\!] = [\![\mathbf{X}]\!]_{(g,:),:,:} = [\![\mathbf{X}]_{(g,:),:,:,:}]\!]$ (or $[\![\mathbf{X}_{n,g}]\!] = [\![\mathbf{X}_n]\!]_{(g,:),:,:} = [\![\mathbf{X}]_{n,(g,:),:,:}]\!]$ in the batched setting).

**KFC/KFAC-expand grouped convolutions:** Applying the regular KFC approximation to the kernel of group $g$, this yields the Fisher approximation $\mathbf{\Omega}_g \otimes \mathbf{\Gamma}_g$ with $\mathbf{\Gamma}_g \in \mathbb{R}^{\tilde{C}_{\text{out}} \times \tilde{C}_{\text{out}}}$ and $\mathbf{\Omega}_g = 1/N \sum_{n=1}^{N} [\![\mathbf{X}_{n,g}]\!]^\top [\![\mathbf{X}_{n,g}]\!] \in \mathbb{R}^{\tilde{C}_{\text{in}} K_1 K_2 \times \tilde{C}_{\text{in}} K_1 K_2}$ where $\mathbf{X}_{n,g}$ is the input tensor for sample $n$ and group $g$ (remember the index structure $\mathbf{X}_{n,(g,\tilde{c}_{\text{in}}),i_1,i_2}$). Figure B12a shows the diagram for $\{N\mathbf{\Omega}_g\}_{g=1}^{G}$.

**KFAC-reduce for grouped convolutions:** Proceeding in the same way, but using the unfolded input averaged over output locations, we obtain the Fisher approximation $\hat{\mathbf{\Omega}}_g \otimes \hat{\mathbf{\Gamma}}_g$ with $\hat{\mathbf{\Gamma}}_g \in \mathbb{R}^{\tilde{C}_{\text{out}} \times \tilde{C}_{\text{out}}}$ and $\hat{\mathbf{\Omega}}_g = 1/N(O_1 O_2)^2 \sum_{n=1}^{N} (\mathbf{1}_{O_1 O_2}^\top [\![\mathbf{X}_{n,g}]\!])^\top \mathbf{1}_{O_1 O_2}^\top [\![\mathbf{X}_{n,g}]\!] \in \mathbb{R}^{\tilde{C}_{\text{in}} K_1 K_2 \times \tilde{C}_{\text{in}} K_1 K_2}$ for the kernel of group $g$. Figure B12b shows the diagram for $\{N(O_1 O_2)^2 \hat{\mathbf{\Omega}}_g\}_{g=1}^{G}$.

## B.4 FURTHER OPERATIONS & EXTENSIVE OVERVIEW

**Consecutive convolutions:** We can chain two, or more, convolutions into a single TN diagram (Figure B13) to obtain a deep linear CNN Singh et al. (2023) similar to deep linear networks which are popular for analytical studies.

**Convolution weight/input JVPs:** In the main text, we derived the Jacobians of convolution (§3.1) which can be used to derive the JVPs. A JVP propagates perturbations $\mathbf{V}^{(\mathbf{W})} \in \mathbb{R}^{C_{\text{out}} \times C_{\text{in}} \times K_1 \times K_2}$ and $\mathbf{V}^{(\mathbf{X})} \in \mathbb{R}^{C_{\text{in}} \times I_1 \times I_2}$ in the input space to perturbations in the output space by contracting the perturbation with the Jacobian. See Table B2 for the general einsum expressions.

**Batched convolution weight VJP:** To obtain per-sample gradients, the weight VJP must be carried out without summing over the batch axis which amounts to keeping the batch index in the output index tuple.

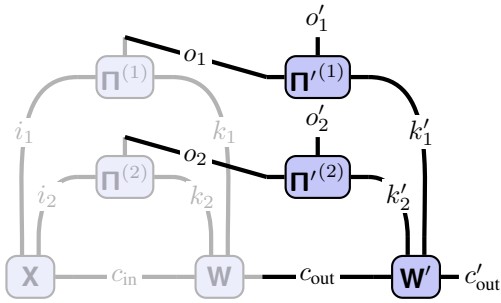

Figure B13: TN of two consecutive convolutions without groups and without batch axis.

**VJPs and JVPs of `im2col`:** With the TN differentiation technique described in §3.1 we can compute the Jacobian of the unfolding operation, then contract it with perturbations $V^{(\mathbf{X})} \in \mathbb{R}^{C_{\text{in}} \times K_1 \times K_2}$ in input space to obtain the JVP, or with perturbations $V^{(\llbracket \mathbf{X} \rrbracket)} \in \mathbb{R}^{O_1 O_2 \times C_{\text{in}} K_1 K_2}$ to obtain the VJP.

**Approximate Hessian diagonals (HesScale/BL89):** Becker & Lecun (1989); Elsayed & Mahmood (2023) proposed approximate procedures for the Hessian diagonal which cost roughly a gradient. They can be understood as modifications of the Hessian backpropagation equations from Dangel et al. (2020a).

Consider a layer with input $\boldsymbol{x}$, output $\boldsymbol{y}$, and weights $\boldsymbol{w}$ inside a sequential feedforward neural network (for a convolutional layer, these correspond to the flattened input, output, and kernel). To compute per-layer Hessians of a loss $\ell$, each layer backpropagates its incoming Hessian $\nabla_{\boldsymbol{y}}^2 \ell$ according to Dangel et al. (2020a)

$$
\begin{aligned}
\nabla_{\boldsymbol{x}}^2 \ell &= (\boldsymbol{J_x y})^\top \nabla_{\boldsymbol{y}}^2 \ell (\boldsymbol{J_x y}) + \sum_i \frac{\partial \ell}{\partial y_i} \nabla_{\boldsymbol{x}}^2 y_i \,, \\
\nabla_{\boldsymbol{w}}^2 \ell &= (\boldsymbol{J_w y})^\top \nabla_{\boldsymbol{y}}^2 \ell (\boldsymbol{J_w y}) + \sum_i \frac{\partial \ell}{\partial y_i} \nabla_{\boldsymbol{w}}^2 y_i \,.
\end{aligned}
\tag{B10}
$$

The scheme of Becker & Lecun (1989); Elsayed & Mahmood (2023) imposes diagonal structure on the backpropagated quantity. A layer receives a backpropagated diagonal $\boldsymbol{d}^{(\boldsymbol{y})}$ such that $\operatorname{diag}(\boldsymbol{d}^{(\boldsymbol{y})}) \approx \nabla_{\boldsymbol{y}}^2 \ell$, and backpropagates it according to Equation (B10), but with a post-processing step to obtain a diagonal backpropagated quantity,

$$
\begin{aligned}
\boldsymbol{d}^{(\boldsymbol{x})} &= \operatorname{diag}\left( (\boldsymbol{J_x y})^\top \operatorname{diag}(\boldsymbol{d}^{(\boldsymbol{y})})(\boldsymbol{J_x y}) \right) + \operatorname{diag}\left( \sum_i \frac{\partial \ell}{\partial y_i} \nabla_{\boldsymbol{x}}^2 y_i \right) \,, \\
\boldsymbol{d}^{(\boldsymbol{w})} &= \operatorname{diag}\left( (\boldsymbol{J_w y})^\top \operatorname{diag}(\boldsymbol{d}^{(\boldsymbol{w})})(\boldsymbol{J_w y}) \right) + \operatorname{diag}\left( \sum_i \frac{\partial \ell}{\partial y_i} \nabla_{\boldsymbol{w}}^2 y_i \right) \,,
\end{aligned}
\tag{B11}
$$

where $\operatorname{diag}(\boldsymbol{d}^{(\boldsymbol{x})}) \approx \nabla_{\boldsymbol{x}}^2 \ell$ and $\operatorname{diag}(\boldsymbol{d}^{(\boldsymbol{w})}) \approx \nabla_{\boldsymbol{w}}^2 \ell$ is an approximation to the Hessian diagonal.

For convolutional layers, which are linear in the input and weight, the second summands are zero due to $\nabla_{\boldsymbol{x}}^2 y_i = \boldsymbol{0} = \nabla_{\boldsymbol{w}}^2 y_i$. The first terms of Equation (B11) require (i) embedding a diagonal vector into a matrix, (ii) applying MJPs and JMPs, and (iii) extracting the result's diagonal. Those can be expressed as a single TN. We show the diagrams in Figure B14, using tensors rather than their flattened versions, that is $(\boldsymbol{x}, \boldsymbol{y}, \boldsymbol{w}, \boldsymbol{d}^{(\boldsymbol{x})}, \boldsymbol{d}^{(\boldsymbol{y})}, \boldsymbol{d}^{(\boldsymbol{w})}) \to (\mathbf{X}, \mathbf{Y}, \mathbf{W}, \mathbf{D}^{(\mathbf{X})}, \mathbf{D}^{(\mathbf{Y})}, \mathbf{D}^{(\mathbf{W})})$.

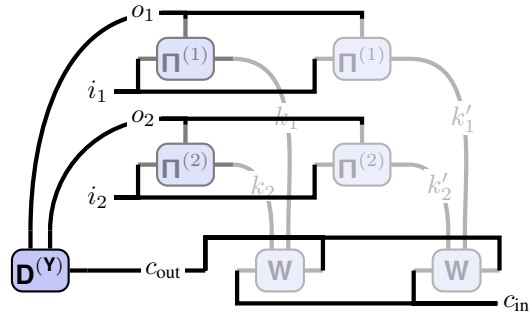

(a) HesScale/BL89 input backpropagation

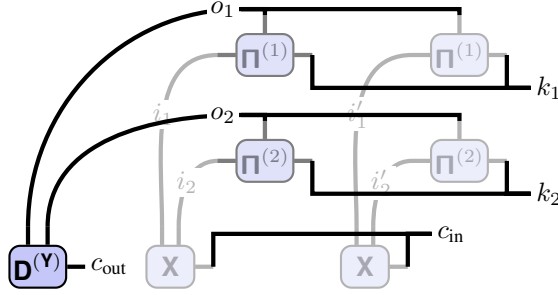

(b) HesScale/BL89 weight backpropagation

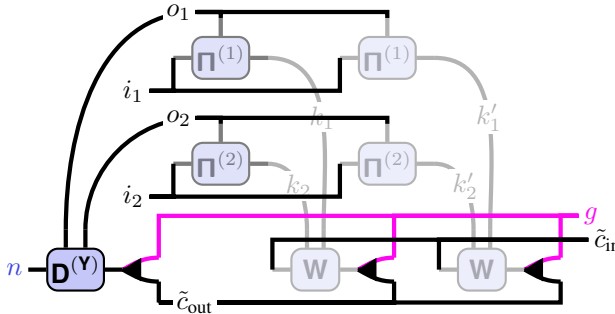

(c) HesScale/BL89 input backpropagation (+ batch, groups)

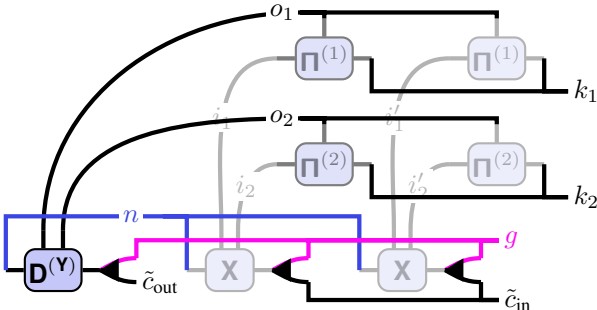

(d) HesScale/BL89 weight backpropagation (+ batch, groups)

Figure B14: TN diagrams for HesScale/BL89 Becker & Lecun (1989); Elsayed & Mahmood (2023) backpropagations through convolutional layers to approximate the Hessian diagonals $\mathbf{D}^{(\mathbf{X})}$, $\mathbf{D}^{(\mathbf{W})}$. JMPs and MJPs are shaded. (a, b) show the simple versions without batching and without channel groups. (c, d) include batching and channel groups.

Table B2: Extensive list of convolution and related operations (extension from Table 1 in the main text). All operations consider two spatial dimensions and support batching and channel groups. Generalization to other dimensions follow by introducing more spatial indices `i_3, o_3, ...` and kernel indices `k_3, ....`

| Operation | Operands | Contraction string (**einops** Rogozhnikov (2022) **convention**) |
|---|---|---|
| Convolution (no bias) Hayashi et al. (2019) | $\mathbf{X}, \mathbf{\Pi}^{(1)}, \mathbf{\Pi}^{(2)}, \mathbf{W}$ | `"n (g c_in) i1 i2, i1 o1 k1, i2 o2 k2, (g c_out) c_in k1 k2 -> n (g c_out) o1 o2"` |
| Unfolded input (`im2col`) | $\mathbf{X}, \mathbf{\Pi}^{(1)}, \mathbf{\Pi}^{(2)}$ | `"n c_in i1 i2, i1 o1 k1, i2 o2 k2 -> n (c_in k1 k2) (o1 o2)"` |
| Unfolded kernel (Toeplitz) | $\mathbf{\Pi}^{(1)}, \mathbf{\Pi}^{(2)}, \mathbf{W}$ | `"i1 o1 k1, i2 o2 k2, c_out c_in k1 k2 -> (c_out o1 o2) (c_in i1 i2)"` |
| Convolution weight VJP | $\mathbf{X}, \mathbf{\Pi}^{(1)}, \mathbf{\Pi}^{(2)}, \mathbf{V}^{(\mathbf{Y})}$ | `"n (g c_in) i1 i2, i1 o1 k1, i2 o2 k2, n (g c_out) o1 o2 -> c_out c_in k1 k2"` |
| Convolution input VJP (transpose convolution) | $\mathbf{W}, \mathbf{\Pi}^{(1)}, \mathbf{\Pi}^{(2)}, \mathbf{V}^{(\mathbf{Y})}$ | `"(g c_out) c_in k1 k2, i1 o1 k1, i2 o2 k2, n (g c_out) o1 o2 -> n (g c_in) i1 i2"` |
| Convolution weight VJP (per-sample/batched) Rochette et al. (2019) | $\mathbf{X}, \mathbf{\Pi}^{(1)}, \mathbf{\Pi}^{(2)}, \mathbf{V}^{(\mathbf{Y})}$ | `"n (g c_in) i1 i2, i1 o1 k1, i2 o2 k2, n (g c_out) o1 o2 -> n (g c_out) c_in k1 k2"` |
| Convolution weight JVP | $\mathbf{X}, \mathbf{\Pi}^{(1)}, \mathbf{\Pi}^{(2)}, \mathbf{V}^{(\mathbf{W})}$ | `"n (g c_in) i1 i2, i1 o1 k1, i2 o2 k2, (g c_out) c_in k1 k2 -> n (g c_out) o1 o2"` |
| Convolution input JVP | $\mathbf{W}, \mathbf{\Pi}^{(1)}, \mathbf{\Pi}^{(2)}, \mathbf{V}^{(\mathbf{X})}$ | `"(g c_out) c_in i1 i2, i1 o1 k1, i2 o2 k2, n (g c_in) i1 i2 -> n (g c_out) o1 o2"` |
| `im2col` VJP | $\mathbf{\Pi}^{(1)}, \mathbf{\Pi}^{(2)}, \mathbf{V}^{([\![\mathbf{X}]\!])}$ | `"i1 o1 k1, i2 o2 k2, n (c_in k1 k2) (o1 o2) -> n c_in i1 i2"` |
| `im2col` JVP | $\mathbf{\Pi}^{(1)}, \mathbf{\Pi}^{(2)}, \mathbf{V}^{(\mathbf{X})}$ | `"i1 o1 k1, i2 o2 k2, n c_in i1 i2 -> n (c_in k1 k2) (o1 o2)"` |
| KFC/KFAC-expand Grosse & Martens (2016); Eschenhagen (2022) | $\mathbf{X}, \mathbf{\Pi}^{(1)}, \mathbf{\Pi}^{(2)}, \mathbf{X}, \mathbf{\Pi}^{(1)}, \mathbf{\Pi}^{(2)}$ | `"n (g c_in) i1 i2, i1 o1 k1, i2 o2 k2, n (g c_in_) i1_ i2_, i1_ o1 k1_, i2_ o2 k2_ -> g (c_in k1 k2) (c_in_ k1_ k2_)"` |
| KFAC-reduce Eschenhagen (2022) | $\mathbf{X}, \mathbf{\Pi}^{(1)}, \mathbf{\Pi}^{(2)}, \mathbf{X}, \mathbf{\Pi}^{(1)}, \mathbf{\Pi}^{(2)}$ | `"n (g c_in) i1 i2, i1 o1 k1, i2 o2 k2, n (g c_in_) i1_ i2_, i1_ o1_ k1_, i2_ o2_ k2_ -> g (c k1 k2) (c_ k1_ k2_)"` |
| GGN Gram/empirical NTK matrix Dangel et al. (2022); Osawa et al. (2023); Novak et al. (2022) | $\mathbf{X}, \mathbf{\Pi}^{(1)}, \mathbf{\Pi}^{(2)}, \mathbf{S}^{(\mathbf{Y})}, \mathbf{X}, \mathbf{\Pi}^{(1)}, \mathbf{\Pi}^{(2)}, \mathbf{S}^{(\mathbf{Y})}$ | `"n (g c_in) i1 i2, i1 o1 k1, i2 o2 k2, c n (g c_out) o1 o2, n_ (g c_in) i1_ i2_, i1_ o1_ k1, i2_ o2_ k2, c_ n_ (g c_out) o1_ o2_ -> (c n) (c_ n_)"` |
| GGN/Fisher diagonal Dangel et al. (2020b); Osawa et al. (2023) | $\mathbf{X}, \mathbf{\Pi}^{(1)}, \mathbf{\Pi}^{(2)}, \mathbf{S}^{(\mathbf{Y})}, \mathbf{X}, \mathbf{\Pi}^{(1)}, \mathbf{\Pi}^{(2)}, \mathbf{S}^{(\mathbf{Y})}$ | `"n (g c_in) i1 i2, i1 o1 k1, i2 o2 k2, c n (g c_out) o1 o2, n (g c_in) i1_ i2_, i1_ o1_ k1, i2_ o2_ k2, c n (g c_out) o1_ o2_ -> (g c_out) c_in k1 k2"` |
| GGN/Fisher diagonal (per-sample/batched) | $\mathbf{X}, \mathbf{\Pi}^{(1)}, \mathbf{\Pi}^{(2)}, \mathbf{S}^{(\mathbf{Y})}, \mathbf{X}, \mathbf{\Pi}^{(1)}, \mathbf{\Pi}^{(2)}, \mathbf{S}^{(\mathbf{Y})}$ | `"n (g c_in) i1 i2, i1 o1 k1, i2 o2 k2, c n (g c_out) o1 o2, n (g c_in) i1_ i2_, i1_ o1_ k1, i2_ o2_ k2, c n (g c_out) o1_ o2_ -> n (g c_out) c_in k1 k2"` |
| Approximate weight Hessian diagonal Becker & Lecun (1989); Elsayed & Mahmood (2023) | $\mathbf{X}, \mathbf{\Pi}^{(1)}, \mathbf{\Pi}^{(2)}, \mathbf{D}^{(\mathbf{Y})}, \mathbf{X}, \mathbf{\Pi}^{(1)}, \mathbf{\Pi}^{(2)}$ | `"n (g c_in) i1 i2, i1 o1 k1, i2 o2 k2, n (g c_out) o1 o2, n (g c_in) i1_ i2_, i1_ o1 k1, i2_ o2 k2 -> (g c_out) c_in k1 k2"` |
| Approximate input Hessian diagonal Becker & Lecun (1989); Elsayed & Mahmood (2023) | $\mathbf{W}, \mathbf{\Pi}^{(1)}, \mathbf{\Pi}^{(2)}, \mathbf{D}^{(\mathbf{Y})}, \mathbf{W}, \mathbf{\Pi}^{(1)}, \mathbf{\Pi}^{(2)}$ | `"(g c_out) c_in k1 k2, i1 o1 k1, i2 o2 k2, n (g c_out) o1 o2, (g c_out) c_in k1_ k2_, i1 o1 k1_, i2 o2 k2_ -> n (g c_in) i1 i2"` |
| Approximate weight Hessian diagonal (per-sample/batched) | $\mathbf{X}, \mathbf{\Pi}^{(1)}, \mathbf{\Pi}^{(2)}, \mathbf{D}^{(\mathbf{Y})}, \mathbf{X}, \mathbf{\Pi}^{(1)}, \mathbf{\Pi}^{(2)}$ | `"n (g c_in) i1 i2, i1 o1 k1, i2 o2 k2, n (g c_out) o1 o2, n (g c_in) i1_ i2_, i1_ o1 k1, i2_ o2 k2 -> n (g c_out) c_in k1 k2"` |

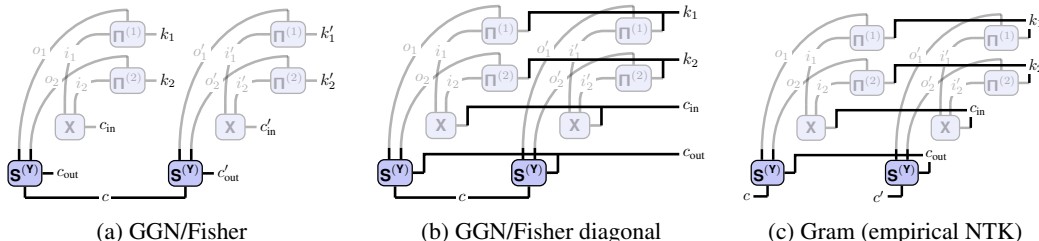

|  (a) GGN/Fisher | (b) GGN/Fisher diagonal | (c) Gram (empirical NTK) |

Figure C15: TN composition and sub-tensor extraction for second-order information. Weight MJPs from Figure 4a are shaded. (a) exact and (b) diagonal of the kernel's GGN (the same applies to structurally similar matrices like the gradient covariance Jastrzebski et al. (2020)). (c) TN of the GGN Gram matrix.

## C    EXACT SECOND-ORDER INFORMATION

Here, we look at computing second-order information of a loss w.r.t. to the kernel of a convolution. Its computation can be phrased as backpropagation with a final extraction step Dangel (2023) which contains less standard operations like Jacobian-matrix products (JMPs) and sub-tensor extraction. TNs can express this extraction step in a single diagram.

Consider a datum $(\boldsymbol{x}, \boldsymbol{t})$ and its loss $\ell(\boldsymbol{w}) = \ell(\boldsymbol{f}, \boldsymbol{t})$ where $\boldsymbol{f} := f_{\boldsymbol{w}}(\boldsymbol{x}) \in \mathbb{R}^C$ is the prediction of a CNN with a convolution with flattened kernel $\boldsymbol{w}$ and flattened output $\boldsymbol{y}$ (derivations carry over to a batch loss). The kernel's generalized Gauss-Newton (GGN) matrix Schraudolph (2002) $\boldsymbol{G}(\boldsymbol{w}) = (\boldsymbol{J}_{\boldsymbol{w}}\boldsymbol{f})^{\top} \nabla_{\boldsymbol{f}}^2 \ell (\boldsymbol{J}_{\boldsymbol{w}}\boldsymbol{f}) \in \mathbb{R}^{C_{\text{out}} C_{\text{in}} K_1 K_2 \times C_{\text{out}} C_{\text{in}} K_1 K_2}$ is a positive semi-definite Hessian proxy preferred by many applications (e.g. Daxberger et al., 2021; Martens, 2010) and coincides with the Fisher information matrix for many common losses Martens (2020). It is the self-outer product of a backpropagated symmetric factorization $\boldsymbol{S}^{(\boldsymbol{y})} = (\boldsymbol{J}_{\boldsymbol{y}}\boldsymbol{f})^{\top} \boldsymbol{S}^{(\boldsymbol{f})} \in \mathbb{R}^{C_{\text{out}} O_1 O_2 \times C}$ of the loss Hessian, $\nabla_{\boldsymbol{f}}^2 \ell(\boldsymbol{f}, \boldsymbol{y}) = \boldsymbol{S}^{(\boldsymbol{f})}(\boldsymbol{S}^{(\boldsymbol{f})})^{\top}$. During backpropagation, the convolution extracts information about $\boldsymbol{G}(\boldsymbol{w}) = (\boldsymbol{J}_{\boldsymbol{w}}\boldsymbol{y})^{\top} \boldsymbol{S}^{(\boldsymbol{y})}(\boldsymbol{S}^{(\boldsymbol{y})})^{\top} \boldsymbol{J}_{\boldsymbol{w}}\boldsymbol{y}$.

In TN notation, this is easy to express without flattening: We simply compose two VJP diagrams from Figure 4a with an extra leg (MJP) and add the outer-product contraction to obtain the tensor version $\boldsymbol{\mathsf{G}}(\boldsymbol{\mathsf{W}}) \in \mathbb{R}^{C_{\text{out}} \times C_{\text{in}} \times K_1 \times K_2 \times C_{\text{out}} \times C_{\text{in}} \times K_1 \times K_2}$ of $\boldsymbol{G}(\boldsymbol{w})$ (Figure C15a). The GGN is often further approximated by sub-tensors as it is too large. These slicing operations are also easy to integrate into the diagrams, e.g. to extract diagonal elements (Figure C15b Dangel et al. (2020b); Osawa et al. (2023)), or mini-block diagonals (Figure B11 (Dangel et al., 2020a; Bahamou et al., 2023)). This also removes redundant computations compared to computing, then slicing, the matrix. The same ideas apply to the GGN Gram matrix $(\boldsymbol{S}^{(\boldsymbol{w})})^{\top} \boldsymbol{S}^{(\boldsymbol{w})} \in \mathbb{R}^{C \times C}$ (Figure C15c). It contains the GGN spectrum Dangel et al. (2022) and is related to the empirical NTK for square loss Novak et al. (2022).

# D  IMPLEMENTATION DETAILS

Here we present details on the index pattern computation, and additional transformations.

## D.1  INDEX PATTERN TENSOR COMPUTATION FOR CONVOLUTIONS

Algorithm D1 lists pseudo-code for the index pattern computation from the convolution hyper-parameters $K, S, P, D$, and the spatial input dimension $I$, that is $\boldsymbol{\Pi}(I, K, S, P, D)$. *Unlike in the main text, we use 0-based indexing which is more common in numerical libraries.* For self-consistency, we re-state the relation of the hyper-parameters to output dimension from (Dumoulin & Visin, 2016, Relationship 15),

$$O(I, K, S, P, D) = 1 + \left\lfloor \frac{I + 2P - K - (K-1)(D-1)}{S} \right\rfloor . \tag{D12}$$

---

**Algorithm D1** Computing the convolution index pattern tensor $\boldsymbol{\Pi}$ for a spatial dimension.

---

**Require:** Input size $I \in \mathbb{N}^+$, kernel size $K \in \mathbb{N}^+$, stride $S \in \mathbb{N}^+$, padding $P \in \mathbb{N}_0^+$, dilation $D \in \mathbb{N}^+$

$O \leftarrow 1 + \left\lfloor \frac{I+2P-K-(K-1)(D-1)}{S} \right\rfloor$ ▷ Compute output dimension (Dumoulin & Visin, 2016, Relationship 15)

$\boldsymbol{\Pi} \leftarrow \mathbf{0}_{I \times O \times K}$ ▷ Initialize index pattern tensor

**for** $o = 0, \ldots, O-1, k = 0, \ldots, K-1$ **do** ▷ Use 0-based indexing!

  $i \leftarrow kD + oS - P$ ▷ Reconstruct contributing input element

  **if** $0 \leq i \leq I - 1$ **then** ▷ Check in bounds

    $\boldsymbol{\Pi}_{i,o,k} \leftarrow 1$

  **end if**

**end for**

**return** Index pattern tensor $\boldsymbol{\Pi} \in \{0, 1\}^{I \times O \times K}$

---

## D.2  INDEX PATTERN TENSOR FOR STANDALONE TRANSPOSE CONVOLUTION

Although a transpose convolution is defined w.r.t. a reference convolution with hyper-parameters $K, S, P, D$, most libraries offer standalone implementations of transpose convolution. We describe the transpose convolution by its associated convolution, that is as a mapping from $\mathbb{R}^{C_{\text{out}} \times O_1 \times O_2}$ (the convolution's output space) to $\mathbb{R}^{C_{\text{in}} \times I_1 \times I_2}$ (the convolution's input space). For convolution with $S > 1$, we cannot infer $I$ from $O, K, S, P, D$, as multiple $I$s map to the same $O$ if $(I + 2P - K - (K-1)(D-1)) \mod S \neq 0$ (see the floor operation in Algorithm D1). We need to either supply $I$ directly, or the remainder

$$A = I + 2P - K - (K-1)(D-1) - S(O-1)$$

(often called `output_padding`) to make $I$ unambiguous. Then, we compute

$$I = (O-1)S - 2P + K + (K-1)(D-1) + A . \tag{D13}$$

to get $I(O, A)$ and call Algorithm D1 to obtain $\boldsymbol{\Pi}(I(O, A), K, S, P, D)$.

## D.3  DETAILS ON INDEX PATTERN SIMPLIFICATIONS

In the following, we will assume the absence of boundary pixels that don't overlap with the kernel, that is

$$I + 2P - (K + (K-1)(D-1)) \mod S = 0 , \tag{D14}$$

where the floor operation in $O(I, K, S, P, D)$ is obsolete. This can always be assured by narrowing $\mathbf{X}$ before a convolution. Based on our hyper-parameter analysis of real-world CNNs (§E), we identify:

**Transformation D1 (Dense convolutions)** *Assume Equation (D14). For $K = S$ with default padding and dilation ($P = 0$, $D = 1$), patches are adjacent non-overlapping tiles, accessible by un-grouping the input index $i$ into a tuple index $(\tilde{i}, \tilde{k})$ of size $I/K \times K$:*

$$[\boldsymbol{\Pi}(I, K, K, 0, 1)]_{i,o,k} = [\boldsymbol{\Pi}(I, K, K, 0, 1)]_{(\tilde{i}, \tilde{k}), o, k} = \delta_{\tilde{i}, o} \delta_{\tilde{k}, k} .$$

*Point-wise convolutions ($K = S = 1$) are a special case with pattern $[\boldsymbol{\Pi}(I, 1, 1, 0, 1)]_{i,o,k} = \delta_{i,o}$.*

Point-wise convolutions with $K = S = 1$ are common in DenseNets Huang et al. (2017), MobileNets Howard et al. (2017); Sandler et al. (2018) and ResNets He et al. (2016). InceptionV3 Szegedy et al. (2016) has 2d 'mixed dense' convolutions that are point-wise along one spatial dimension. ConvNeXt Liu et al. (2022) uses dense convolutions with $K = S \in \{2, 4\}$.

**Transformation D2 (Down-sampling convolutions)** *For $S > K$ with default padding and dilation ($P = 0$, $D = 1$), some elements do not overlap with the kernel. If the input dimension $i$ is summed, all participating tensors can be pruned to remove the explicit zeros. Assume $I \mod S = 0$. Then, pruning amounts to un-grouping $i$ into $(i', s)$ of size $I/s \times S$, narrowing $s$ to $K$ entries, and grouping back into an index $\tilde{i}$ of size $KI/s$. After pruning, the index pattern represents a dense convolution with input size $KI/s$, kernel size $K$, and stride $K$. In a contraction with some tensor $\mathbf{V}$,*

$$\sum_{i=1}^{I} [\mathbf{V}]_{...,i,...} [\mathbf{\Pi}(I, K, S > K, 0, 1)]_{i,o,k} = \sum_{i=1}^{I/S} [\tilde{\mathbf{V}}]_{...,\tilde{i},...} [\mathbf{\Pi}(KI/s, K, K, 0, 1)]_{\tilde{i},o,k}$$

*with sub-tensor $[\tilde{\mathbf{V}}]_{...,\tilde{i},...} = [[\mathbf{V}]_{...,(i',s),...}]_{...,(:,:K),...}$ where $:K$ means narrowing to $K$ elements.*

Transformation D2 converts down-sampling convolutions to dense convolutions, which can be further simplified with Transformation D1. We find down-sampling convolutions with $S = 2 > K = 1$ in ResNet18 He et al. (2016), ResNext101 Xie et al. (2017), and WideResNet101 Zagoruyko & Komodakis (2016). Those convolutions discard 75 % of their input! Knowledge that an operation only consumes a fraction of its input could be used to eliminate those 'dead' computations in preceding operations, reducing FLOPS and memory.

**Transformation D3 (Kernel-output dimension swap)** *Assume Equation (D14). Transposing kernel and output dimensions in an index pattern yields another index pattern with same input size, kernel size $O(I, K, S, P, D)$, and swapped stride and dilation:*

$$[\mathbf{\Pi}(I, K, S, P, D)]_{i,o,k} = [\mathbf{\Pi}(I, O, D, P, S)]_{i,k,o} .$$

This transformation is easy to see from the symmetry of $(k, D)$ and $(o, S)$ in Equation (8) and $O(I, K, S, P, D)$. It converts index pattern contractions over output into kernel dimensions, like in convolutions. An example is the weight VJP from Figure 4a, which—after swapping kernel and output dimensions—resembles the TN for convolution from Figure 2 with kernel $\mathbf{V}$. Rochette et al. (2019) use this to phrase the computation of per-example gradients as convolution.

§D.3 presents more properties of $\mathbf{\Pi}$ based on the sub-sampling interpretation of stride and dilation along the output and kernel dimensions. We also provide a transformation for swapping input and output dimensions, relating convolution and transpose convolution as described in Dumoulin & Visin (2016).

For completeness, we state additional index pattern tensor properties here (using 1-based indexing):

**Transformation D4 (Sub-sampling interpretation of stride)** *Strided convolutions ($S > 1$) subsample non-strided convolutions along the output dimension, ignoring all but every $S$th output Dumoulin & Visin (2016). In other words, $[\mathbf{\Pi}(I, K, S, P, D)]_{i,o,k} = [\mathbf{\Pi}(I, K, 1, P, D)]_{i,1+S(o-1),k}$ or, in tensor notation ($[\cdot]_{::S}$ denotes slicing with steps of $S$),*

$$\mathbf{\Pi}(I, K, S, P, D) = [\mathbf{\Pi}(I, K, 1, P, D)]_{:,::S,:} .$$

**Transformation D5 (Sub-sampling interpretation of dilation)** *Dilated convolutions ($D > 1$) with kernel size $K$ sub-sample the kernel of a non-dilated convolution of kernel size $K + (D - 1)(K - 1)$, ignoring all but every $D$th kernel element. In other words, $[\mathbf{\Pi}(I, K, S, P, D)]_{i,o,k} = [\mathbf{\Pi}(I, K + (K - 1)(D - 1), S, P, 1)]_{i,o,1+D(k-1)}$ or, in tensor notation,*

$$\mathbf{\Pi}(I, K, S, P, D) = [\mathbf{\Pi}(I, K + (K - 1)(D - 1), S, P, 1)]_{:,:,::D} .$$

**Transformation D6 (Transpose convolution as convolution)** *Assume Equation (D14). Consider a non-strided ($S = 1$), non-dilated ($D = 1$) convolution with index pattern $\mathbf{\Pi}(I, K, 1, P, 1)$ and output dimension $O(I, K, 1, P, 1)$. Transposing the spatial dimensions and flipping the kernel dimension yields another index pattern with modified padding $P' = K - P - 1$. In other words, for all $i = 1, \ldots, I$, $k = 1, \ldots, K$, $o = 1, \ldots, O$*

$$[\mathbf{\Pi}(I, K, 1, P, 1)]_{i,o,k} = [\mathbf{\Pi}(O, K, 1, P', 1)]_{o,i,K+1-k} .$$

# E  CONVOLUTION LAYER HYPER-PARAMETER ANALYSIS

Here we give an overview of and characterize convolutions in popular architectures (see Table E3). We include moderately deep CNNs on Fashion MNIST, CIFAR-10, and CIFAR-100 from the DeepOBS benchmark Schneider et al. (2019), and deep CNNs on ImageNet (AlexNet, ResNet18, InceptionV3, MobileNetV2, ResNext101). Regarding the hyper-parameters, we make the following observations:

- Many CNNs do not use a bias term. This is because the output of those layers feeds directly into a batch normalization layer, which is invariant under the addition of a bias term.

- All investigated convolutions use default dilation.

- Group convolutions are rarely used. MobileNetV2 and ConvNeXt-base (Tables E3g and E3i) use group convolutions that interpret each individual channel as a group. ResNext101 (Table E3f) uses group convolutions that interpret a collection of channels as a group. ConvNeXt-base (Table E3g) uses dense convolutions with $P = 0$ and $S = K \in \{2, 4\}$.

- Many networks use dense convolutions, that is convolutions with unit kernel size ($K = 1$), unit stride ($S = 1$), and no padding ($P = 0$). These convolutions have a trivial index pattern and can therefore be simplified.

- InceptionV3 (Table E3h) uses two-dimensional convolutions with one trivial dimension ('mixed dense') with unit kernel size, unit stride, and no padding along one direction. For this spatial dimension, the index pattern can be simplified.

- ResNet18 (Table E3e) and ResNext101 (Table E3f) use convolutions with $S > K$ for down-sampling whose kernel only overlaps with a fraction of the input. The index pattern can be simplified.

Table E3: Hyper-parameters of convolutions in different CNNs. For convolutions with identical hyper-parameters, we only show one instance and its multiplicity.

(a) 3c3d, CIFAR-10 (3, 32, 32)

| Name (count) | Input shape | Output shape | Kernel | Stride | Padding | Dilation | Groups | Bias | Type |
|---|---|---|---|---|---|---|---|---|---|
| conv1.0 (1) | (3, 32, 32) | (64, 28, 28) | (5, 5) | (1, 1) | (0, 0) | (1, 1) | 1 | Yes | General |
| conv2.0 (1) | (64, 14, 14) | (96, 12, 12) | (3, 3) | (1, 1) | (0, 0) | (1, 1) | 1 | Yes | General |
| conv3.1 (1) | (96, 8, 8) | (128, 6, 6) | (3, 3) | (1, 1) | (0, 0) | (1, 1) | 1 | Yes | General |

(b) 2c2d, Fashion MNIST (1, 28, 28)

| Name (count) | Input shape | Output shape | Kernel | Stride | Padding | Dilation | Groups | Bias | Type |
|---|---|---|---|---|---|---|---|---|---|
| conv1.1 (1) | (1, 32, 32) | (32, 28, 28) | (5, 5) | (1, 1) | (0, 0) | (1, 1) | 1 | Yes | General |
| conv2.1 (1) | (32, 18, 18) | (64, 14, 14) | (5, 5) | (1, 1) | (0, 0) | (1, 1) | 1 | Yes | General |

(c) All-CNN-C, CIFAR-100 (3, 32, 32)

| Name (count) | Input shape | Output shape | Kernel | Stride | Padding | Dilation | Groups | Bias | Type |
|---|---|---|---|---|---|---|---|---|---|
| conv1.1 (1) | (3, 34, 34) | (96, 32, 32) | (3, 3) | (1, 1) | (0, 0) | (1, 1) | 1 | Yes | General |
| conv2.1 (1) | (96, 34, 34) | (96, 32, 32) | (3, 3) | (1, 1) | (0, 0) | (1, 1) | 1 | Yes | General |
| conv3.1 (1) | (96, 33, 33) | (96, 16, 16) | (3, 3) | (2, 2) | (0, 0) | (1, 1) | 1 | Yes | General |
| conv4.1 (1) | (96, 18, 18) | (192, 16, 16) | (3, 3) | (1, 1) | (0, 0) | (1, 1) | 1 | Yes | General |
| conv5.1 (1) | (192, 18, 18) | (192, 16, 16) | (3, 3) | (1, 1) | (0, 0) | (1, 1) | 1 | Yes | General |
| conv6.1 (1) | (192, 17, 17) | (192, 8, 8) | (3, 3) | (2, 2) | (0, 0) | (1, 1) | 1 | Yes | General |
| conv7.0 (1) | (192, 8, 8) | (192, 6, 6) | (3, 3) | (1, 1) | (0, 0) | (1, 1) | 1 | Yes | General |
| conv8.1 (1) | (192, 6, 6) | (192, 6, 6) | (1, 1) | (1, 1) | (0, 0) | (1, 1) | 1 | Yes | Dense |
| conv9.1 (1) | (192, 6, 6) | (100, 6, 6) | (1, 1) | (1, 1) | (0, 0) | (1, 1) | 1 | Yes | Dense |

(d) AlexNet, ImageNet (3, 256, 256)

| Name (count) | Input shape | Output shape | Kernel | Stride | Padding | Dilation | Groups | Bias | Type |
|---|---|---|---|---|---|---|---|---|---|
| features.0 (1) | (3, 256, 256) | (64, 63, 63) | (11, 11) | (4, 4) | (2, 2) | (1, 1) | 1 | Yes | General |
| features.3 (1) | (64, 31, 31) | (192, 31, 31) | (5, 5) | (1, 1) | (2, 2) | (1, 1) | 1 | Yes | General |
| features.6 (1) | (192, 15, 15) | (384, 15, 15) | (3, 3) | (1, 1) | (1, 1) | (1, 1) | 1 | Yes | General |
| features.8 (1) | (384, 15, 15) | (256, 15, 15) | (3, 3) | (1, 1) | (1, 1) | (1, 1) | 1 | Yes | General |
| features.10 (1) | (256, 15, 15) | (256, 15, 15) | (3, 3) | (1, 1) | (1, 1) | (1, 1) | 1 | Yes | General |

(e) ResNet18, ImageNet (3, 256, 256)

| Name (count) | Input shape | Output shape | Kernel | Stride | Padding | Dilation | Groups | Bias | Type |
|---|---|---|---|---|---|---|---|---|---|
| conv1 (1) | (3, 256, 256) | (64, 128, 128) | (7, 7) | (2, 2) | (3, 3) | (1, 1) | 1 | No | General |
| layer1.0.conv1 (4) | (64, 64, 64) | (64, 64, 64) | (3, 3) | (1, 1) | (1, 1) | (1, 1) | 1 | No | General |
| layer2.0.conv1 (1) | (64, 64, 64) | (128, 32, 32) | (3, 3) | (2, 2) | (1, 1) | (1, 1) | 1 | No | General |
| layer2.0.conv2 (3) | (128, 32, 32) | (128, 32, 32) | (3, 3) | (1, 1) | (1, 1) | (1, 1) | 1 | No | General |
| layer2.0.downsample.0 (1) | (64, 64, 64) | (128, 32, 32) | (1, 1) | (2, 2) | (0, 0) | (1, 1) | 1 | No | Down |
| layer3.0.conv1 (1) | (128, 32, 32) | (256, 16, 16) | (3, 3) | (2, 2) | (1, 1) | (1, 1) | 1 | No | General |
| layer3.0.conv2 (3) | (256, 16, 16) | (256, 16, 16) | (3, 3) | (1, 1) | (1, 1) | (1, 1) | 1 | No | General |
| layer3.0.downsample.0 (1) | (128, 32, 32) | (256, 16, 16) | (1, 1) | (2, 2) | (0, 0) | (1, 1) | 1 | No | Down |
| layer4.0.conv1 (1) | (256, 16, 16) | (512, 8, 8) | (3, 3) | (2, 2) | (1, 1) | (1, 1) | 1 | No | General |
| layer4.0.conv2 (3) | (512, 8, 8) | (512, 8, 8) | (3, 3) | (1, 1) | (1, 1) | (1, 1) | 1 | No | General |
| layer4.0.downsample.0 (1) | (256, 16, 16) | (512, 8, 8) | (1, 1) | (2, 2) | (0, 0) | (1, 1) | 1 | No | Down |

(f) ResNext101_32x8d, ImageNet (3, 256, 256)

| Name (count) | Input shape | Output shape | Kernel | Stride | Padding | Dilation | Groups | Bias | Type |
|---|---|---|---|---|---|---|---|---|---|
| conv1 (1) | (3, 256, 256) | (64, 128, 128) | (7, 7) | (2, 2) | (3, 3) | (1, 1) | 1 | No | General |
| layer1.0.conv1 (2) | (64, 64, 64) | (256, 64, 64) | (1, 1) | (1, 1) | (0, 0) | (1, 1) | 1 | No | Dense |
| layer1.0.conv2 (3) | (256, 64, 64) | (256, 64, 64) | (3, 3) | (1, 1) | (1, 1) | (1, 1) | 32 | No | General |
| layer1.0.conv3 (5) | (256, 64, 64) | (256, 64, 64) | (1, 1) | (1, 1) | (0, 0) | (1, 1) | 1 | No | Dense |
| layer2.0.conv1 (1) | (256, 64, 64) | (512, 64, 64) | (1, 1) | (1, 1) | (0, 0) | (1, 1) | 1 | No | Dense |
| layer2.0.conv2 (1) | (512, 64, 64) | (512, 32, 32) | (3, 3) | (2, 2) | (1, 1) | (1, 1) | 32 | No | General |
| layer2.0.conv3 (7) | (512, 32, 32) | (512, 32, 32) | (1, 1) | (1, 1) | (0, 0) | (1, 1) | 1 | No | Dense |
| layer2.0.downsample.0 (1) | (256, 64, 64) | (512, 32, 32) | (1, 1) | (2, 2) | (0, 0) | (1, 1) | 1 | No | Down |
| layer2.1.conv2 (3) | (512, 32, 32) | (512, 32, 32) | (3, 3) | (1, 1) | (1, 1) | (1, 1) | 32 | No | General |
| layer3.0.conv1 (1) | (512, 32, 32) | (1024, 32, 32) | (1, 1) | (1, 1) | (0, 0) | (1, 1) | 1 | No | Dense |
| layer3.0.conv2 (1) | (1024, 32, 32) | (1024, 16, 16) | (3, 3) | (2, 2) | (1, 1) | (1, 1) | 32 | No | General |
| layer3.0.conv3 (45) | (1024, 16, 16) | (1024, 16, 16) | (1, 1) | (1, 1) | (0, 0) | (1, 1) | 1 | No | Dense |
| layer3.0.downsample.0 (1) | (512, 32, 32) | (1024, 16, 16) | (1, 1) | (2, 2) | (0, 0) | (1, 1) | 1 | No | Down |
| layer3.1.conv2 (22) | (1024, 16, 16) | (1024, 16, 16) | (3, 3) | (1, 1) | (1, 1) | (1, 1) | 32 | No | General |
| layer4.0.conv1 (1) | (1024, 16, 16) | (2048, 16, 16) | (1, 1) | (1, 1) | (0, 0) | (1, 1) | 1 | No | Dense |
| layer4.0.conv2 (1) | (2048, 16, 16) | (2048, 8, 8) | (3, 3) | (2, 2) | (1, 1) | (1, 1) | 32 | No | General |
| layer4.0.conv3 (5) | (2048, 8, 8) | (2048, 8, 8) | (1, 1) | (1, 1) | (0, 0) | (1, 1) | 1 | No | Dense |
| layer4.0.downsample.0 (1) | (1024, 16, 16) | (2048, 8, 8) | (1, 1) | (2, 2) | (0, 0) | (1, 1) | 1 | No | Down |
| layer4.1.conv2 (2) | (2048, 8, 8) | (2048, 8, 8) | (3, 3) | (1, 1) | (1, 1) | (1, 1) | 32 | No | General |

(g) ConvNeXt-base, ImageNet (3, 256, 256)

| Name (count) | Input shape | Output shape | Kernel | Stride | Padding | Dilation | Groups | Bias | Type |
|---|---|---|---|---|---|---|---|---|---|
| features.0.0 (1) | (3, 256, 256) | (128, 64, 64) | (4, 4) | (4, 4) | (0, 0) | (1, 1) | 1 | Yes | Dense |
| features.1.0.block.0 (3) | (128, 64, 64) | (128, 64, 64) | (7, 7) | (1, 1) | (3, 3) | (1, 1) | 128 | Yes | General |
| features.2.1 (1) | (128, 64, 64) | (256, 32, 32) | (2, 2) | (2, 2) | (0, 0) | (1, 1) | 1 | Yes | Dense |
| features.3.0.block.0 (3) | (256, 32, 32) | (256, 32, 32) | (7, 7) | (1, 1) | (3, 3) | (1, 1) | 256 | Yes | General |
| features.4.1 (1) | (256, 32, 32) | (512, 16, 16) | (2, 2) | (2, 2) | (0, 0) | (1, 1) | 1 | Yes | Dense |
| features.5.0.block.0 (27) | (512, 16, 16) | (512, 16, 16) | (7, 7) | (1, 1) | (3, 3) | (1, 1) | 512 | Yes | General |
| features.6.1 (1) | (512, 16, 16) | (1024, 8, 8) | (2, 2) | (2, 2) | (0, 0) | (1, 1) | 1 | Yes | Dense |
| features.7.0.block.0 (3) | (1024, 8, 8) | (1024, 8, 8) | (7, 7) | (1, 1) | (3, 3) | (1, 1) | 1024 | Yes | General |

(h) InceptionV3, ImageNet (3, 299, 299)

| Name (count) | Input shape | Output shape | Kernel | Stride | Padding | Dilation | Groups | Bias | Type |
|---|---|---|---|---|---|---|---|---|---|
| Conv2d_1a_3x3.conv (1) | (3, 299, 299) | (32, 149, 149) | (3, 3) | (2, 2) | (0, 0) | (1, 1) | 1 | No | General |
| Conv2d_2a_3x3.conv (1) | (32, 149, 149) | (32, 147, 147) | (3, 3) | (1, 1) | (0, 0) | (1, 1) | 1 | No | General |
| Conv2d_2b_3x3.conv (1) | (32, 147, 147) | (64, 147, 147) | (3, 3) | (1, 1) | (1, 1) | (1, 1) | 1 | No | General |
| Conv2d_3b_1x1.conv (1) | (64, 73, 73) | (80, 73, 73) | (1, 1) | (1, 1) | (0, 0) | (1, 1) | 1 | No | Dense |
| Conv2d_4a_3x3.conv (1) | (80, 73, 73) | (192, 71, 71) | (3, 3) | (1, 1) | (0, 0) | (1, 1) | 1 | No | General |
| Mixed_5b.branch1x1.conv (2) | (192, 35, 35) | (64, 35, 35) | (1, 1) | (1, 1) | (0, 0) | (1, 1) | 1 | No | Dense |
| Mixed_5b.branch5x5_1.conv (1) | (192, 35, 35) | (48, 35, 35) | (1, 1) | (1, 1) | (0, 0) | (1, 1) | 1 | No | Dense |
| Mixed_5b.branch5x5_2.conv (3) | (48, 35, 35) | (64, 35, 35) | (5, 5) | (1, 1) | (2, 2) | (1, 1) | 1 | No | General |
| Mixed_5b.branch3x3dbl_2.conv (4) | (64, 35, 35) | (96, 35, 35) | (3, 3) | (1, 1) | (1, 1) | (1, 1) | 1 | No | General |
| Mixed_5b.branch3x3dbl_3.conv (3) | (96, 35, 35) | (96, 35, 35) | (3, 3) | (1, 1) | (1, 1) | (1, 1) | 1 | No | General |
| Mixed_5b.branch_pool.conv (1) | (192, 35, 35) | (32, 35, 35) | (1, 1) | (1, 1) | (0, 0) | (1, 1) | 1 | No | Dense |
| Mixed_5c.branch1x1.conv (3) | (256, 35, 35) | (64, 35, 35) | (1, 1) | (1, 1) | (0, 0) | (1, 1) | 1 | No | Dense |
| Mixed_5c.branch5x5_1.conv (1) | (256, 35, 35) | (48, 35, 35) | (1, 1) | (1, 1) | (0, 0) | (1, 1) | 1 | No | Dense |
| Mixed_5d.branch1x1.conv (4) | (288, 35, 35) | (64, 35, 35) | (1, 1) | (1, 1) | (0, 0) | (1, 1) | 1 | No | Dense |
| Mixed_5d.branch5x5_1.conv (1) | (288, 35, 35) | (48, 35, 35) | (1, 1) | (1, 1) | (0, 0) | (1, 1) | 1 | No | Dense |
| Mixed_6a.branch3x3.conv (1) | (288, 35, 35) | (384, 17, 17) | (3, 3) | (2, 2) | (0, 0) | (1, 1) | 1 | No | General |
| Mixed_6a.branch3x3dbl_3.conv (1) | (96, 35, 35) | (96, 17, 17) | (3, 3) | (2, 2) | (0, 0) | (1, 1) | 1 | No | General |
| Mixed_6b.branch1x1.conv (12) | (768, 17, 17) | (192, 17, 17) | (1, 1) | (1, 1) | (0, 0) | (1, 1) | 1 | No | Dense |
| Mixed_6b.branch7x7_1.conv (2) | (768, 17, 17) | (128, 17, 17) | (1, 1) | (1, 1) | (0, 0) | (1, 1) | 1 | No | Dense |
| Mixed_6b.branch7x7_2.conv (2) | (128, 17, 17) | (128, 17, 17) | (1, 7) | (1, 1) | (0, 3) | (1, 1) | 1 | No | Dense mix |
| Mixed_6b.branch7x7_3.conv (1) | (128, 17, 17) | (192, 17, 17) | (7, 1) | (1, 1) | (3, 0) | (1, 1) | 1 | No | Dense mix |
| Mixed_6b.branch7x7dbl_2.conv (2) | (128, 17, 17) | (128, 17, 17) | (7, 1) | (1, 1) | (3, 0) | (1, 1) | 1 | No | Dense mix |
| Mixed_6b.branch7x7dbl_5.conv (1) | (128, 17, 17) | (192, 17, 17) | (1, 7) | (1, 1) | (0, 3) | (1, 1) | 1 | No | Dense mix |
| Mixed_6c.branch7x7_1.conv (4) | (768, 17, 17) | (160, 17, 17) | (1, 1) | (1, 1) | (0, 0) | (1, 1) | 1 | No | Dense |
| Mixed_6c.branch7x7_2.conv (4) | (160, 17, 17) | (160, 17, 17) | (1, 7) | (1, 1) | (0, 3) | (1, 1) | 1 | No | Dense mix |
| Mixed_6c.branch7x7_3.conv (2) | (160, 17, 17) | (192, 17, 17) | (7, 1) | (1, 1) | (3, 0) | (1, 1) | 1 | No | Dense mix |
| Mixed_6c.branch7x7dbl_2.conv (4) | (160, 17, 17) | (160, 17, 17) | (7, 1) | (1, 1) | (3, 0) | (1, 1) | 1 | No | Dense mix |
| Mixed_6c.branch7x7dbl_5.conv (2) | (160, 17, 17) | (192, 17, 17) | (1, 7) | (1, 1) | (0, 3) | (1, 1) | 1 | No | Dense mix |
| Mixed_6e.branch7x7_2.conv (4) | (192, 17, 17) | (192, 17, 17) | (1, 7) | (1, 1) | (0, 3) | (1, 1) | 1 | No | Dense mix |
| Mixed_6e.branch7x7_3.conv (4) | (192, 17, 17) | (192, 17, 17) | (7, 1) | (1, 1) | (3, 0) | (1, 1) | 1 | No | Dense mix |
| AuxLogits.conv0.conv (1) | (768, 5, 5) | (128, 5, 5) | (1, 1) | (1, 1) | (0, 0) | (1, 1) | 1 | No | Dense |
| AuxLogits.conv1.conv (1) | (128, 5, 5) | (768, 1, 1) | (5, 5) | (1, 1) | (0, 0) | (1, 1) | 1 | No | General |
| Mixed_7a.branch3x3_2.conv (1) | (192, 17, 17) | (320, 8, 8) | (3, 3) | (2, 2) | (0, 0) | (1, 1) | 1 | No | General |
| Mixed_7a.branch7x7x3_4.conv (1) | (192, 17, 17) | (192, 8, 8) | (3, 3) | (2, 2) | (0, 0) | (1, 1) | 1 | No | General |
| Mixed_7b.branch1x1.conv (1) | (1280, 8, 8) | (320, 8, 8) | (1, 1) | (1, 1) | (0, 0) | (1, 1) | 1 | No | Dense |
| Mixed_7b.branch3x3_1.conv (1) | (1280, 8, 8) | (384, 8, 8) | (1, 1) | (1, 1) | (0, 0) | (1, 1) | 1 | No | Dense |
| Mixed_7b.branch3x3_2a.conv (4) | (384, 8, 8) | (384, 8, 8) | (1, 3) | (1, 1) | (0, 1) | (1, 1) | 1 | No | Dense mix |
| Mixed_7b.branch3x3_2b.conv (4) | (384, 8, 8) | (384, 8, 8) | (3, 1) | (1, 1) | (1, 0) | (1, 1) | 1 | No | Dense mix |
| Mixed_7b.branch3x3dbl_1.conv (1) | (1280, 8, 8) | (448, 8, 8) | (1, 1) | (1, 1) | (0, 0) | (1, 1) | 1 | No | Dense |
| Mixed_7b.branch3x3dbl_2.conv (2) | (448, 8, 8) | (384, 8, 8) | (3, 3) | (1, 1) | (1, 1) | (1, 1) | 1 | No | General |
| Mixed_7b.branch_pool.conv (1) | (1280, 8, 8) | (192, 8, 8) | (1, 1) | (1, 1) | (0, 0) | (1, 1) | 1 | No | Dense |
| Mixed_7c.branch1x1.conv (1) | (2048, 8, 8) | (320, 8, 8) | (1, 1) | (1, 1) | (0, 0) | (1, 1) | 1 | No | Dense |
| Mixed_7c.branch3x3_1.conv (1) | (2048, 8, 8) | (384, 8, 8) | (1, 1) | (1, 1) | (0, 0) | (1, 1) | 1 | No | Dense |
| Mixed_7c.branch3x3dbl_1.conv (1) | (2048, 8, 8) | (448, 8, 8) | (1, 1) | (1, 1) | (0, 0) | (1, 1) | 1 | No | Dense |
| Mixed_7c.branch_pool.conv (1) | (2048, 8, 8) | (192, 8, 8) | (1, 1) | (1, 1) | (0, 0) | (1, 1) | 1 | No | Dense |

(i) MobileNetV2, ImageNet (3, 256, 256)

| Name (count) | Input shape | Output shape | Kernel | Stride | Padding | Dilation | Groups | Bias | Type |
|---|---|---|---|---|---|---|---|---|---|
| features.0.0 (1) | (3, 256, 256) | (32, 128, 128) | (3, 3) | (2, 2) | (1, 1) | (1, 1) | 1 | No | General |
| features.1.conv.0.0 (1) | (32, 128, 128) | (32, 128, 128) | (3, 3) | (1, 1) | (1, 1) | (1, 1) | 32 | No | General |
| features.1.conv.1 (1) | (32, 128, 128) | (16, 128, 128) | (1, 1) | (1, 1) | (0, 0) | (1, 1) | 1 | No | Dense |
| features.2.conv.0.0 (1) | (16, 128, 128) | (96, 128, 128) | (1, 1) | (1, 1) | (0, 0) | (1, 1) | 1 | No | Dense |
| features.2.conv.1.0 (1) | (96, 128, 128) | (96, 64, 64) | (3, 3) | (2, 2) | (1, 1) | (1, 1) | 96 | No | General |
| features.2.conv.2 (1) | (96, 64, 64) | (24, 64, 64) | (1, 1) | (1, 1) | (0, 0) | (1, 1) | 1 | No | Dense |
| features.3.conv.0.0 (2) | (24, 64, 64) | (144, 64, 64) | (1, 1) | (1, 1) | (0, 0) | (1, 1) | 1 | No | Dense |
| features.3.conv.1.0 (1) | (144, 64, 64) | (144, 64, 64) | (3, 3) | (1, 1) | (1, 1) | (1, 1) | 144 | No | General |
| features.3.conv.2 (1) | (144, 64, 64) | (24, 64, 64) | (1, 1) | (1, 1) | (0, 0) | (1, 1) | 1 | No | Dense |
| features.4.conv.1.0 (1) | (144, 64, 64) | (144, 32, 32) | (3, 3) | (2, 2) | (1, 1) | (1, 1) | 144 | No | General |
| features.4.conv.2 (1) | (144, 32, 32) | (32, 32, 32) | (1, 1) | (1, 1) | (0, 0) | (1, 1) | 1 | No | Dense |
| features.5.conv.0.0 (3) | (32, 32, 32) | (192, 32, 32) | (1, 1) | (1, 1) | (0, 0) | (1, 1) | 1 | No | Dense |
| features.5.conv.1.0 (2) | (192, 32, 32) | (192, 32, 32) | (3, 3) | (1, 1) | (1, 1) | (1, 1) | 192 | No | General |
| features.5.conv.2 (2) | (192, 32, 32) | (32, 32, 32) | (1, 1) | (1, 1) | (0, 0) | (1, 1) | 1 | No | Dense |
| features.7.conv.1.0 (1) | (192, 32, 32) | (192, 16, 16) | (3, 3) | (2, 2) | (1, 1) | (1, 1) | 192 | No | General |
| features.7.conv.2 (1) | (192, 16, 16) | (64, 16, 16) | (1, 1) | (1, 1) | (0, 0) | (1, 1) | 1 | No | Dense |
| features.8.conv.0.0 (4) | (64, 16, 16) | (384, 16, 16) | (1, 1) | (1, 1) | (0, 0) | (1, 1) | 1 | No | Dense |
| features.8.conv.1.0 (4) | (384, 16, 16) | (384, 16, 16) | (3, 3) | (1, 1) | (1, 1) | (1, 1) | 384 | No | General |
| features.8.conv.2 (3) | (384, 16, 16) | (64, 16, 16) | (1, 1) | (1, 1) | (0, 0) | (1, 1) | 1 | No | Dense |
| features.11.conv.2 (1) | (384, 16, 16) | (96, 16, 16) | (1, 1) | (1, 1) | (0, 0) | (1, 1) | 1 | No | Dense |
| features.12.conv.0.0 (3) | (96, 16, 16) | (576, 16, 16) | (1, 1) | (1, 1) | (0, 0) | (1, 1) | 1 | No | Dense |
| features.12.conv.1.0 (2) | (576, 16, 16) | (576, 16, 16) | (3, 3) | (1, 1) | (1, 1) | (1, 1) | 576 | No | General |
| features.12.conv.2 (2) | (576, 16, 16) | (96, 16, 16) | (1, 1) | (1, 1) | (0, 0) | (1, 1) | 1 | No | Dense |
| features.14.conv.1.0 (1) | (576, 16, 16) | (576, 8, 8) | (3, 3) | (2, 2) | (1, 1) | (1, 1) | 576 | No | General |
| features.14.conv.2 (1) | (576, 8, 8) | (160, 8, 8) | (1, 1) | (1, 1) | (0, 0) | (1, 1) | 1 | No | Dense |
| features.15.conv.0.0 (3) | (160, 8, 8) | (960, 8, 8) | (1, 1) | (1, 1) | (0, 0) | (1, 1) | 1 | No | Dense |
| features.15.conv.1.0 (3) | (960, 8, 8) | (960, 8, 8) | (3, 3) | (1, 1) | (1, 1) | (1, 1) | 960 | No | General |
| features.15.conv.2 (2) | (960, 8, 8) | (160, 8, 8) | (1, 1) | (1, 1) | (0, 0) | (1, 1) | 1 | No | Dense |
| features.17.conv.2 (1) | (960, 8, 8) | (320, 8, 8) | (1, 1) | (1, 1) | (0, 0) | (1, 1) | 1 | No | Dense |
| features.18.0 (1) | (320, 8, 8) | (1280, 8, 8) | (1, 1) | (1, 1) | (0, 0) | (1, 1) | 1 | No | Dense |

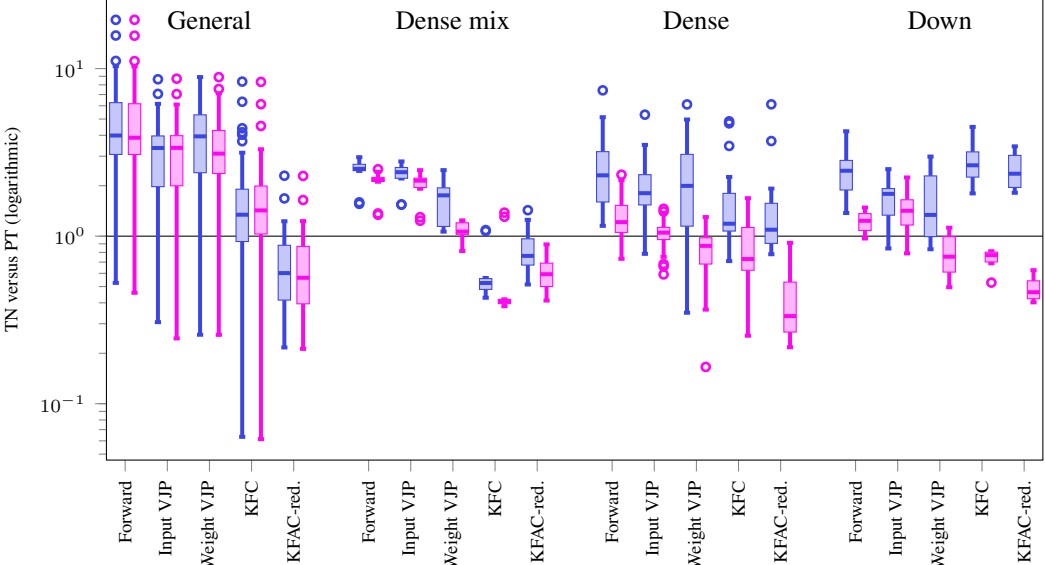

Figure F16: Benchmark overview. We measure the performance ratios of our TN implementation w.r.t. a base line in PyTorch (PT). Blue boxes show the performance ratios of TN versus PT, second-color boxes show the performance ratios of TN+opt versus PT.

## F  RUN TIME EVALUATION DETAILS (GPU)

Here we provide all details on the run time evaluation from the main text. We consider the convolutions from the CNNs from §E. Experiments were carried out on an Nvidia Tesla T4 (16 GB memory). We use a batch size of 32 for the ImageNet architectures, and 128 for the others.

### F.1  PROTOCOL & OVERVIEW

We compare different implementations of the same operations in PyTorch. The base line (referenced by 'PT') uses PyTorch's built-in functionalities for convolutions and related operations, such as `torch.nn.functional.conv2d` (forward), `torch.nn.functional.unfold` (KFC, KFAC-reduce), and PyTorch's built-in automatic differentiation `torch.autograd.grad` (VJPs).

Our TN implementation (referenced by 'TN') sets up operands and the string-valued equation for each routine. Optionally, we can apply the simplifications from §4 as a post-processing step before contraction, which yields a modified equation and operand list ('TN + opt'). Finally, we determine the contraction path using `opt_einsum.contract_path` and perform the contraction with its PyTorch back-end (`opt_einsum.contract`). We only measure the contraction time as in practical settings, the contraction path search would be executed once, then cached. We also exclude final operations to obtain the correct shape or scale (flattening, reshaping, scaling by constant) in all implementations (including the base line).

For each operation and each convolution layer, we perform 50 independent repetitions and report the minimum time in tables. To summarize those tables, we extract the performance ratios, that is the TN implementation's run time divided by the base line's. Ratios larger than 1 mean that the TN implementation is slower, ratios smaller than 1 indicate that it is faster than the base line. We collect those ratios for the different convolution types (general, mixed dense, dense, sub-sampling) and display them separately using box plots. Each operation has two boxes, corresponding to the un-simplified (TN), and the simplified (TN + opt) implementation. For the box plots, we use `matplotlib`'s default settings (a box extends from the first quartile to the third quartile of the data, with a line at the median; whiskers extend from the box by 1.5x the inter-quartile range; flier points are those past the end of the whiskers). Figure F16 summarizes the entire GPU benchmark. Figure F17 shows the same information with each convolution type as an individual plot.

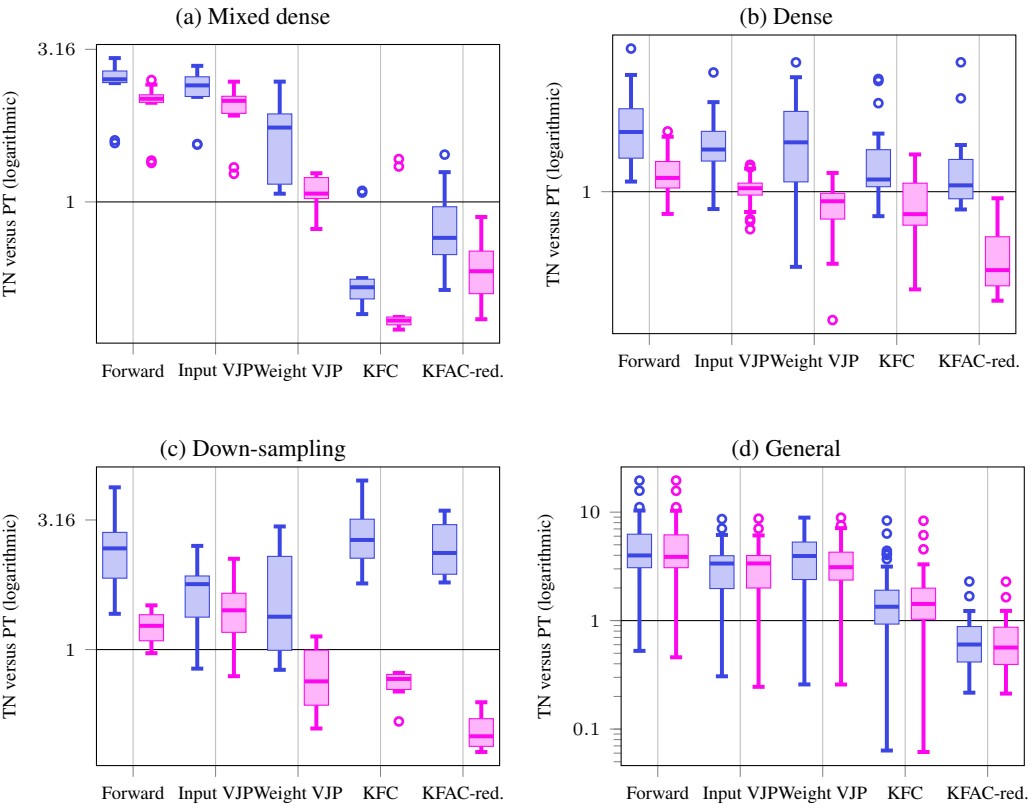

Figure F17: Impact of TN simplifications (non-simplified performance ratios shown in blue). TN simplifications improve performance on (a) mixed dense, (b) dense, and (c) down-sampling convolutions. (d) General convolutions are not affected by TN simplifications.

## F.2 FORWARD PASS

We compare TN and TN+opt with PyTorch's `torch.nn.functional.conv2d`. Figure F18 visualizes the performance ratios for different convolution categories. Table F4 contains the detailed run times and performance factors.

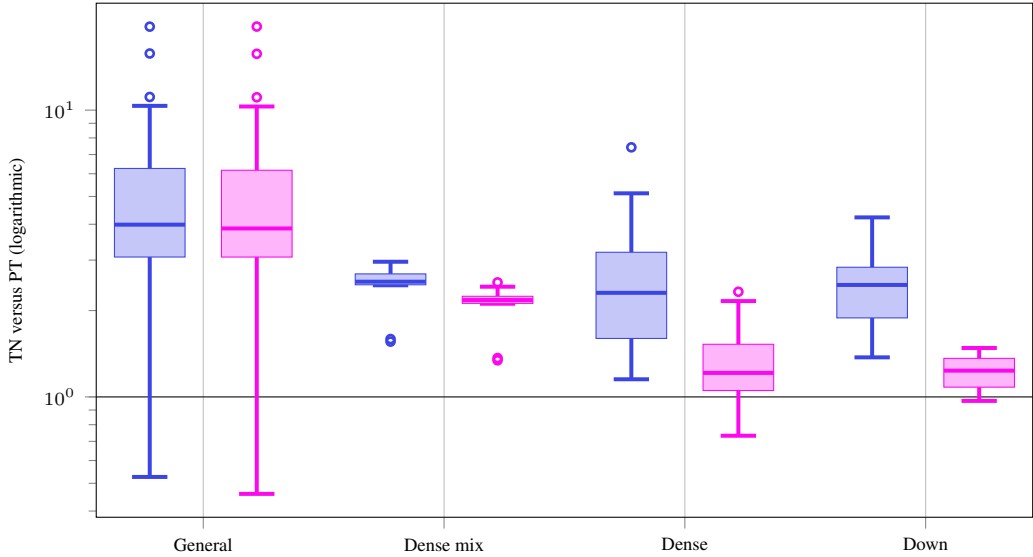

Figure F18: Forward pass performance ratios of TN versus PT and TN+opt versus PT for different convolution types on GPU.

Table F4: Forward pass performance comparison on GPU.

(a) 3c3d, CIFAR-10, input shape (128, 3, 32, 32)

| Name | TN [s] | PT [s] | Factor | TN + opt [s] | PT [s] | Factor | Type |
|---|---|---|---|---|---|---|---|
| conv1.0 | $1.26 \cdot 10^{-3}$ | $4.63 \cdot 10^{-4}$ | 2.73 x | $1.23 \cdot 10^{-3}$ | $4.64 \cdot 10^{-4}$ | 2.64 x | General |
| conv2.0 | $1.91 \cdot 10^{-3}$ | $4.52 \cdot 10^{-4}$ | 4.22 x | $1.79 \cdot 10^{-3}$ | $4.53 \cdot 10^{-4}$ | 3.95 x | General |
| conv3.1 | $1.21 \cdot 10^{-3}$ | $4.11 \cdot 10^{-4}$ | 2.94 x | $1.16 \cdot 10^{-3}$ | $4.10 \cdot 10^{-4}$ | 2.83 x | General |

(b) F-MNIST 2c2d, input shape (128, 1, 28, 28)

| Name | TN [s] | PT [s] | Factor | TN + opt [s] | PT [s] | Factor | Type |
|---|---|---|---|---|---|---|---|
| conv1.1 | $8.21 \cdot 10^{-4}$ | $2.25 \cdot 10^{-4}$ | 3.65 x | $7.67 \cdot 10^{-4}$ | $2.25 \cdot 10^{-4}$ | 3.41 x | General |
| conv2.1 | $3.56 \cdot 10^{-3}$ | $7.43 \cdot 10^{-4}$ | 4.79 x | $3.24 \cdot 10^{-3}$ | $7.83 \cdot 10^{-4}$ | 4.14 x | General |

(c) CIFAR-100 All-CNN-C, input shape (128, 3, 32, 32)

| Name | TN [s] | PT [s] | Factor | TN + opt [s] | PT [s] | Factor | Type |
|---|---|---|---|---|---|---|---|
| conv1.1 | $1.01 \cdot 10^{-3}$ | $4.20 \cdot 10^{-4}$ | 2.41 x | $9.45 \cdot 10^{-4}$ | $4.19 \cdot 10^{-4}$ | 2.25 x | General |
| conv2.1 | $1.94 \cdot 10^{-2}$ | $3.09 \cdot 10^{-3}$ | 6.26 x | $1.88 \cdot 10^{-2}$ | $3.10 \cdot 10^{-3}$ | 6.08 x | General |
| conv3.1 | $8.56 \cdot 10^{-3}$ | $2.86 \cdot 10^{-3}$ | 3.00 x | $7.77 \cdot 10^{-3}$ | $2.86 \cdot 10^{-3}$ | 2.72 x | General |
| conv4.1 | $8.58 \cdot 10^{-3}$ | $1.75 \cdot 10^{-3}$ | 4.91 x | $7.77 \cdot 10^{-3}$ | $1.75 \cdot 10^{-3}$ | 4.45 x | General |
| conv5.1 | $1.67 \cdot 10^{-2}$ | $2.91 \cdot 10^{-3}$ | 5.74 x | $1.51 \cdot 10^{-2}$ | $2.91 \cdot 10^{-3}$ | 5.19 x | General |
| conv6.1 | $5.13 \cdot 10^{-3}$ | $2.24 \cdot 10^{-3}$ | 2.29 x | $5.08 \cdot 10^{-3}$ | $2.24 \cdot 10^{-3}$ | 2.27 x | General |
| conv7.0 | $2.58 \cdot 10^{-3}$ | $8.26 \cdot 10^{-4}$ | 3.12 x | $2.51 \cdot 10^{-3}$ | $8.27 \cdot 10^{-4}$ | 3.03 x | General |
| conv8.1 | $8.20 \cdot 10^{-4}$ | $2.96 \cdot 10^{-4}$ | 2.77 x | $3.42 \cdot 10^{-4}$ | $2.97 \cdot 10^{-4}$ | 1.15 x | Dense |
| conv9.1 | $7.52 \cdot 10^{-4}$ | $2.35 \cdot 10^{-4}$ | 3.19 x | $3.01 \cdot 10^{-4}$ | $2.35 \cdot 10^{-4}$ | 1.28 x | Dense |

(d) Alexnet, input shape (32, 3, 256, 256)

| Name | TN [s] | PT [s] | Factor | TN + opt [s] | PT [s] | Factor | Type |
|---|---|---|---|---|---|---|---|
| features.0 | $1.83 \cdot 10^{-2}$ | $2.45 \cdot 10^{-3}$ | 7.47 x | $1.79 \cdot 10^{-2}$ | $2.44 \cdot 10^{-3}$ | 7.35 x | General |
| features.3 | $7.43 \cdot 10^{-3}$ | $2.67 \cdot 10^{-3}$ | 2.79 x | $7.27 \cdot 10^{-3}$ | $2.85 \cdot 10^{-3}$ | 2.55 x | General |
| features.6 | $4.68 \cdot 10^{-3}$ | $1.04 \cdot 10^{-3}$ | 4.52 x | $3.22 \cdot 10^{-3}$ | $1.02 \cdot 10^{-3}$ | 3.14 x | General |
| features.8 | $6.15 \cdot 10^{-3}$ | $1.86 \cdot 10^{-3}$ | 3.31 x | $6.16 \cdot 10^{-3}$ | $1.84 \cdot 10^{-3}$ | 3.34 x | General |
| features.10 | $4.41 \cdot 10^{-3}$ | $1.31 \cdot 10^{-3}$ | 3.36 x | $4.38 \cdot 10^{-3}$ | $1.31 \cdot 10^{-3}$ | 3.35 x | General |

(e) ResNet18, input shape (32, 3, 256, 256)

| Name | TN [s] | PT [s] | Factor | TN + opt [s] | PT [s] | Factor | Type |
|---|---|---|---|---|---|---|---|
| conv1 | $1.44 \cdot 10^{-2}$ | $4.07 \cdot 10^{-3}$ | 3.53 x | $1.44 \cdot 10^{-2}$ | $4.08 \cdot 10^{-3}$ | 3.53 x | General |
| layer1.0.conv1 | $1.05 \cdot 10^{-2}$ | $1.78 \cdot 10^{-3}$ | 5.91 x | $1.05 \cdot 10^{-2}$ | $1.79 \cdot 10^{-3}$ | 5.87 x | General |
| layer2.0.conv1 | $6.44 \cdot 10^{-3}$ | $1.89 \cdot 10^{-3}$ | 3.41 x | $6.46 \cdot 10^{-3}$ | $1.89 \cdot 10^{-3}$ | 3.42 x | General |
| layer2.0.conv2 | $6.88 \cdot 10^{-3}$ | $1.51 \cdot 10^{-3}$ | 4.54 x | $6.91 \cdot 10^{-3}$ | $1.52 \cdot 10^{-3}$ | 4.54 x | General |
| layer2.0.downsample.0 | $1.60 \cdot 10^{-3}$ | $3.79 \cdot 10^{-4}$ | 4.23 x | $5.19 \cdot 10^{-4}$ | $3.80 \cdot 10^{-4}$ | 1.37 x | Down |
| layer3.0.conv1 | $3.82 \cdot 10^{-3}$ | $2.00 \cdot 10^{-3}$ | 1.91 x | $3.56 \cdot 10^{-3}$ | $2.01 \cdot 10^{-3}$ | 1.77 x | General |
| layer3.0.conv2 | $5.02 \cdot 10^{-3}$ | $1.30 \cdot 10^{-3}$ | 3.85 x | $5.05 \cdot 10^{-3}$ | $1.31 \cdot 10^{-3}$ | 3.87 x | General |
| layer3.0.downsample.0 | $1.10 \cdot 10^{-3}$ | $3.78 \cdot 10^{-4}$ | 2.91 x | $5.61 \cdot 10^{-4}$ | $3.79 \cdot 10^{-4}$ | 1.48 x | Down |
| layer4.0.conv1 | $2.87 \cdot 10^{-3}$ | $2.36 \cdot 10^{-3}$ | 1.21 x | $2.86 \cdot 10^{-3}$ | $2.36 \cdot 10^{-3}$ | 1.21 x | General |
| layer4.0.conv2 | $4.47 \cdot 10^{-3}$ | $1.40 \cdot 10^{-3}$ | 3.18 x | $4.51 \cdot 10^{-3}$ | $1.40 \cdot 10^{-3}$ | 3.21 x | General |
| layer4.0.downsample.0 | $9.90 \cdot 10^{-4}$ | $3.81 \cdot 10^{-4}$ | 2.60 x | $5.16 \cdot 10^{-4}$ | $3.83 \cdot 10^{-4}$ | 1.35 x | Down |

(f) ResNext101, input shape (32, 3, 256, 256)

| Name | TN [s] | PT [s] | Factor | TN + opt [s] | PT [s] | Factor | Type |
|---|---|---|---|---|---|---|---|
| conv1 | $1.45 \cdot 10^{-2}$ | $4.07 \cdot 10^{-3}$ | 3.57 x | $1.44 \cdot 10^{-2}$ | $4.07 \cdot 10^{-3}$ | 3.54 x | General |
| layer1.0.conv1 | $4.31 \cdot 10^{-3}$ | $1.22 \cdot 10^{-3}$ | 3.54 x | $2.26 \cdot 10^{-3}$ | $1.22 \cdot 10^{-3}$ | 1.85 x | Dense |
| layer1.0.conv2 | $3.03 \cdot 10^{-2}$ | $9.86 \cdot 10^{-3}$ | 3.07 x | $3.03 \cdot 10^{-2}$ | $9.86 \cdot 10^{-3}$ | 3.08 x | General |
| layer1.0.conv3 | $1.51 \cdot 10^{-2}$ | $6.54 \cdot 10^{-3}$ | 2.31 x | $7.49 \cdot 10^{-3}$ | $6.54 \cdot 10^{-3}$ | 1.15 x | Dense |
| layer2.0.conv1 | $2.08 \cdot 10^{-2}$ | $1.29 \cdot 10^{-2}$ | 1.61 x | $1.36 \cdot 10^{-2}$ | $1.29 \cdot 10^{-2}$ | 1.05 x | Dense |
| layer2.0.conv2 | $3.33 \cdot 10^{-2}$ | $4.93 \cdot 10^{-3}$ | 6.75 x | $3.33 \cdot 10^{-2}$ | $4.93 \cdot 10^{-3}$ | 6.75 x | General |
| layer2.0.conv3 | $1.05 \cdot 10^{-2}$ | $6.24 \cdot 10^{-3}$ | 1.69 x | $6.84 \cdot 10^{-3}$ | $6.24 \cdot 10^{-3}$ | 1.10 x | Dense |
| layer2.0.downsample.0 | $7.65 \cdot 10^{-3}$ | $3.30 \cdot 10^{-3}$ | 2.31 x | $3.71 \cdot 10^{-3}$ | $3.31 \cdot 10^{-3}$ | 1.12 x | Down |
| layer2.1.conv2 | $1.50 \cdot 10^{-2}$ | $4.59 \cdot 10^{-3}$ | 3.27 x | $1.50 \cdot 10^{-2}$ | $4.59 \cdot 10^{-3}$ | 3.27 x | General |
| layer3.0.conv1 | $1.67 \cdot 10^{-2}$ | $1.23 \cdot 10^{-2}$ | 1.35 x | $1.28 \cdot 10^{-2}$ | $1.23 \cdot 10^{-2}$ | 1.04 x | Dense |
| layer3.0.conv2 | $1.76 \cdot 10^{-2}$ | $2.65 \cdot 10^{-3}$ | 6.65 x | $1.76 \cdot 10^{-2}$ | $2.66 \cdot 10^{-3}$ | 6.65 x | General |
| layer3.0.conv3 | $8.27 \cdot 10^{-3}$ | $6.14 \cdot 10^{-3}$ | 1.35 x | $6.44 \cdot 10^{-3}$ | $6.14 \cdot 10^{-3}$ | 1.05 x | Dense |
| layer3.0.downsample.0 | $5.58 \cdot 10^{-3}$ | $3.20 \cdot 10^{-3}$ | 1.74 x | $3.42 \cdot 10^{-3}$ | $3.20 \cdot 10^{-3}$ | 1.07 x | Down |
| layer3.1.conv2 | $7.64 \cdot 10^{-3}$ | $2.49 \cdot 10^{-3}$ | 3.07 x | $7.64 \cdot 10^{-3}$ | $2.48 \cdot 10^{-3}$ | 3.07 x | General |
| layer4.0.conv1 | $1.43 \cdot 10^{-2}$ | $1.22 \cdot 10^{-2}$ | 1.18 x | $1.24 \cdot 10^{-2}$ | $1.22 \cdot 10^{-2}$ | 1.02 x | Dense |
| layer4.0.conv2 | $8.07 \cdot 10^{-3}$ | $2.02 \cdot 10^{-3}$ | 3.99 x | $8.08 \cdot 10^{-3}$ | $2.02 \cdot 10^{-3}$ | 4.00 x | General |
| layer4.0.conv3 | $7.85 \cdot 10^{-3}$ | $6.28 \cdot 10^{-3}$ | 1.25 x | $6.33 \cdot 10^{-3}$ | $6.28 \cdot 10^{-3}$ | 1.01 x | Dense |
| layer4.0.downsample.0 | $4.73 \cdot 10^{-3}$ | $3.44 \cdot 10^{-3}$ | 1.37 x | $3.34 \cdot 10^{-3}$ | $3.44 \cdot 10^{-3}$ | **0.97 x** | Down |
| layer4.1.conv2 | $4.76 \cdot 10^{-3}$ | $1.36 \cdot 10^{-3}$ | 3.51 x | $4.77 \cdot 10^{-3}$ | $1.35 \cdot 10^{-3}$ | 3.52 x | General |

(g) ConvNeXt-base, input shape (32, 3, 256, 256)

| Name | TN [s] | PT [s] | Factor | TN + opt [s] | PT [s] | Factor | Type |
|---|---|---|---|---|---|---|---|
| features.0.0 | $4.26 \cdot 10^{-3}$ | $9.88 \cdot 10^{-4}$ | 4.31 x | $1.20 \cdot 10^{-3}$ | $9.94 \cdot 10^{-4}$ | 1.21 x | Dense |
| features.1.0.block.0 | $5.07 \cdot 10^{-2}$ | $7.61 \cdot 10^{-3}$ | 6.66 x | $5.07 \cdot 10^{-2}$ | $7.61 \cdot 10^{-3}$ | 6.66 x | General |
| features.2.1 | $7.60 \cdot 10^{-3}$ | $3.21 \cdot 10^{-3}$ | 2.37 x | $3.89 \cdot 10^{-3}$ | $3.20 \cdot 10^{-3}$ | 1.21 x | Dense |
| features.3.0.block.0 | $2.36 \cdot 10^{-2}$ | $3.81 \cdot 10^{-3}$ | 6.18 x | $2.35 \cdot 10^{-2}$ | $3.81 \cdot 10^{-3}$ | 6.17 x | General |
| features.4.1 | $5.41 \cdot 10^{-3}$ | $3.38 \cdot 10^{-3}$ | 1.60 x | $3.52 \cdot 10^{-3}$ | $3.38 \cdot 10^{-3}$ | 1.04 x | Dense |
| features.5.0.block.0 | $1.11 \cdot 10^{-2}$ | $1.94 \cdot 10^{-3}$ | 5.70 x | $1.10 \cdot 10^{-2}$ | $1.94 \cdot 10^{-3}$ | 5.69 x | General |
| features.6.1 | $4.54 \cdot 10^{-3}$ | $3.69 \cdot 10^{-3}$ | 1.23 x | $3.44 \cdot 10^{-3}$ | $3.70 \cdot 10^{-3}$ | **0.93 x** | Dense |
| features.7.0.block.0 | $1.06 \cdot 10^{-3}$ | $1.01 \cdot 10^{-3}$ | 1.05 x | $1.02 \cdot 10^{-3}$ | $1.01 \cdot 10^{-3}$ | 1.01 x | General |

(h) InceptionV3, input shape (32, 3, 299, 299)

| Name | TN [s] | PT [s] | Factor | TN + opt [s] | PT [s] | Factor | Type |
|---|---|---|---|---|---|---|---|
| Conv2d_1a_3x3.conv | $1.02 \cdot 10^{-2}$ | $9.85 \cdot 10^{-4}$ | 10.35 x | $1.01 \cdot 10^{-2}$ | $9.79 \cdot 10^{-4}$ | 10.30 x | General |
| Conv2d_2a_3x3.conv | $3.23 \cdot 10^{-2}$ | $5.14 \cdot 10^{-3}$ | 6.30 x | $3.19 \cdot 10^{-2}$ | $5.16 \cdot 10^{-3}$ | 6.18 x | General |
| Conv2d_2b_3x3.conv | $4.83 \cdot 10^{-2}$ | $8.14 \cdot 10^{-3}$ | 5.93 x | $4.78 \cdot 10^{-2}$ | $8.14 \cdot 10^{-3}$ | 5.87 x | General |
| Conv2d_3b_1x1.conv | $4.96 \cdot 10^{-3}$ | $1.17 \cdot 10^{-3}$ | 4.24 x | $1.72 \cdot 10^{-3}$ | $1.17 \cdot 10^{-3}$ | 1.48 x | Dense |
| Conv2d_4a_3x3.conv | $3.69 \cdot 10^{-2}$ | $7.64 \cdot 10^{-3}$ | 4.83 x | $3.65 \cdot 10^{-2}$ | $7.64 \cdot 10^{-3}$ | 4.77 x | General |
| Mixed_5b.branch1x1.conv | $1.85 \cdot 10^{-3}$ | $5.04 \cdot 10^{-4}$ | 3.68 x | $8.17 \cdot 10^{-4}$ | $5.03 \cdot 10^{-4}$ | 1.62 x | Dense |
| Mixed_5b.branch5x5_1.conv | $1.64 \cdot 10^{-3}$ | $4.97 \cdot 10^{-4}$ | 3.30 x | $8.11 \cdot 10^{-4}$ | $4.99 \cdot 10^{-4}$ | 1.63 x | Dense |
| Mixed_5b.branch5x5_2.conv | $5.01 \cdot 10^{-3}$ | $1.23 \cdot 10^{-3}$ | 4.07 x | $4.83 \cdot 10^{-3}$ | $1.23 \cdot 10^{-3}$ | 3.94 x | General |
| Mixed_5b.branch3x3dbl_2.conv | $4.40 \cdot 10^{-3}$ | $1.31 \cdot 10^{-3}$ | 3.38 x | $4.31 \cdot 10^{-3}$ | $1.31 \cdot 10^{-3}$ | 3.30 x | General |
| Mixed_5b.branch3x3dbl_3.conv | $5.82 \cdot 10^{-3}$ | $1.66 \cdot 10^{-3}$ | 3.50 x | $5.66 \cdot 10^{-3}$ | $1.66 \cdot 10^{-3}$ | 3.40 x | General |
| Mixed_5b.branch_pool.conv | $1.33 \cdot 10^{-3}$ | $3.26 \cdot 10^{-4}$ | 4.09 x | $7.04 \cdot 10^{-4}$ | $3.27 \cdot 10^{-4}$ | 2.15 x | Dense |
| Mixed_5c.branch1x1.conv | $2.08 \cdot 10^{-3}$ | $6.41 \cdot 10^{-4}$ | 3.24 x | $1.03 \cdot 10^{-3}$ | $6.40 \cdot 10^{-4}$ | 1.61 x | Dense |
| Mixed_5c.branch5x5_1.conv | $1.87 \cdot 10^{-3}$ | $6.29 \cdot 10^{-4}$ | 2.97 x | $1.03 \cdot 10^{-3}$ | $6.30 \cdot 10^{-4}$ | 1.63 x | Dense |
| Mixed_5d.branch1x1.conv | $2.18 \cdot 10^{-3}$ | $6.99 \cdot 10^{-4}$ | 3.12 x | $1.13 \cdot 10^{-3}$ | $6.98 \cdot 10^{-4}$ | 1.62 x | Dense |
| Mixed_5d.branch5x5_1.conv | $1.96 \cdot 10^{-3}$ | $6.91 \cdot 10^{-4}$ | 2.84 x | $1.13 \cdot 10^{-3}$ | $6.88 \cdot 10^{-4}$ | 1.64 x | Dense |
| Mixed_6a.branch3x3.conv | $1.15 \cdot 10^{-2}$ | $7.12 \cdot 10^{-3}$ | 1.61 x | $1.07 \cdot 10^{-2}$ | $7.13 \cdot 10^{-3}$ | 1.51 x | General |
| Mixed_6a.branch3x3dbl_3.conv | $2.61 \cdot 10^{-3}$ | $8.99 \cdot 10^{-4}$ | 2.90 x | $2.36 \cdot 10^{-3}$ | $9.00 \cdot 10^{-4}$ | 2.62 x | General |
| Mixed_6b.branch1x1.conv | $2.16 \cdot 10^{-3}$ | $1.22 \cdot 10^{-3}$ | 1.77 x | $1.41 \cdot 10^{-3}$ | $1.22 \cdot 10^{-3}$ | 1.15 x | Dense |
| Mixed_6b.branch7x7_1.conv | $1.67 \cdot 10^{-3}$ | $8.15 \cdot 10^{-4}$ | 2.05 x | $1.10 \cdot 10^{-3}$ | $8.16 \cdot 10^{-4}$ | 1.35 x | Dense |
| Mixed_6b.branch7x7_2.conv | $2.14 \cdot 10^{-3}$ | $8.04 \cdot 10^{-4}$ | 2.66 x | $1.76 \cdot 10^{-3}$ | $8.05 \cdot 10^{-4}$ | 2.19 x | Dense mix |
| Mixed_6b.branch7x7_3.conv | $2.59 \cdot 10^{-3}$ | $1.06 \cdot 10^{-3}$ | 2.45 x | $2.27 \cdot 10^{-3}$ | $1.06 \cdot 10^{-3}$ | 2.15 x | Dense mix |
| Mixed_6b.branch7x7dbl_2.conv | $2.17 \cdot 10^{-3}$ | $7.88 \cdot 10^{-4}$ | 2.76 x | $1.78 \cdot 10^{-3}$ | $7.88 \cdot 10^{-4}$ | 2.26 x | Dense mix |
| Mixed_6b.branch7x7dbl_5.conv | $2.63 \cdot 10^{-3}$ | $1.07 \cdot 10^{-3}$ | 2.46 x | $2.25 \cdot 10^{-3}$ | $1.07 \cdot 10^{-3}$ | 2.11 x | Dense mix |
| Mixed_6c.branch7x7_1.conv | $2.05 \cdot 10^{-3}$ | $1.16 \cdot 10^{-3}$ | 1.77 x | $1.41 \cdot 10^{-3}$ | $1.16 \cdot 10^{-3}$ | 1.21 x | Dense |
| Mixed_6c.branch7x7_2.conv | $3.19 \cdot 10^{-3}$ | $1.12 \cdot 10^{-3}$ | 2.84 x | $2.72 \cdot 10^{-3}$ | $1.12 \cdot 10^{-3}$ | 2.42 x | Dense mix |
| Mixed_6c.branch7x7_3.conv | $3.12 \cdot 10^{-3}$ | $1.25 \cdot 10^{-3}$ | 2.50 x | $2.76 \cdot 10^{-3}$ | $1.25 \cdot 10^{-3}$ | 2.21 x | Dense mix |
| Mixed_6c.branch7x7dbl_2.conv | $3.25 \cdot 10^{-3}$ | $1.10 \cdot 10^{-3}$ | 2.96 x | $2.75 \cdot 10^{-3}$ | $1.10 \cdot 10^{-3}$ | 2.51 x | Dense mix |
| Mixed_6c.branch7x7dbl_5.conv | $3.19 \cdot 10^{-3}$ | $1.28 \cdot 10^{-3}$ | 2.49 x | $2.73 \cdot 10^{-3}$ | $1.29 \cdot 10^{-3}$ | 2.12 x | Dense mix |
| Mixed_6e.branch7x7_2.conv | $3.78 \cdot 10^{-3}$ | $1.48 \cdot 10^{-3}$ | 2.54 x | $3.21 \cdot 10^{-3}$ | $1.48 \cdot 10^{-3}$ | 2.16 x | Dense mix |
| Mixed_6e.branch7x7_3.conv | $3.87 \cdot 10^{-3}$ | $1.45 \cdot 10^{-3}$ | 2.66 x | $3.26 \cdot 10^{-3}$ | $1.46 \cdot 10^{-3}$ | 2.24 x | Dense mix |
| AuxLogits.conv0.conv | $6.40 \cdot 10^{-4}$ | $2.38 \cdot 10^{-4}$ | 2.69 x | $3.20 \cdot 10^{-4}$ | $2.39 \cdot 10^{-4}$ | 1.34 x | Dense |
| AuxLogits.conv1.conv | $8.06 \cdot 10^{-4}$ | $1.53 \cdot 10^{-3}$ | **0.53 x** | $6.98 \cdot 10^{-4}$ | $1.52 \cdot 10^{-3}$ | **0.46 x** | General |
| Mixed_7a.branch3x3_2.conv | $1.08 \cdot 10^{-3}$ | $4.37 \cdot 10^{-4}$ | 2.48 x | $1.09 \cdot 10^{-3}$ | $5.01 \cdot 10^{-4}$ | 2.18 x | General |
| Mixed_7a.branch7x7x3_4.conv | $1.54 \cdot 10^{-3}$ | $8.89 \cdot 10^{-4}$ | 1.73 x | $1.52 \cdot 10^{-3}$ | $8.88 \cdot 10^{-4}$ | 1.71 x | General |
| Mixed_7b.branch1x1.conv | $1.29 \cdot 10^{-3}$ | $7.43 \cdot 10^{-4}$ | 1.73 x | $8.76 \cdot 10^{-4}$ | $7.43 \cdot 10^{-4}$ | 1.18 x | Dense |
| Mixed_7b.branch3x3_1.conv | $1.47 \cdot 10^{-3}$ | $1.03 \cdot 10^{-3}$ | 1.42 x | $1.02 \cdot 10^{-3}$ | $1.03 \cdot 10^{-3}$ | **0.99 x** | Dense |
| Mixed_7b.branch3x3_2a.conv | $1.49 \cdot 10^{-3}$ | $9.36 \cdot 10^{-4}$ | 1.59 x | $1.26 \cdot 10^{-3}$ | $9.38 \cdot 10^{-4}$ | 1.34 x | Dense mix |
| Mixed_7b.branch3x3_2b.conv | $1.46 \cdot 10^{-3}$ | $9.37 \cdot 10^{-4}$ | 1.56 x | $1.28 \cdot 10^{-3}$ | $9.37 \cdot 10^{-4}$ | 1.37 x | Dense mix |
| Mixed_7b.branch3x3dbl_1.conv | $1.67 \cdot 10^{-3}$ | $1.04 \cdot 10^{-3}$ | 1.61 x | $1.17 \cdot 10^{-3}$ | $1.04 \cdot 10^{-3}$ | 1.13 x | Dense |
| Mixed_7b.branch3x3dbl_2.conv | $3.18 \cdot 10^{-3}$ | $9.82 \cdot 10^{-4}$ | 3.23 x | $3.21 \cdot 10^{-3}$ | $9.83 \cdot 10^{-4}$ | 3.26 x | General |
| Mixed_7b.branch_pool.conv | $9.54 \cdot 10^{-4}$ | $6.76 \cdot 10^{-4}$ | 1.41 x | $6.30 \cdot 10^{-4}$ | $6.75 \cdot 10^{-4}$ | **0.93 x** | Dense |
| Mixed_7c.branch1x1.conv | $1.68 \cdot 10^{-3}$ | $1.08 \cdot 10^{-3}$ | 1.56 x | $1.27 \cdot 10^{-3}$ | $1.08 \cdot 10^{-3}$ | 1.18 x | Dense |
| Mixed_7c.branch3x3_1.conv | $1.98 \cdot 10^{-3}$ | $1.60 \cdot 10^{-3}$ | 1.23 x | $1.51 \cdot 10^{-3}$ | $1.60 \cdot 10^{-3}$ | **0.94 x** | Dense |
| Mixed_7c.branch3x3dbl_1.conv | $2.25 \cdot 10^{-3}$ | $1.56 \cdot 10^{-3}$ | 1.44 x | $1.73 \cdot 10^{-3}$ | $1.56 \cdot 10^{-3}$ | 1.11 x | Dense |
| Mixed_7c.branch_pool.conv | $1.25 \cdot 10^{-3}$ | $1.04 \cdot 10^{-3}$ | 1.20 x | $9.35 \cdot 10^{-4}$ | $1.04 \cdot 10^{-3}$ | **0.90 x** | Dense |

(i) MobileNetV2, input shape (32, 3, 256, 256)

| Name | TN [s] | PT [s] | Factor | TN + opt [s] | PT [s] | Factor | Type |
|---|---|---|---|---|---|---|---|
| features.0.0 | $6.91 \cdot 10^{-3}$ | $7.23 \cdot 10^{-4}$ | 9.55 x | $6.92 \cdot 10^{-3}$ | $7.24 \cdot 10^{-4}$ | 9.56 x | General |
| features.1.conv.0.0 | $2.28 \cdot 10^{-2}$ | $2.05 \cdot 10^{-3}$ | 11.11 x | $2.28 \cdot 10^{-2}$ | $2.06 \cdot 10^{-3}$ | 11.09 x | General |
| features.1.conv.1 | $5.64 \cdot 10^{-3}$ | $7.61 \cdot 10^{-4}$ | 7.42 x | $1.56 \cdot 10^{-3}$ | $7.57 \cdot 10^{-4}$ | 2.06 x | Dense |
| features.2.conv.0.0 | $4.27 \cdot 10^{-3}$ | $1.74 \cdot 10^{-3}$ | 2.45 x | $2.02 \cdot 10^{-3}$ | $1.74 \cdot 10^{-3}$ | 1.16 x | Dense |
| features.2.conv.1.0 | $3.31 \cdot 10^{-2}$ | $1.69 \cdot 10^{-3}$ | 19.55 x | $3.31 \cdot 10^{-2}$ | $1.69 \cdot 10^{-3}$ | 19.56 x | General |
| features.2.conv.2 | $2.53 \cdot 10^{-3}$ | $4.93 \cdot 10^{-4}$ | 5.13 x | $1.08 \cdot 10^{-3}$ | $4.99 \cdot 10^{-4}$ | 2.16 x | Dense |
| features.3.conv.0.0 | $1.78 \cdot 10^{-3}$ | $7.88 \cdot 10^{-4}$ | 2.26 x | $9.63 \cdot 10^{-4}$ | $7.88 \cdot 10^{-4}$ | 1.22 x | Dense |
| features.3.conv.1.0 | $2.09 \cdot 10^{-2}$ | $2.30 \cdot 10^{-3}$ | 9.07 x | $2.09 \cdot 10^{-2}$ | $2.30 \cdot 10^{-3}$ | 9.06 x | General |
| features.3.conv.2 | $2.93 \cdot 10^{-3}$ | $6.33 \cdot 10^{-4}$ | 4.63 x | $1.47 \cdot 10^{-3}$ | $6.34 \cdot 10^{-4}$ | 2.33 x | Dense |
| features.4.conv.1.0 | $1.04 \cdot 10^{-2}$ | $6.62 \cdot 10^{-4}$ | 15.76 x | $1.04 \cdot 10^{-2}$ | $6.63 \cdot 10^{-4}$ | 15.72 x | General |
| features.4.conv.2 | $1.10 \cdot 10^{-3}$ | $2.61 \cdot 10^{-4}$ | 4.23 x | $5.03 \cdot 10^{-4}$ | $2.61 \cdot 10^{-4}$ | 1.92 x | Dense |
| features.5.conv.0.0 | $9.24 \cdot 10^{-4}$ | $3.32 \cdot 10^{-4}$ | 2.78 x | $5.07 \cdot 10^{-4}$ | $3.33 \cdot 10^{-4}$ | 1.52 x | Dense |
| features.5.conv.1.0 | $5.44 \cdot 10^{-3}$ | $7.87 \cdot 10^{-4}$ | 6.91 x | $5.42 \cdot 10^{-3}$ | $7.88 \cdot 10^{-4}$ | 6.88 x | General |
| features.5.conv.2 | $1.22 \cdot 10^{-3}$ | $3.11 \cdot 10^{-4}$ | 3.93 x | $6.16 \cdot 10^{-4}$ | $3.11 \cdot 10^{-4}$ | 1.98 x | Dense |
| features.7.conv.1.0 | $2.38 \cdot 10^{-3}$ | $2.49 \cdot 10^{-4}$ | 9.58 x | $2.37 \cdot 10^{-3}$ | $2.51 \cdot 10^{-4}$ | 9.44 x | General |
| features.7.conv.2 | $7.49 \cdot 10^{-4}$ | $2.09 \cdot 10^{-4}$ | 3.58 x | $3.20 \cdot 10^{-4}$ | $2.10 \cdot 10^{-4}$ | 1.53 x | Dense |
| features.8.conv.0.0 | $8.05 \cdot 10^{-4}$ | $2.91 \cdot 10^{-4}$ | 2.77 x | $4.42 \cdot 10^{-4}$ | $2.92 \cdot 10^{-4}$ | 1.51 x | Dense |
| features.8.conv.1.0 | $2.29 \cdot 10^{-3}$ | $4.14 \cdot 10^{-4}$ | 5.53 x | $2.27 \cdot 10^{-3}$ | $4.15 \cdot 10^{-4}$ | 5.48 x | General |
| features.8.conv.2 | $7.98 \cdot 10^{-4}$ | $3.07 \cdot 10^{-4}$ | 2.60 x | $4.63 \cdot 10^{-4}$ | $3.06 \cdot 10^{-4}$ | 1.51 x | Dense |
| features.11.conv.2 | $9.88 \cdot 10^{-4}$ | $4.08 \cdot 10^{-4}$ | 2.42 x | $5.67 \cdot 10^{-4}$ | $4.07 \cdot 10^{-4}$ | 1.39 x | Dense |
| features.12.conv.0.0 | $1.06 \cdot 10^{-3}$ | $4.92 \cdot 10^{-4}$ | 2.16 x | $5.64 \cdot 10^{-4}$ | $4.92 \cdot 10^{-4}$ | 1.14 x | Dense |
| features.12.conv.1.0 | $4.18 \cdot 10^{-3}$ | $6.04 \cdot 10^{-4}$ | 6.91 x | $4.16 \cdot 10^{-3}$ | $6.05 \cdot 10^{-4}$ | 6.87 x | General |
| features.12.conv.2 | $1.16 \cdot 10^{-3}$ | $5.53 \cdot 10^{-4}$ | 2.10 x | $7.40 \cdot 10^{-4}$ | $5.55 \cdot 10^{-4}$ | 1.33 x | Dense |
| features.14.conv.1.0 | $1.73 \cdot 10^{-3}$ | $2.29 \cdot 10^{-4}$ | 7.57 x | $1.72 \cdot 10^{-3}$ | $2.28 \cdot 10^{-4}$ | 7.53 x | General |
| features.14.conv.2 | $6.95 \cdot 10^{-4}$ | $3.90 \cdot 10^{-4}$ | 1.78 x | $4.10 \cdot 10^{-4}$ | $3.90 \cdot 10^{-4}$ | 1.05 x | Dense |
| features.15.conv.0.0 | $9.24 \cdot 10^{-4}$ | $3.53 \cdot 10^{-4}$ | 2.62 x | $4.36 \cdot 10^{-4}$ | $3.53 \cdot 10^{-4}$ | 1.23 x | Dense |
| features.15.conv.1.0 | $1.49 \cdot 10^{-3}$ | $2.72 \cdot 10^{-4}$ | 5.46 x | $1.47 \cdot 10^{-3}$ | $2.73 \cdot 10^{-4}$ | 5.39 x | General |
| features.15.conv.2 | $8.32 \cdot 10^{-4}$ | $5.80 \cdot 10^{-4}$ | 1.43 x | $5.44 \cdot 10^{-4}$ | $5.80 \cdot 10^{-4}$ | **0.94 x** | Dense |
| features.17.conv.2 | $1.12 \cdot 10^{-3}$ | $9.74 \cdot 10^{-4}$ | 1.15 x | $7.14 \cdot 10^{-4}$ | $9.75 \cdot 10^{-4}$ | **0.73 x** | Dense |
| features.18.0 | $1.25 \cdot 10^{-3}$ | $7.31 \cdot 10^{-4}$ | 1.71 x | $8.01 \cdot 10^{-4}$ | $7.31 \cdot 10^{-4}$ | 1.10 x | Dense |

## F.3 INPUT VJP

We compare TN and TN+opt with a PyTorch implementation of the input VJP via `torch.autograd.grad`. Figure F19 visualizes the performance ratios for different convolution categories. Table F5 contains the detailed run times and performance factors.

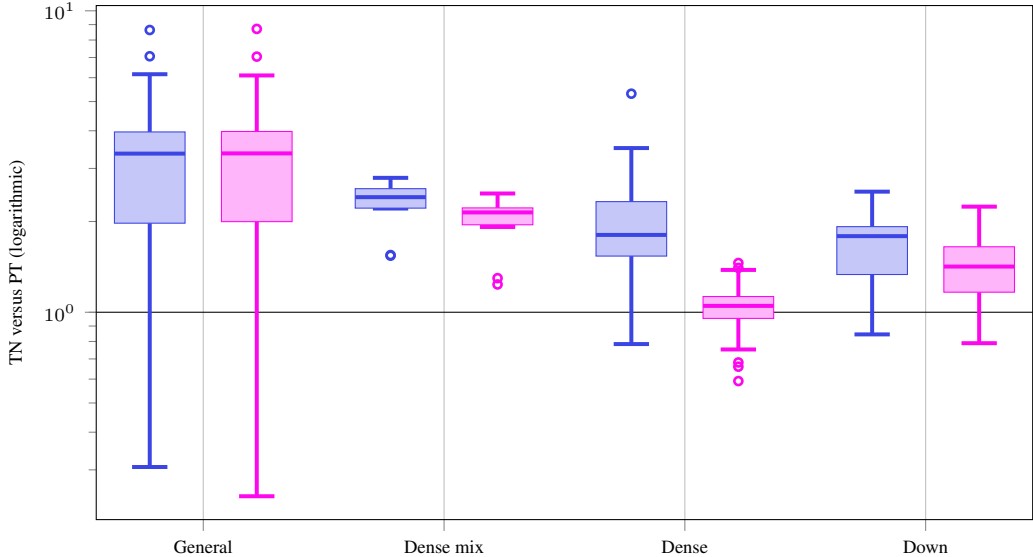

Figure F19: Input VJP performance ratios of TN versus PT and TN+opt versus PT for different convolution types on GPU.

Table F5: Input VJP performance comparison on GPU.

(a) 3c3d, CIFAR-10, input shape (128, 3, 32, 32)

| Name | TN [s] | PT [s] | Factor | TN + opt [s] | PT [s] | Factor | Type |
|------|--------|--------|--------|--------------|--------|--------|------|
| conv1.0 | $2.24 \cdot 10^{-3}$ | $1.39 \cdot 10^{-3}$ | 1.61 x | $2.19 \cdot 10^{-3}$ | $1.34 \cdot 10^{-3}$ | 1.63 x | General |
| conv2.0 | $2.61 \cdot 10^{-3}$ | $8.29 \cdot 10^{-4}$ | 3.15 x | $2.55 \cdot 10^{-3}$ | $7.86 \cdot 10^{-4}$ | 3.25 x | General |
| conv3.1 | $1.46 \cdot 10^{-3}$ | $5.04 \cdot 10^{-4}$ | 2.90 x | $1.42 \cdot 10^{-3}$ | $4.69 \cdot 10^{-4}$ | 3.02 x | General |

(b) F-MNIST 2c2d, input shape (128, 1, 28, 28)

| Name | TN [s] | PT [s] | Factor | TN + opt [s] | PT [s] | Factor | Type |
|------|--------|--------|--------|--------------|--------|--------|------|
| conv1.1 | $9.47 \cdot 10^{-4}$ | $4.36 \cdot 10^{-4}$ | 2.17 x | $8.86 \cdot 10^{-4}$ | $4.40 \cdot 10^{-4}$ | 2.02 x | General |
| conv2.1 | $3.67 \cdot 10^{-3}$ | $9.83 \cdot 10^{-4}$ | 3.74 x | $3.62 \cdot 10^{-3}$ | $9.83 \cdot 10^{-4}$ | 3.69 x | General |

(c) CIFAR-100 All-CNN-C, input shape (128, 3, 32, 32)

| Name | TN [s] | PT [s] | Factor | TN + opt [s] | PT [s] | Factor | Type |
|------|--------|--------|--------|--------------|--------|--------|------|
| conv1.1 | $1.91 \cdot 10^{-3}$ | $9.84 \cdot 10^{-4}$ | 1.94 x | $1.87 \cdot 10^{-3}$ | $9.37 \cdot 10^{-4}$ | 2.00 x | General |
| conv2.1 | $2.00 \cdot 10^{-2}$ | $5.95 \cdot 10^{-3}$ | 3.35 x | $2.00 \cdot 10^{-2}$ | $5.92 \cdot 10^{-3}$ | 3.37 x | General |
| conv3.1 | $7.82 \cdot 10^{-3}$ | $5.05 \cdot 10^{-3}$ | 1.55 x | $7.77 \cdot 10^{-3}$ | $5.01 \cdot 10^{-3}$ | 1.55 x | General |
| conv4.1 | $8.23 \cdot 10^{-3}$ | $3.11 \cdot 10^{-3}$ | 2.65 x | $8.17 \cdot 10^{-3}$ | $3.10 \cdot 10^{-3}$ | 2.63 x | General |
| conv5.1 | $1.56 \cdot 10^{-2}$ | $4.36 \cdot 10^{-3}$ | 3.57 x | $1.55 \cdot 10^{-2}$ | $4.36 \cdot 10^{-3}$ | 3.56 x | General |
| conv6.1 | $4.58 \cdot 10^{-3}$ | $3.96 \cdot 10^{-3}$ | 1.16 x | $4.53 \cdot 10^{-3}$ | $3.96 \cdot 10^{-3}$ | 1.14 x | General |
| conv7.0 | $2.86 \cdot 10^{-3}$ | $8.32 \cdot 10^{-4}$ | 3.44 x | $2.81 \cdot 10^{-3}$ | $8.68 \cdot 10^{-4}$ | 3.24 x | General |
| conv8.1 | $8.31 \cdot 10^{-4}$ | $2.91 \cdot 10^{-4}$ | 2.85 x | $3.47 \cdot 10^{-4}$ | $3.32 \cdot 10^{-4}$ | 1.04 x | Dense |
| conv9.1 | $7.76 \cdot 10^{-4}$ | $2.21 \cdot 10^{-4}$ | 3.51 x | $2.90 \cdot 10^{-4}$ | $2.61 \cdot 10^{-4}$ | 1.11 x | Dense |

(d) Alexnet, input shape (32, 3, 256, 256)

| Name | TN [s] | PT [s] | Factor | TN + opt [s] | PT [s] | Factor | Type |
|------|--------|--------|--------|--------------|--------|--------|------|
| features.0 | $1.92 \cdot 10^{-2}$ | $5.48 \cdot 10^{-3}$ | 3.50 x | $1.92 \cdot 10^{-2}$ | $5.54 \cdot 10^{-3}$ | 3.46 x | General |
| features.3 | $1.15 \cdot 10^{-2}$ | $4.16 \cdot 10^{-3}$ | 2.77 x | $1.15 \cdot 10^{-2}$ | $4.20 \cdot 10^{-3}$ | 2.75 x | General |
| features.6 | $5.36 \cdot 10^{-3}$ | $1.49 \cdot 10^{-3}$ | 3.60 x | $5.36 \cdot 10^{-3}$ | $1.49 \cdot 10^{-3}$ | 3.60 x | General |
| features.8 | $6.26 \cdot 10^{-3}$ | $1.86 \cdot 10^{-3}$ | 3.36 x | $6.25 \cdot 10^{-3}$ | $1.86 \cdot 10^{-3}$ | 3.37 x | General |
| features.10 | $4.41 \cdot 10^{-3}$ | $1.32 \cdot 10^{-3}$ | 3.35 x | $4.40 \cdot 10^{-3}$ | $1.35 \cdot 10^{-3}$ | 3.26 x | General |

(e) ResNet18, input shape (32, 3, 256, 256)

| Name | TN [s] | PT [s] | Factor | TN + opt [s] | PT [s] | Factor | Type |
|------|--------|--------|--------|--------------|--------|--------|------|
| conv1 | $3.38 \cdot 10^{-2}$ | $8.56 \cdot 10^{-3}$ | 3.96 x | $3.38 \cdot 10^{-2}$ | $8.49 \cdot 10^{-3}$ | 3.98 x | General |
| layer1.0.conv1 | $1.06 \cdot 10^{-2}$ | $2.17 \cdot 10^{-3}$ | 4.87 x | $1.05 \cdot 10^{-2}$ | $2.11 \cdot 10^{-3}$ | 4.99 x | General |
| layer2.0.conv1 | $7.03 \cdot 10^{-3}$ | $3.72 \cdot 10^{-3}$ | 1.89 x | $6.95 \cdot 10^{-3}$ | $3.66 \cdot 10^{-3}$ | 1.90 x | General |
| layer2.0.conv2 | $6.91 \cdot 10^{-3}$ | $1.55 \cdot 10^{-3}$ | 4.47 x | $6.90 \cdot 10^{-3}$ | $1.51 \cdot 10^{-3}$ | 4.56 x | General |
| layer2.0.downsample.0 | $2.02 \cdot 10^{-3}$ | $8.02 \cdot 10^{-4}$ | 2.51 x | $1.71 \cdot 10^{-3}$ | $7.64 \cdot 10^{-4}$ | 2.24 x | Down |
| layer3.0.conv1 | $3.94 \cdot 10^{-3}$ | $3.05 \cdot 10^{-3}$ | 1.29 x | $3.88 \cdot 10^{-3}$ | $3.01 \cdot 10^{-3}$ | 1.29 x | General |
| layer3.0.conv2 | $5.07 \cdot 10^{-3}$ | $1.31 \cdot 10^{-3}$ | 3.87 x | $5.07 \cdot 10^{-3}$ | $1.36 \cdot 10^{-3}$ | 3.74 x | General |
| layer3.0.downsample.0 | $1.15 \cdot 10^{-3}$ | $5.96 \cdot 10^{-4}$ | 1.94 x | $9.54 \cdot 10^{-4}$ | $6.40 \cdot 10^{-4}$ | 1.49 x | Down |
| layer4.0.conv1 | $2.89 \cdot 10^{-3}$ | $3.08 \cdot 10^{-3}$ | **0.94 x** | $2.84 \cdot 10^{-3}$ | $3.12 \cdot 10^{-3}$ | **0.91 x** | General |
| layer4.0.conv2 | $4.50 \cdot 10^{-3}$ | $1.40 \cdot 10^{-3}$ | 3.21 x | $4.49 \cdot 10^{-3}$ | $1.44 \cdot 10^{-3}$ | 3.12 x | General |
| layer4.0.downsample.0 | $9.35 \cdot 10^{-4}$ | $5.51 \cdot 10^{-4}$ | 1.70 x | $7.93 \cdot 10^{-4}$ | $5.90 \cdot 10^{-4}$ | 1.34 x | Down |

(f) ResNext101, input shape (32, 3, 256, 256)

| Name | TN [s] | PT [s] | Factor | TN + opt [s] | PT [s] | Factor | Type |
|------|--------|--------|--------|--------------|--------|--------|------|
| conv1 | $3.38 \cdot 10^{-2}$ | $8.52 \cdot 10^{-3}$ | 3.97 x | $3.38 \cdot 10^{-2}$ | $8.48 \cdot 10^{-3}$ | 3.98 x | General |
| layer1.0.conv1 | $6.18 \cdot 10^{-3}$ | $1.96 \cdot 10^{-3}$ | 3.15 x | $2.86 \cdot 10^{-3}$ | $1.96 \cdot 10^{-3}$ | 1.46 x | Dense |
| layer1.0.conv2 | $3.04 \cdot 10^{-2}$ | $1.17 \cdot 10^{-2}$ | 2.60 x | $3.05 \cdot 10^{-2}$ | $1.17 \cdot 10^{-2}$ | 2.61 x | General |
| layer1.0.conv3 | $1.46 \cdot 10^{-2}$ | $6.57 \cdot 10^{-3}$ | 2.22 x | $7.39 \cdot 10^{-3}$ | $6.58 \cdot 10^{-3}$ | 1.12 x | Dense |
| layer2.0.conv1 | $2.40 \cdot 10^{-2}$ | $1.14 \cdot 10^{-2}$ | 2.10 x | $1.44 \cdot 10^{-2}$ | $1.17 \cdot 10^{-2}$ | 1.23 x | Dense |
| layer2.0.conv2 | $2.75 \cdot 10^{-2}$ | $1.96 \cdot 10^{-2}$ | 1.40 x | $2.75 \cdot 10^{-2}$ | $1.95 \cdot 10^{-2}$ | 1.41 x | General |
| layer2.0.conv3 | $1.04 \cdot 10^{-2}$ | $6.43 \cdot 10^{-3}$ | 1.61 x | $6.74 \cdot 10^{-3}$ | $6.43 \cdot 10^{-3}$ | 1.05 x | Dense |
| layer2.0.downsample.0 | $8.99 \cdot 10^{-3}$ | $4.78 \cdot 10^{-3}$ | 1.88 x | $8.06 \cdot 10^{-3}$ | $4.74 \cdot 10^{-3}$ | 1.70 x | Down |
| layer2.1.conv2 | $1.51 \cdot 10^{-2}$ | $4.46 \cdot 10^{-3}$ | 3.38 x | $1.51 \cdot 10^{-2}$ | $4.45 \cdot 10^{-3}$ | 3.39 x | General |
| layer3.0.conv1 | $1.94 \cdot 10^{-2}$ | $1.25 \cdot 10^{-2}$ | 1.55 x | $1.32 \cdot 10^{-2}$ | $1.25 \cdot 10^{-2}$ | 1.06 x | Dense |
| layer3.0.conv2 | $1.76 \cdot 10^{-2}$ | $8.33 \cdot 10^{-3}$ | 2.11 x | $1.76 \cdot 10^{-2}$ | $8.34 \cdot 10^{-3}$ | 2.11 x | General |
| layer3.0.conv3 | $8.21 \cdot 10^{-3}$ | $6.32 \cdot 10^{-3}$ | 1.30 x | $6.39 \cdot 10^{-3}$ | $6.32 \cdot 10^{-3}$ | 1.01 x | Dense |
| layer3.0.downsample.0 | $5.51 \cdot 10^{-3}$ | $4.54 \cdot 10^{-3}$ | 1.21 x | $5.00 \cdot 10^{-3}$ | $4.52 \cdot 10^{-3}$ | 1.11 x | Down |
| layer3.1.conv2 | $7.60 \cdot 10^{-3}$ | $1.97 \cdot 10^{-3}$ | 3.85 x | $7.60 \cdot 10^{-3}$ | $1.98 \cdot 10^{-3}$ | 3.84 x | General |
| layer4.0.conv1 | $1.51 \cdot 10^{-2}$ | $1.24 \cdot 10^{-2}$ | 1.22 x | $1.26 \cdot 10^{-2}$ | $1.24 \cdot 10^{-2}$ | 1.02 x | Dense |
| layer4.0.conv2 | $8.24 \cdot 10^{-3}$ | $5.43 \cdot 10^{-3}$ | 1.52 x | $8.24 \cdot 10^{-3}$ | $5.44 \cdot 10^{-3}$ | 1.51 x | General |
| layer4.0.conv3 | $7.65 \cdot 10^{-3}$ | $6.72 \cdot 10^{-3}$ | 1.14 x | $6.25 \cdot 10^{-3}$ | $6.73 \cdot 10^{-3}$ | **0.93 x** | Dense |
| layer4.0.downsample.0 | $4.61 \cdot 10^{-3}$ | $5.45 \cdot 10^{-3}$ | **0.84 x** | $4.31 \cdot 10^{-3}$ | $5.45 \cdot 10^{-3}$ | **0.79 x** | Down |
| layer4.1.conv2 | $4.79 \cdot 10^{-3}$ | $1.34 \cdot 10^{-3}$ | 3.57 x | $4.79 \cdot 10^{-3}$ | $1.39 \cdot 10^{-3}$ | 3.44 x | General |

(g) ConvNeXt-base, input shape (32, 3, 256, 256)

| Name | TN [s] | PT [s] | Factor | TN + opt [s] | PT [s] | Factor | Type |
|------|--------|--------|--------|--------------|--------|--------|------|
| features.0.0 | $5.36 \cdot 10^{-3}$ | $1.79 \cdot 10^{-3}$ | 2.99 x | $1.57 \cdot 10^{-3}$ | $1.79 \cdot 10^{-3}$ | **0.88 x** | Dense |
| features.1.0.block.0 | $4.63 \cdot 10^{-2}$ | $8.60 \cdot 10^{-3}$ | 5.38 x | $4.63 \cdot 10^{-2}$ | $8.58 \cdot 10^{-3}$ | 5.40 x | General |
| features.2.1 | $8.85 \cdot 10^{-3}$ | $5.37 \cdot 10^{-3}$ | 1.65 x | $3.55 \cdot 10^{-3}$ | $5.38 \cdot 10^{-3}$ | **0.66 x** | Dense |
| features.3.0.block.0 | $2.14 \cdot 10^{-2}$ | $4.21 \cdot 10^{-3}$ | 5.09 x | $2.14 \cdot 10^{-2}$ | $4.21 \cdot 10^{-3}$ | 5.09 x | General |
| features.4.1 | $5.64 \cdot 10^{-3}$ | $4.43 \cdot 10^{-3}$ | 1.27 x | $3.34 \cdot 10^{-3}$ | $4.43 \cdot 10^{-3}$ | **0.75 x** | Dense |
| features.5.0.block.0 | $1.05 \cdot 10^{-2}$ | $2.16 \cdot 10^{-3}$ | 4.87 x | $1.05 \cdot 10^{-2}$ | $2.16 \cdot 10^{-3}$ | 4.86 x | General |
| features.6.1 | $4.31 \cdot 10^{-3}$ | $5.50 \cdot 10^{-3}$ | **0.78 x** | $3.25 \cdot 10^{-3}$ | $5.50 \cdot 10^{-3}$ | **0.59 x** | Dense |
| features.7.0.block.0 | $1.09 \cdot 10^{-3}$ | $1.17 \cdot 10^{-3}$ | **0.93 x** | $1.06 \cdot 10^{-3}$ | $1.15 \cdot 10^{-3}$ | **0.92 x** | General |

(h) InceptionV3, input shape (32, 3, 299, 299)

| Name | TN [s] | PT [s] | Factor | TN + opt [s] | PT [s] | Factor | Type |
|---|---|---|---|---|---|---|---|
| Conv2d_1a_3x3.conv | $1.27 \cdot 10^{-2}$ | $3.19 \cdot 10^{-3}$ | 3.97 x | $1.26 \cdot 10^{-2}$ | $3.21 \cdot 10^{-3}$ | 3.92 x | General |
| Conv2d_2a_3x3.conv | $3.16 \cdot 10^{-2}$ | $5.13 \cdot 10^{-3}$ | 6.16 x | $3.16 \cdot 10^{-2}$ | $5.17 \cdot 10^{-3}$ | 6.10 x | General |
| Conv2d_2b_3x3.conv | $4.32 \cdot 10^{-2}$ | $8.11 \cdot 10^{-3}$ | 5.33 x | $4.24 \cdot 10^{-2}$ | $8.17 \cdot 10^{-3}$ | 5.19 x | General |
| Conv2d_3b_1x1.conv | $5.76 \cdot 10^{-3}$ | $1.09 \cdot 10^{-3}$ | 5.31 x | $1.37 \cdot 10^{-3}$ | $1.09 \cdot 10^{-3}$ | 1.25 x | Dense |
| Conv2d_4a_3x3.conv | $3.71 \cdot 10^{-2}$ | $1.12 \cdot 10^{-2}$ | 3.30 x | $3.71 \cdot 10^{-2}$ | $1.12 \cdot 10^{-2}$ | 3.30 x | General |
| Mixed_5b.branch1x1.conv | $1.37 \cdot 10^{-3}$ | $6.89 \cdot 10^{-4}$ | 1.99 x | $6.67 \cdot 10^{-4}$ | $6.88 \cdot 10^{-4}$ | **0.97 x** | Dense |
| Mixed_5b.branch5x5_1.conv | $1.13 \cdot 10^{-3}$ | $5.87 \cdot 10^{-4}$ | 1.92 x | $5.70 \cdot 10^{-4}$ | $5.88 \cdot 10^{-4}$ | **0.97 x** | Dense |
| Mixed_5b.branch5x5_2.conv | $4.97 \cdot 10^{-3}$ | $1.39 \cdot 10^{-3}$ | 3.58 x | $4.97 \cdot 10^{-3}$ | $1.39 \cdot 10^{-3}$ | 3.57 x | General |
| Mixed_5b.branch3x3dbl_2.conv | $4.23 \cdot 10^{-3}$ | $1.07 \cdot 10^{-3}$ | 3.98 x | $4.23 \cdot 10^{-3}$ | $1.06 \cdot 10^{-3}$ | 3.98 x | General |
| Mixed_5b.branch3x3dbl_3.conv | $5.68 \cdot 10^{-3}$ | $1.66 \cdot 10^{-3}$ | 3.41 x | $5.68 \cdot 10^{-3}$ | $1.66 \cdot 10^{-3}$ | 3.41 x | General |
| Mixed_5b.branch_pool.conv | $9.70 \cdot 10^{-4}$ | $5.10 \cdot 10^{-4}$ | 1.90 x | $4.77 \cdot 10^{-4}$ | $5.13 \cdot 10^{-4}$ | **0.93 x** | Dense |
| Mixed_5c.branch1x1.conv | $1.48 \cdot 10^{-3}$ | $8.10 \cdot 10^{-4}$ | 1.82 x | $8.07 \cdot 10^{-4}$ | $8.10 \cdot 10^{-4}$ | **1.00 x** | Dense |
| Mixed_5c.branch5x5_1.conv | $1.23 \cdot 10^{-3}$ | $6.85 \cdot 10^{-4}$ | 1.79 x | $6.81 \cdot 10^{-4}$ | $6.87 \cdot 10^{-4}$ | **0.99 x** | Dense |
| Mixed_5d.branch1x1.conv | $1.68 \cdot 10^{-3}$ | $1.04 \cdot 10^{-3}$ | 1.61 x | $9.40 \cdot 10^{-4}$ | $1.05 \cdot 10^{-3}$ | **0.90 x** | Dense |
| Mixed_5d.branch5x5_1.conv | $1.38 \cdot 10^{-3}$ | $8.69 \cdot 10^{-4}$ | 1.59 x | $7.84 \cdot 10^{-4}$ | $8.12 \cdot 10^{-4}$ | **0.96 x** | Dense |
| Mixed_6a.branch3x3.conv | $1.14 \cdot 10^{-2}$ | $1.32 \cdot 10^{-2}$ | **0.86 x** | $1.14 \cdot 10^{-2}$ | $1.32 \cdot 10^{-2}$ | **0.86 x** | General |
| Mixed_6a.branch3x3dbl_3.conv | $2.52 \cdot 10^{-3}$ | $1.70 \cdot 10^{-3}$ | 1.48 x | $2.46 \cdot 10^{-3}$ | $1.70 \cdot 10^{-3}$ | 1.45 x | General |
| Mixed_6b.branch1x1.conv | $1.78 \cdot 10^{-3}$ | $1.18 \cdot 10^{-3}$ | 1.50 x | $1.24 \cdot 10^{-3}$ | $1.19 \cdot 10^{-3}$ | 1.05 x | Dense |
| Mixed_6b.branch7x7_1.conv | $1.37 \cdot 10^{-3}$ | $8.69 \cdot 10^{-4}$ | 1.58 x | $9.27 \cdot 10^{-4}$ | $8.70 \cdot 10^{-4}$ | 1.07 x | Dense |
| Mixed_6b.branch7x7_2.conv | $2.13 \cdot 10^{-3}$ | $8.27 \cdot 10^{-4}$ | 2.58 x | $1.79 \cdot 10^{-3}$ | $8.27 \cdot 10^{-4}$ | 2.16 x | Dense mix |
| Mixed_6b.branch7x7_3.conv | $2.54 \cdot 10^{-3}$ | $1.08 \cdot 10^{-3}$ | 2.36 x | $2.22 \cdot 10^{-3}$ | $1.08 \cdot 10^{-3}$ | 2.05 x | Dense mix |
| Mixed_6b.branch7x7dbl_2.conv | $2.08 \cdot 10^{-3}$ | $8.10 \cdot 10^{-4}$ | 2.57 x | $1.80 \cdot 10^{-3}$ | $8.09 \cdot 10^{-4}$ | 2.22 x | Dense mix |
| Mixed_6b.branch7x7dbl_5.conv | $2.45 \cdot 10^{-3}$ | $1.11 \cdot 10^{-3}$ | 2.21 x | $2.14 \cdot 10^{-3}$ | $1.11 \cdot 10^{-3}$ | 1.92 x | Dense mix |
| Mixed_6c.branch7x7_1.conv | $1.56 \cdot 10^{-3}$ | $1.03 \cdot 10^{-3}$ | 1.52 x | $1.09 \cdot 10^{-3}$ | $1.03 \cdot 10^{-3}$ | 1.06 x | Dense |
| Mixed_6c.branch7x7_2.conv | $3.19 \cdot 10^{-3}$ | $1.14 \cdot 10^{-3}$ | 2.79 x | $2.76 \cdot 10^{-3}$ | $1.13 \cdot 10^{-3}$ | 2.43 x | Dense mix |
| Mixed_6c.branch7x7_3.conv | $3.06 \cdot 10^{-3}$ | $1.28 \cdot 10^{-3}$ | 2.40 x | $2.72 \cdot 10^{-3}$ | $1.28 \cdot 10^{-3}$ | 2.12 x | Dense mix |
| Mixed_6c.branch7x7dbl_2.conv | $3.10 \cdot 10^{-3}$ | $1.12 \cdot 10^{-3}$ | 2.78 x | $2.77 \cdot 10^{-3}$ | $1.12 \cdot 10^{-3}$ | 2.48 x | Dense mix |
| Mixed_6c.branch7x7dbl_5.conv | $2.96 \cdot 10^{-3}$ | $1.33 \cdot 10^{-3}$ | 2.22 x | $2.61 \cdot 10^{-3}$ | $1.33 \cdot 10^{-3}$ | 1.96 x | Dense mix |
| Mixed_6e.branch7x7_2.conv | $3.77 \cdot 10^{-3}$ | $1.54 \cdot 10^{-3}$ | 2.45 x | $3.26 \cdot 10^{-3}$ | $1.50 \cdot 10^{-3}$ | 2.17 x | Dense mix |
| Mixed_6e.branch7x7_3.conv | $3.65 \cdot 10^{-3}$ | $1.51 \cdot 10^{-3}$ | 2.42 x | $3.27 \cdot 10^{-3}$ | $1.47 \cdot 10^{-3}$ | 2.22 x | Dense mix |
| AuxLogits.conv0.conv | $5.53 \cdot 10^{-4}$ | $2.74 \cdot 10^{-4}$ | 2.02 x | $3.03 \cdot 10^{-4}$ | $2.34 \cdot 10^{-4}$ | 1.30 x | Dense |
| AuxLogits.conv1.conv | $6.27 \cdot 10^{-4}$ | $2.04 \cdot 10^{-3}$ | **0.31 x** | $4.94 \cdot 10^{-4}$ | $2.02 \cdot 10^{-3}$ | **0.25 x** | General |
| Mixed_7a.branch3x3_2.conv | $1.56 \cdot 10^{-3}$ | $7.08 \cdot 10^{-4}$ | 2.21 x | $1.47 \cdot 10^{-3}$ | $6.64 \cdot 10^{-4}$ | 2.22 x | General |
| Mixed_7a.branch7x7x3_4.conv | $1.46 \cdot 10^{-3}$ | $1.10 \cdot 10^{-3}$ | 1.33 x | $1.42 \cdot 10^{-3}$ | $1.14 \cdot 10^{-3}$ | 1.25 x | General |
| Mixed_7b.branch1x1.conv | $1.31 \cdot 10^{-3}$ | $7.40 \cdot 10^{-4}$ | 1.77 x | $8.47 \cdot 10^{-4}$ | $7.89 \cdot 10^{-4}$ | 1.07 x | Dense |
| Mixed_7b.branch3x3_1.conv | $1.44 \cdot 10^{-3}$ | $8.55 \cdot 10^{-4}$ | 1.69 x | $9.54 \cdot 10^{-4}$ | $9.00 \cdot 10^{-4}$ | 1.06 x | Dense |
| Mixed_7b.branch3x3_2a.conv | $1.51 \cdot 10^{-3}$ | $9.77 \cdot 10^{-4}$ | 1.55 x | $1.26 \cdot 10^{-3}$ | $1.02 \cdot 10^{-3}$ | 1.24 x | Dense mix |
| Mixed_7b.branch3x3_2b.conv | $1.50 \cdot 10^{-3}$ | $9.77 \cdot 10^{-4}$ | 1.54 x | $1.27 \cdot 10^{-3}$ | $9.78 \cdot 10^{-4}$ | 1.30 x | Dense mix |
| Mixed_7b.branch3x3dbl_1.conv | $1.56 \cdot 10^{-3}$ | $1.02 \cdot 10^{-3}$ | 1.54 x | $1.07 \cdot 10^{-3}$ | $9.72 \cdot 10^{-4}$ | 1.10 x | Dense |
| Mixed_7b.branch3x3dbl_2.conv | $3.32 \cdot 10^{-3}$ | $1.02 \cdot 10^{-3}$ | 3.24 x | $3.28 \cdot 10^{-3}$ | $9.91 \cdot 10^{-4}$ | 3.31 x | General |
| Mixed_7b.branch_pool.conv | $1.01 \cdot 10^{-3}$ | $5.57 \cdot 10^{-4}$ | 1.81 x | $6.18 \cdot 10^{-4}$ | $5.10 \cdot 10^{-4}$ | 1.21 x | Dense |
| Mixed_7c.branch1x1.conv | $1.69 \cdot 10^{-3}$ | $1.25 \cdot 10^{-3}$ | 1.35 x | $1.21 \cdot 10^{-3}$ | $1.22 \cdot 10^{-3}$ | **0.99 x** | Dense |
| Mixed_7c.branch3x3_1.conv | $1.86 \cdot 10^{-3}$ | $1.45 \cdot 10^{-3}$ | 1.28 x | $1.39 \cdot 10^{-3}$ | $1.45 \cdot 10^{-3}$ | **0.95 x** | Dense |
| Mixed_7c.branch3x3dbl_1.conv | $2.05 \cdot 10^{-3}$ | $1.66 \cdot 10^{-3}$ | 1.23 x | $1.57 \cdot 10^{-3}$ | $1.66 \cdot 10^{-3}$ | **0.95 x** | Dense |
| Mixed_7c.branch_pool.conv | $1.27 \cdot 10^{-3}$ | $8.35 \cdot 10^{-4}$ | 1.53 x | $8.32 \cdot 10^{-4}$ | $8.35 \cdot 10^{-4}$ | **1.00 x** | Dense |

(i) MobileNetV2, input shape (32, 3, 256, 256)

| Name | TN [s] | PT [s] | Factor | TN + opt [s] | PT [s] | Factor | Type |
|---|---|---|---|---|---|---|---|
| features.0.0 | $8.32 \cdot 10^{-3}$ | $2.08 \cdot 10^{-3}$ | 4.01 x | $8.26 \cdot 10^{-3}$ | $2.02 \cdot 10^{-3}$ | 4.09 x | General |
| features.1.conv.0.0 | $2.27 \cdot 10^{-2}$ | $2.63 \cdot 10^{-3}$ | 8.64 x | $2.27 \cdot 10^{-2}$ | $2.60 \cdot 10^{-3}$ | 8.71 x | General |
| features.1.conv.1 | $3.22 \cdot 10^{-3}$ | $1.17 \cdot 10^{-3}$ | 2.75 x | $1.02 \cdot 10^{-3}$ | $1.16 \cdot 10^{-3}$ | **0.87 x** | Dense |
| features.2.conv.0.0 | $7.67 \cdot 10^{-3}$ | $2.61 \cdot 10^{-3}$ | 2.94 x | $3.66 \cdot 10^{-3}$ | $2.61 \cdot 10^{-3}$ | 1.40 x | Dense |
| features.2.conv.1.0 | $2.79 \cdot 10^{-2}$ | $8.11 \cdot 10^{-3}$ | 3.44 x | $2.79 \cdot 10^{-2}$ | $8.11 \cdot 10^{-3}$ | 3.43 x | General |
| features.2.conv.2 | $1.48 \cdot 10^{-3}$ | $6.38 \cdot 10^{-4}$ | 2.33 x | $7.53 \cdot 10^{-4}$ | $6.40 \cdot 10^{-4}$ | 1.18 x | Dense |
| features.3.conv.0.0 | $2.86 \cdot 10^{-3}$ | $1.05 \cdot 10^{-3}$ | 2.73 x | $1.45 \cdot 10^{-3}$ | $1.05 \cdot 10^{-3}$ | 1.38 x | Dense |
| features.3.conv.1.0 | $2.08 \cdot 10^{-2}$ | $2.95 \cdot 10^{-3}$ | 7.07 x | $2.08 \cdot 10^{-2}$ | $2.95 \cdot 10^{-3}$ | 7.05 x | General |
| features.3.conv.2 | $1.77 \cdot 10^{-3}$ | $1.04 \cdot 10^{-3}$ | 1.70 x | $9.75 \cdot 10^{-4}$ | $1.04 \cdot 10^{-3}$ | **0.94 x** | Dense |
| features.4.conv.1.0 | $7.63 \cdot 10^{-3}$ | $3.15 \cdot 10^{-3}$ | 2.42 x | $7.62 \cdot 10^{-3}$ | $3.15 \cdot 10^{-3}$ | 2.42 x | General |
| features.4.conv.2 | $9.49 \cdot 10^{-4}$ | $4.32 \cdot 10^{-4}$ | 2.20 x | $4.38 \cdot 10^{-4}$ | $3.88 \cdot 10^{-4}$ | 1.13 x | Dense |
| features.5.conv.0.0 | $1.20 \cdot 10^{-3}$ | $5.26 \cdot 10^{-4}$ | 2.29 x | $6.09 \cdot 10^{-4}$ | $4.83 \cdot 10^{-4}$ | 1.26 x | Dense |
| features.5.conv.1.0 | $5.41 \cdot 10^{-3}$ | $1.02 \cdot 10^{-3}$ | 5.29 x | $5.39 \cdot 10^{-3}$ | $1.02 \cdot 10^{-3}$ | 5.27 x | General |
| features.5.conv.2 | $9.53 \cdot 10^{-4}$ | $4.00 \cdot 10^{-4}$ | 2.38 x | $4.35 \cdot 10^{-4}$ | $3.98 \cdot 10^{-4}$ | 1.09 x | Dense |
| features.7.conv.1.0 | $2.11 \cdot 10^{-3}$ | $1.07 \cdot 10^{-3}$ | 1.97 x | $2.10 \cdot 10^{-3}$ | $1.07 \cdot 10^{-3}$ | 1.97 x | General |
| features.7.conv.2 | $7.77 \cdot 10^{-4}$ | $2.33 \cdot 10^{-4}$ | 3.34 x | $3.04 \cdot 10^{-4}$ | $2.33 \cdot 10^{-4}$ | 1.30 x | Dense |
| features.8.conv.0.0 | $8.13 \cdot 10^{-4}$ | $3.41 \cdot 10^{-4}$ | 2.38 x | $4.63 \cdot 10^{-4}$ | $3.40 \cdot 10^{-4}$ | 1.36 x | Dense |
| features.8.conv.1.0 | $2.09 \cdot 10^{-3}$ | $5.48 \cdot 10^{-4}$ | 3.81 x | $2.07 \cdot 10^{-3}$ | $5.47 \cdot 10^{-4}$ | 3.79 x | General |
| features.8.conv.2 | $8.65 \cdot 10^{-4}$ | $3.16 \cdot 10^{-4}$ | 2.74 x | $4.04 \cdot 10^{-4}$ | $3.16 \cdot 10^{-4}$ | 1.28 x | Dense |
| features.11.conv.2 | $9.34 \cdot 10^{-4}$ | $4.22 \cdot 10^{-4}$ | 2.21 x | $4.74 \cdot 10^{-4}$ | $4.24 \cdot 10^{-4}$ | 1.12 x | Dense |
| features.12.conv.0.0 | $1.16 \cdot 10^{-3}$ | $7.11 \cdot 10^{-4}$ | 1.64 x | $7.37 \cdot 10^{-4}$ | $7.10 \cdot 10^{-4}$ | 1.04 x | Dense |
| features.12.conv.1.0 | $3.84 \cdot 10^{-3}$ | $7.91 \cdot 10^{-4}$ | 4.85 x | $3.82 \cdot 10^{-3}$ | $7.91 \cdot 10^{-4}$ | 4.83 x | General |
| features.12.conv.2 | $1.08 \cdot 10^{-3}$ | $5.71 \cdot 10^{-4}$ | 1.90 x | $6.13 \cdot 10^{-4}$ | $5.73 \cdot 10^{-4}$ | 1.07 x | Dense |
| features.14.conv.1.0 | $1.61 \cdot 10^{-3}$ | $8.26 \cdot 10^{-4}$ | 1.95 x | $1.60 \cdot 10^{-3}$ | $8.26 \cdot 10^{-4}$ | 1.93 x | General |
| features.14.conv.2 | $8.14 \cdot 10^{-4}$ | $2.87 \cdot 10^{-4}$ | 2.83 x | $3.84 \cdot 10^{-4}$ | $2.87 \cdot 10^{-4}$ | 1.34 x | Dense |
| features.15.conv.0.0 | $8.46 \cdot 10^{-4}$ | $6.29 \cdot 10^{-4}$ | 1.34 x | $5.55 \cdot 10^{-4}$ | $6.08 \cdot 10^{-4}$ | **0.91 x** | Dense |
| features.15.conv.1.0 | $1.52 \cdot 10^{-3}$ | $3.62 \cdot 10^{-4}$ | 4.21 x | $1.50 \cdot 10^{-3}$ | $3.61 \cdot 10^{-4}$ | 4.17 x | General |
| features.15.conv.2 | $9.64 \cdot 10^{-4}$ | $4.43 \cdot 10^{-4}$ | 2.18 x | $4.82 \cdot 10^{-4}$ | $4.44 \cdot 10^{-4}$ | 1.09 x | Dense |
| features.17.conv.2 | $1.23 \cdot 10^{-3}$ | $7.30 \cdot 10^{-4}$ | 1.69 x | $6.98 \cdot 10^{-4}$ | $7.32 \cdot 10^{-4}$ | **0.95 x** | Dense |
| features.18.0 | $1.29 \cdot 10^{-3}$ | $1.28 \cdot 10^{-3}$ | 1.00 x | $8.76 \cdot 10^{-4}$ | $1.28 \cdot 10^{-3}$ | **0.68 x** | Dense |

## F.4 WEIGHT VJP

We compare TN and TN+opt with a PyTorch implementation of the weight VJP via `torch.autograd.grad`. Figure F20 visualizes the performance ratios for different convolution categories. Table F6 contains the detailed run times and performance factors.

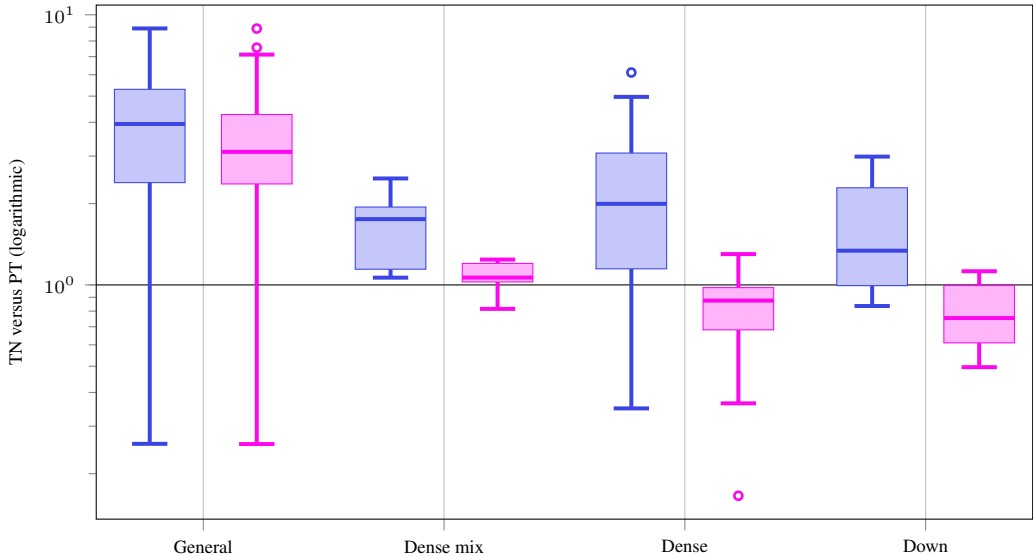

Figure F20: Weight VJP performance ratios of TN versus PT and TN+opt versus PT for different convolution types on GPU.

Table F6: Weight VJP performance comparison on GPU.

(a) 3c3d, CIFAR-10, input shape (128, 3, 32, 32)

| Name | TN [s] | PT [s] | Factor | TN + opt [s] | PT [s] | Factor | Type |
|---|---|---|---|---|---|---|---|
| conv1.0 | $2.27 \cdot 10^{-3}$ | $1.50 \cdot 10^{-3}$ | 1.52 x | $2.27 \cdot 10^{-3}$ | $1.50 \cdot 10^{-3}$ | 1.51 x | General |
| conv2.0 | $3.00 \cdot 10^{-3}$ | $1.12 \cdot 10^{-3}$ | 2.68 x | $2.99 \cdot 10^{-3}$ | $1.07 \cdot 10^{-3}$ | 2.78 x | General |
| conv3.1 | $1.29 \cdot 10^{-3}$ | $5.46 \cdot 10^{-4}$ | 2.37 x | $1.25 \cdot 10^{-3}$ | $5.08 \cdot 10^{-4}$ | 2.47 x | General |

(b) F-MNIST 2c2d, input shape (128, 1, 28, 28)

| Name | TN [s] | PT [s] | Factor | TN + opt [s] | PT [s] | Factor | Type |
|---|---|---|---|---|---|---|---|
| conv1.1 | $1.08 \cdot 10^{-3}$ | $3.81 \cdot 10^{-4}$ | 2.83 x | $1.03 \cdot 10^{-3}$ | $4.05 \cdot 10^{-4}$ | 2.54 x | General |
| conv2.1 | $4.12 \cdot 10^{-3}$ | $1.02 \cdot 10^{-3}$ | 4.02 x | $4.09 \cdot 10^{-3}$ | $1.03 \cdot 10^{-3}$ | 3.98 x | General |

(c) CIFAR-100 All-CNN-C, input shape (128, 3, 32, 32)

| Name | TN [s] | PT [s] | Factor | TN + opt [s] | PT [s] | Factor | Type |
|---|---|---|---|---|---|---|---|
| conv1.1 | $2.43 \cdot 10^{-3}$ | $1.02 \cdot 10^{-3}$ | 2.39 x | $2.42 \cdot 10^{-3}$ | $1.02 \cdot 10^{-3}$ | 2.37 x | General |
| conv2.1 | $3.83 \cdot 10^{-2}$ | $5.62 \cdot 10^{-3}$ | 6.81 x | $1.97 \cdot 10^{-2}$ | $5.62 \cdot 10^{-3}$ | 3.51 x | General |
| conv3.1 | $8.30 \cdot 10^{-3}$ | $4.14 \cdot 10^{-3}$ | 2.00 x | $8.33 \cdot 10^{-3}$ | $4.21 \cdot 10^{-3}$ | 1.98 x | General |
| conv4.1 | $8.66 \cdot 10^{-3}$ | $2.64 \cdot 10^{-3}$ | 3.28 x | $8.68 \cdot 10^{-3}$ | $2.68 \cdot 10^{-3}$ | 3.24 x | General |
| conv5.1 | $1.60 \cdot 10^{-2}$ | $3.38 \cdot 10^{-3}$ | 4.75 x | $1.61 \cdot 10^{-2}$ | $3.42 \cdot 10^{-3}$ | 4.70 x | General |
| conv6.1 | $5.23 \cdot 10^{-3}$ | $2.80 \cdot 10^{-3}$ | 1.87 x | $5.17 \cdot 10^{-3}$ | $2.81 \cdot 10^{-3}$ | 1.84 x | General |
| conv7.0 | $2.68 \cdot 10^{-3}$ | $9.97 \cdot 10^{-4}$ | 2.68 x | $2.59 \cdot 10^{-3}$ | $1.04 \cdot 10^{-3}$ | 2.49 x | General |
| conv8.1 | $9.13 \cdot 10^{-4}$ | $2.62 \cdot 10^{-3}$ | **0.35 x** | $4.33 \cdot 10^{-4}$ | $2.62 \cdot 10^{-3}$ | **0.17 x** | Dense |
| conv9.1 | $8.78 \cdot 10^{-4}$ | $3.54 \cdot 10^{-4}$ | 2.48 x | $3.93 \cdot 10^{-4}$ | $3.50 \cdot 10^{-4}$ | 1.12 x | Dense |

(d) Alexnet, input shape (32, 3, 256, 256)

| Name | TN [s] | PT [s] | Factor | TN + opt [s] | PT [s] | Factor | Type |
|---|---|---|---|---|---|---|---|
| features.0 | $1.82 \cdot 10^{-2}$ | $3.31 \cdot 10^{-3}$ | 5.50 x | $1.82 \cdot 10^{-2}$ | $3.33 \cdot 10^{-3}$ | 5.46 x | General |
| features.3 | $2.02 \cdot 10^{-2}$ | $2.57 \cdot 10^{-3}$ | 7.85 x | $1.14 \cdot 10^{-2}$ | $2.58 \cdot 10^{-3}$ | 4.44 x | General |
| features.6 | $6.98 \cdot 10^{-3}$ | $1.67 \cdot 10^{-3}$ | 4.19 x | $5.17 \cdot 10^{-3}$ | $1.67 \cdot 10^{-3}$ | 3.10 x | General |
| features.8 | $8.16 \cdot 10^{-3}$ | $1.97 \cdot 10^{-3}$ | 4.15 x | $6.13 \cdot 10^{-3}$ | $1.97 \cdot 10^{-3}$ | 3.11 x | General |
| features.10 | $5.80 \cdot 10^{-3}$ | $1.47 \cdot 10^{-3}$ | 3.94 x | $4.34 \cdot 10^{-3}$ | $1.47 \cdot 10^{-3}$ | 2.95 x | General |

(e) ResNet18, input shape (32, 3, 256, 256)

| Name | TN [s] | PT [s] | Factor | TN + opt [s] | PT [s] | Factor | Type |
|---|---|---|---|---|---|---|---|
| conv1 | $3.00 \cdot 10^{-2}$ | $8.23 \cdot 10^{-3}$ | 3.65 x | $2.99 \cdot 10^{-2}$ | $7.83 \cdot 10^{-3}$ | 3.82 x | General |
| layer1.0.conv1 | $2.34 \cdot 10^{-2}$ | $3.18 \cdot 10^{-3}$ | 7.37 x | $1.10 \cdot 10^{-2}$ | $3.22 \cdot 10^{-3}$ | 3.43 x | General |
| layer2.0.conv1 | $5.88 \cdot 10^{-3}$ | $2.97 \cdot 10^{-3}$ | 1.98 x | $5.87 \cdot 10^{-3}$ | $2.96 \cdot 10^{-3}$ | 1.98 x | General |
| layer2.0.conv2 | $1.17 \cdot 10^{-2}$ | $1.66 \cdot 10^{-3}$ | 7.03 x | $6.98 \cdot 10^{-3}$ | $1.66 \cdot 10^{-3}$ | 4.20 x | General |
| layer2.0.downsample.0 | $1.85 \cdot 10^{-3}$ | $7.39 \cdot 10^{-4}$ | 2.51 x | $7.60 \cdot 10^{-4}$ | $7.33 \cdot 10^{-4}$ | 1.04 x | Down |
| layer3.0.conv1 | $3.79 \cdot 10^{-3}$ | $2.71 \cdot 10^{-3}$ | 1.40 x | $3.76 \cdot 10^{-3}$ | $2.71 \cdot 10^{-3}$ | 1.39 x | General |
| layer3.0.conv2 | $6.62 \cdot 10^{-3}$ | $1.48 \cdot 10^{-3}$ | 4.46 x | $4.87 \cdot 10^{-3}$ | $1.48 \cdot 10^{-3}$ | 3.28 x | General |
| layer3.0.downsample.0 | $1.61 \cdot 10^{-3}$ | $5.39 \cdot 10^{-4}$ | 2.99 x | $6.12 \cdot 10^{-4}$ | $5.45 \cdot 10^{-4}$ | 1.12 x | Down |
| layer4.0.conv1 | $2.85 \cdot 10^{-3}$ | $2.46 \cdot 10^{-3}$ | 1.16 x | $2.82 \cdot 10^{-3}$ | $2.46 \cdot 10^{-3}$ | 1.15 x | General |
| layer4.0.conv2 | $4.83 \cdot 10^{-3}$ | $1.72 \cdot 10^{-3}$ | 2.80 x | $4.31 \cdot 10^{-3}$ | $1.72 \cdot 10^{-3}$ | 2.50 x | General |
| layer4.0.downsample.0 | $1.00 \cdot 10^{-3}$ | $1.02 \cdot 10^{-3}$ | **0.98 x** | $5.07 \cdot 10^{-4}$ | $1.02 \cdot 10^{-3}$ | **0.50 x** | Down |

(f) ResNext101, input shape (32, 3, 256, 256)

| Name | TN [s] | PT [s] | Factor | TN + opt [s] | PT [s] | Factor | Type |
|---|---|---|---|---|---|---|---|
| conv1 | $3.00 \cdot 10^{-2}$ | $8.22 \cdot 10^{-3}$ | 3.65 x | $2.99 \cdot 10^{-2}$ | $7.83 \cdot 10^{-3}$ | 3.82 x | General |
| layer1.0.conv1 | $7.08 \cdot 10^{-3}$ | $2.92 \cdot 10^{-3}$ | 2.42 x | $3.75 \cdot 10^{-3}$ | $2.89 \cdot 10^{-3}$ | 1.30 x | Dense |
| layer1.0.conv2 | $6.70 \cdot 10^{-2}$ | $2.53 \cdot 10^{-2}$ | 2.65 x | $6.72 \cdot 10^{-2}$ | $1.91 \cdot 10^{-2}$ | 3.51 x | General |
| layer1.0.conv3 | $3.12 \cdot 10^{-2}$ | $8.78 \cdot 10^{-3}$ | 3.55 x | $1.04 \cdot 10^{-2}$ | $1.02 \cdot 10^{-2}$ | 1.02 x | Dense |
| layer2.0.conv1 | $2.29 \cdot 10^{-2}$ | $1.80 \cdot 10^{-2}$ | 1.28 x | $1.77 \cdot 10^{-2}$ | $1.76 \cdot 10^{-2}$ | 1.01 x | Dense |
| layer2.0.conv2 | $6.64 \cdot 10^{-2}$ | $1.23 \cdot 10^{-2}$ | 5.40 x | $6.63 \cdot 10^{-2}$ | $1.23 \cdot 10^{-2}$ | 5.39 x | General |
| layer2.0.conv3 | $1.82 \cdot 10^{-2}$ | $5.90 \cdot 10^{-3}$ | 3.08 x | $8.44 \cdot 10^{-3}$ | $6.50 \cdot 10^{-3}$ | 1.30 x | Dense |
| layer2.0.downsample.0 | $8.57 \cdot 10^{-3}$ | $5.24 \cdot 10^{-3}$ | 1.64 x | $4.55 \cdot 10^{-3}$ | $5.24 \cdot 10^{-3}$ | **0.87 x** | Down |
| layer2.1.conv2 | $4.04 \cdot 10^{-2}$ | $1.21 \cdot 10^{-2}$ | 3.33 x | $4.04 \cdot 10^{-2}$ | $1.22 \cdot 10^{-2}$ | 3.32 x | General |
| layer3.0.conv1 | $1.84 \cdot 10^{-2}$ | $2.03 \cdot 10^{-2}$ | **0.91 x** | $1.48 \cdot 10^{-2}$ | $2.03 \cdot 10^{-2}$ | **0.73 x** | Dense |
| layer3.0.conv2 | $1.63 \cdot 10^{-2}$ | $5.77 \cdot 10^{-3}$ | 2.83 x | $1.63 \cdot 10^{-2}$ | $5.82 \cdot 10^{-3}$ | 2.81 x | General |
| layer3.0.conv3 | $1.17 \cdot 10^{-2}$ | $1.07 \cdot 10^{-2}$ | 1.10 x | $7.19 \cdot 10^{-3}$ | $1.07 \cdot 10^{-2}$ | **0.67 x** | Dense |
| layer3.0.downsample.0 | $6.19 \cdot 10^{-3}$ | $5.95 \cdot 10^{-3}$ | 1.04 x | $3.85 \cdot 10^{-3}$ | $6.01 \cdot 10^{-3}$ | **0.64 x** | Down |
| layer3.1.conv1 | $1.47 \cdot 10^{-2}$ | $3.17 \cdot 10^{-3}$ | 4.65 x | $1.47 \cdot 10^{-2}$ | $3.14 \cdot 10^{-3}$ | 4.67 x | General |
| layer4.0.conv1 | $1.55 \cdot 10^{-2}$ | $2.10 \cdot 10^{-2}$ | **0.74 x** | $1.33 \cdot 10^{-2}$ | $2.11 \cdot 10^{-2}$ | **0.63 x** | Dense |
| layer4.0.conv2 | $8.07 \cdot 10^{-3}$ | $3.13 \cdot 10^{-2}$ | **0.26 x** | $8.06 \cdot 10^{-3}$ | $3.13 \cdot 10^{-2}$ | **0.26 x** | General |
| layer4.0.conv3 | $8.23 \cdot 10^{-3}$ | $1.06 \cdot 10^{-2}$ | **0.78 x** | $6.75 \cdot 10^{-3}$ | $1.06 \cdot 10^{-2}$ | **0.63 x** | Dense |
| layer4.0.downsample.0 | $4.96 \cdot 10^{-3}$ | $5.94 \cdot 10^{-3}$ | **0.84 x** | $3.59 \cdot 10^{-3}$ | $5.99 \cdot 10^{-3}$ | **0.60 x** | Down |
| layer4.1.conv2 | $6.63 \cdot 10^{-3}$ | $1.40 \cdot 10^{-3}$ | 4.72 x | $6.62 \cdot 10^{-3}$ | $1.45 \cdot 10^{-3}$ | 4.55 x | General |

(g) ConvNeXt-base, input shape (32, 3, 256, 256)

| Name | TN [s] | PT [s] | Factor | TN + opt [s] | PT [s] | Factor | Type |
|---|---|---|---|---|---|---|---|
| features.0.0 | $5.93 \cdot 10^{-3}$ | $1.99 \cdot 10^{-3}$ | 2.98 x | $1.87 \cdot 10^{-3}$ | $1.97 \cdot 10^{-3}$ | **0.95 x** | Dense |
| features.1.0.block.0 | $2.53 \cdot 10^{-2}$ | $1.09 \cdot 10^{-2}$ | 2.33 x | $2.53 \cdot 10^{-2}$ | $1.09 \cdot 10^{-2}$ | 2.33 x | General |
| features.2.1 | $8.29 \cdot 10^{-3}$ | $4.53 \cdot 10^{-3}$ | 1.83 x | $4.32 \cdot 10^{-3}$ | $4.52 \cdot 10^{-3}$ | **0.96 x** | Dense |
| features.3.0.block.0 | $1.23 \cdot 10^{-2}$ | $5.85 \cdot 10^{-3}$ | 2.10 x | $1.22 \cdot 10^{-2}$ | $5.82 \cdot 10^{-3}$ | 2.10 x | General |
| features.4.1 | $5.74 \cdot 10^{-3}$ | $5.30 \cdot 10^{-3}$ | 1.08 x | $3.74 \cdot 10^{-3}$ | $5.29 \cdot 10^{-3}$ | **0.71 x** | Dense |
| features.5.0.block.0 | $6.05 \cdot 10^{-3}$ | $3.63 \cdot 10^{-3}$ | 1.66 x | $6.03 \cdot 10^{-3}$ | $3.64 \cdot 10^{-3}$ | 1.66 x | General |
| features.6.1 | $4.74 \cdot 10^{-3}$ | $5.28 \cdot 10^{-3}$ | **0.90 x** | $3.53 \cdot 10^{-3}$ | $5.17 \cdot 10^{-3}$ | **0.68 x** | Dense |
| features.7.0.block.0 | $9.08 \cdot 10^{-4}$ | $3.13 \cdot 10^{-3}$ | **0.29 x** | $8.87 \cdot 10^{-4}$ | $3.13 \cdot 10^{-3}$ | **0.28 x** | General |

(h) InceptionV3, input shape (32, 3, 299, 299)

| Name | TN [s] | PT [s] | Factor | TN + opt [s] | PT [s] | Factor | Type |
|---|---|---|---|---|---|---|---|
| Conv2d_1a_3x3.conv | $1.07 \cdot 10^{-2}$ | $1.70 \cdot 10^{-3}$ | 6.31 x | $1.07 \cdot 10^{-2}$ | $1.70 \cdot 10^{-3}$ | 6.29 x | General |
| Conv2d_2a_3x3.conv | $6.00 \cdot 10^{-2}$ | $1.16 \cdot 10^{-2}$ | 5.18 x | $3.11 \cdot 10^{-2}$ | $1.16 \cdot 10^{-2}$ | 2.68 x | General |
| Conv2d_2b_3x3.conv | $6.10 \cdot 10^{-2}$ | $1.34 \cdot 10^{-2}$ | 4.55 x | $4.27 \cdot 10^{-2}$ | $1.53 \cdot 10^{-2}$ | 2.78 x | General |
| Conv2d_3b_1x1.conv | $5.48 \cdot 10^{-3}$ | $1.82 \cdot 10^{-3}$ | 3.01 x | $2.26 \cdot 10^{-3}$ | $2.12 \cdot 10^{-3}$ | 1.07 x | Dense |
| Conv2d_4a_3x3.conv | $5.28 \cdot 10^{-2}$ | $1.29 \cdot 10^{-2}$ | 4.08 x | $3.39 \cdot 10^{-2}$ | $1.29 \cdot 10^{-2}$ | 2.62 x | General |
| Mixed_5b.branch1x1.conv | $5.14 \cdot 10^{-3}$ | $1.16 \cdot 10^{-3}$ | 4.43 x | $1.41 \cdot 10^{-3}$ | $1.48 \cdot 10^{-3}$ | **0.95 x** | Dense |
| Mixed_5b.branch5x5_1.conv | $4.92 \cdot 10^{-3}$ | $1.46 \cdot 10^{-3}$ | 3.37 x | $1.39 \cdot 10^{-3}$ | $1.47 \cdot 10^{-3}$ | **0.95 x** | Dense |
| Mixed_5b.branch5x5_2.conv | $9.28 \cdot 10^{-3}$ | $1.28 \cdot 10^{-3}$ | 7.23 x | $4.83 \cdot 10^{-3}$ | $1.28 \cdot 10^{-3}$ | 3.77 x | General |
| Mixed_5b.branch3x3dbl_2.conv | $7.78 \cdot 10^{-3}$ | $1.75 \cdot 10^{-3}$ | 4.45 x | $4.22 \cdot 10^{-3}$ | $1.75 \cdot 10^{-3}$ | 2.41 x | General |
| Mixed_5b.branch3x3dbl_3.conv | $1.05 \cdot 10^{-2}$ | $1.86 \cdot 10^{-3}$ | 5.63 x | $6.00 \cdot 10^{-3}$ | $1.87 \cdot 10^{-3}$ | 3.21 x | General |
| Mixed_5b.branch_pool.conv | $4.52 \cdot 10^{-3}$ | $9.10 \cdot 10^{-4}$ | 4.97 x | $1.16 \cdot 10^{-3}$ | $8.96 \cdot 10^{-4}$ | 1.30 x | Dense |
| Mixed_5c.branch1x1.conv | $6.55 \cdot 10^{-3}$ | $2.00 \cdot 10^{-3}$ | 3.27 x | $1.67 \cdot 10^{-3}$ | $1.93 \cdot 10^{-3}$ | **0.86 x** | Dense |
| Mixed_5c.branch5x5_1.conv | $6.33 \cdot 10^{-3}$ | $1.93 \cdot 10^{-3}$ | 3.28 x | $1.64 \cdot 10^{-3}$ | $1.86 \cdot 10^{-3}$ | **0.88 x** | Dense |
| Mixed_5d.branch1x1.conv | $7.46 \cdot 10^{-3}$ | $2.34 \cdot 10^{-3}$ | 3.19 x | $2.03 \cdot 10^{-3}$ | $2.31 \cdot 10^{-3}$ | **0.88 x** | Dense |
| Mixed_5d.branch5x5_1.conv | $7.24 \cdot 10^{-3}$ | $2.16 \cdot 10^{-3}$ | 3.36 x | $2.00 \cdot 10^{-3}$ | $2.15 \cdot 10^{-3}$ | **0.93 x** | Dense |
| Mixed_6a.branch3x3.conv | $1.10 \cdot 10^{-2}$ | $8.34 \cdot 10^{-3}$ | 1.32 x | $1.09 \cdot 10^{-2}$ | $8.34 \cdot 10^{-3}$ | 1.31 x | General |
| Mixed_6a.branch3x3dbl_3.conv | $2.46 \cdot 10^{-3}$ | $1.12 \cdot 10^{-3}$ | 2.20 x | $2.42 \cdot 10^{-3}$ | $1.12 \cdot 10^{-3}$ | 2.17 x | General |
| Mixed_6b.branch1x1.conv | $5.17 \cdot 10^{-3}$ | $2.05 \cdot 10^{-3}$ | 2.52 x | $1.86 \cdot 10^{-3}$ | $2.07 \cdot 10^{-3}$ | **0.90 x** | Dense |
| Mixed_6b.branch7x7_1.conv | $4.63 \cdot 10^{-3}$ | $1.56 \cdot 10^{-3}$ | 2.96 x | $1.46 \cdot 10^{-3}$ | $1.61 \cdot 10^{-3}$ | **0.91 x** | Dense |
| Mixed_6b.branch7x7_2.conv | $3.09 \cdot 10^{-3}$ | $1.59 \cdot 10^{-3}$ | 1.94 x | $2.01 \cdot 10^{-3}$ | $1.64 \cdot 10^{-3}$ | 1.22 x | Dense mix |
| Mixed_6b.branch7x7_3.conv | $3.44 \cdot 10^{-3}$ | $2.29 \cdot 10^{-3}$ | 1.50 x | $2.37 \cdot 10^{-3}$ | $2.33 \cdot 10^{-3}$ | 1.01 x | Dense mix |
| Mixed_6b.branch7x7dbl_2.conv | $4.06 \cdot 10^{-3}$ | $1.64 \cdot 10^{-3}$ | 2.48 x | $1.81 \cdot 10^{-3}$ | $1.68 \cdot 10^{-3}$ | 1.08 x | Dense mix |
| Mixed_6b.branch7x7dbl_5.conv | $2.51 \cdot 10^{-3}$ | $2.24 \cdot 10^{-3}$ | 1.12 x | $2.85 \cdot 10^{-3}$ | $2.29 \cdot 10^{-3}$ | 1.24 x | Dense mix |
| Mixed_6c.branch7x7_1.conv | $4.99 \cdot 10^{-3}$ | $1.98 \cdot 10^{-3}$ | 2.53 x | $1.77 \cdot 10^{-3}$ | $2.03 \cdot 10^{-3}$ | **0.87 x** | Dense |
| Mixed_6c.branch7x7_2.conv | $4.87 \cdot 10^{-3}$ | $2.71 \cdot 10^{-3}$ | 1.80 x | $3.30 \cdot 10^{-3}$ | $2.75 \cdot 10^{-3}$ | 1.20 x | Dense mix |
| Mixed_6c.branch7x7_3.conv | $4.85 \cdot 10^{-3}$ | $2.84 \cdot 10^{-3}$ | 1.71 x | $2.99 \cdot 10^{-3}$ | $2.87 \cdot 10^{-3}$ | 1.04 x | Dense mix |
| Mixed_6c.branch7x7dbl_2.conv | $5.43 \cdot 10^{-3}$ | $2.80 \cdot 10^{-3}$ | 1.94 x | $2.95 \cdot 10^{-3}$ | $2.80 \cdot 10^{-3}$ | 1.05 x | Dense mix |
| Mixed_6c.branch7x7dbl_5.conv | $3.20 \cdot 10^{-3}$ | $2.78 \cdot 10^{-3}$ | 1.15 x | $3.41 \cdot 10^{-3}$ | $2.82 \cdot 10^{-3}$ | 1.21 x | Dense mix |
| Mixed_6e.branch7x7_2.conv | $5.96 \cdot 10^{-3}$ | $3.19 \cdot 10^{-3}$ | 1.87 x | $3.83 \cdot 10^{-3}$ | $3.24 \cdot 10^{-3}$ | 1.18 x | Dense mix |
| Mixed_6e.branch7x7_3.conv | $6.50 \cdot 10^{-3}$ | $3.26 \cdot 10^{-3}$ | 1.99 x | $3.40 \cdot 10^{-3}$ | $3.30 \cdot 10^{-3}$ | 1.03 x | Dense mix |
| AuxLogits.conv0.conv | $6.48 \cdot 10^{-4}$ | $3.45 \cdot 10^{-4}$ | 1.87 x | $3.45 \cdot 10^{-4}$ | $3.88 \cdot 10^{-4}$ | **0.89 x** | Dense |
| AuxLogits.conv1.conv | $5.34 \cdot 10^{-4}$ | $2.09 \cdot 10^{-4}$ | 2.56 x | $4.59 \cdot 10^{-4}$ | $2.76 \cdot 10^{-4}$ | 1.66 x | General |
| Mixed_7a.branch3x3_2.conv | $1.80 \cdot 10^{-3}$ | $5.61 \cdot 10^{-4}$ | 3.22 x | $1.78 \cdot 10^{-3}$ | $6.16 \cdot 10^{-4}$ | 2.90 x | General |
| Mixed_7a.branch7x7x3_4.conv | $1.55 \cdot 10^{-3}$ | $8.46 \cdot 10^{-4}$ | 1.83 x | $1.52 \cdot 10^{-3}$ | $8.50 \cdot 10^{-4}$ | 1.79 x | General |
| Mixed_7b.branch1x1.conv | $2.08 \cdot 10^{-3}$ | $1.62 \cdot 10^{-3}$ | 1.28 x | $1.07 \cdot 10^{-3}$ | $1.63 \cdot 10^{-3}$ | **0.66 x** | Dense |
| Mixed_7b.branch3x3_1.conv | $2.19 \cdot 10^{-3}$ | $1.65 \cdot 10^{-3}$ | 1.33 x | $1.17 \cdot 10^{-3}$ | $1.65 \cdot 10^{-3}$ | **0.71 x** | Dense |
| Mixed_7b.branch3x3_2a.conv | $1.56 \cdot 10^{-3}$ | $1.47 \cdot 10^{-3}$ | 1.06 x | $1.20 \cdot 10^{-3}$ | $1.47 \cdot 10^{-3}$ | **0.82 x** | Dense mix |
| Mixed_7b.branch3x3_2b.conv | $1.66 \cdot 10^{-3}$ | $1.50 \cdot 10^{-3}$ | 1.11 x | $1.22 \cdot 10^{-3}$ | $1.50 \cdot 10^{-3}$ | **0.82 x** | Dense mix |
| Mixed_7b.branch3x3dbl_1.conv | $2.34 \cdot 10^{-3}$ | $1.65 \cdot 10^{-3}$ | 1.42 x | $1.33 \cdot 10^{-3}$ | $1.66 \cdot 10^{-3}$ | **0.80 x** | Dense |
| Mixed_7b.branch3x3dbl_2.conv | $3.55 \cdot 10^{-3}$ | $1.23 \cdot 10^{-3}$ | 2.90 x | $3.10 \cdot 10^{-3}$ | $1.26 \cdot 10^{-3}$ | 2.45 x | General |
| Mixed_7b.branch_pool.conv | $1.84 \cdot 10^{-3}$ | $1.46 \cdot 10^{-3}$ | 1.26 x | $8.67 \cdot 10^{-4}$ | $1.46 \cdot 10^{-3}$ | **0.59 x** | Dense |
| Mixed_7c.branch1x1.conv | $3.07 \cdot 10^{-3}$ | $3.08 \cdot 10^{-3}$ | **1.00 x** | $1.55 \cdot 10^{-3}$ | $3.12 \cdot 10^{-3}$ | **0.50 x** | Dense |
| Mixed_7c.branch3x3_1.conv | $3.30 \cdot 10^{-3}$ | $3.11 \cdot 10^{-3}$ | 1.06 x | $1.79 \cdot 10^{-3}$ | $3.11 \cdot 10^{-3}$ | **0.58 x** | Dense |
| Mixed_7c.branch3x3dbl_1.conv | $3.56 \cdot 10^{-3}$ | $3.11 \cdot 10^{-3}$ | 1.15 x | $2.03 \cdot 10^{-3}$ | $3.10 \cdot 10^{-3}$ | **0.65 x** | Dense |
| Mixed_7c.branch_pool.conv | $2.70 \cdot 10^{-3}$ | $1.61 \cdot 10^{-3}$ | 1.68 x | $1.22 \cdot 10^{-3}$ | $1.61 \cdot 10^{-3}$ | **0.76 x** | Dense |

(i) MobileNetV2, input shape (32, 3, 256, 256)

| Name | TN [s] | PT [s] | Factor | TN + opt [s] | PT [s] | Factor | Type |
|---|---|---|---|---|---|---|---|
| features.0.0 | $7.70 \cdot 10^{-3}$ | $1.45 \cdot 10^{-3}$ | 5.30 x | $7.67 \cdot 10^{-3}$ | $1.46 \cdot 10^{-3}$ | 5.26 x | General |
| features.1.conv.0.0 | $1.59 \cdot 10^{-2}$ | $2.46 \cdot 10^{-3}$ | 6.48 x | $1.59 \cdot 10^{-2}$ | $2.47 \cdot 10^{-3}$ | 6.46 x | General |
| features.1.conv.1 | $1.47 \cdot 10^{-2}$ | $2.40 \cdot 10^{-3}$ | 6.11 x | $2.64 \cdot 10^{-3}$ | $2.39 \cdot 10^{-3}$ | 1.10 x | Dense |
| features.2.conv.0.0 | $8.97 \cdot 10^{-3}$ | $4.95 \cdot 10^{-3}$ | 1.81 x | $4.22 \cdot 10^{-3}$ | $4.95 \cdot 10^{-3}$ | **0.85 x** | Dense |
| features.2.conv.1.0 | $2.14 \cdot 10^{-2}$ | $2.40 \cdot 10^{-3}$ | 8.90 x | $2.14 \cdot 10^{-2}$ | $2.41 \cdot 10^{-3}$ | 8.89 x | General |
| features.2.conv.2 | $7.38 \cdot 10^{-3}$ | $1.96 \cdot 10^{-3}$ | 3.76 x | $1.91 \cdot 10^{-3}$ | $1.89 \cdot 10^{-3}$ | 1.01 x | Dense |
| features.3.conv.0.0 | $3.55 \cdot 10^{-3}$ | $2.24 \cdot 10^{-3}$ | 1.58 x | $1.83 \cdot 10^{-3}$ | $2.19 \cdot 10^{-3}$ | **0.84 x** | Dense |
| features.3.conv.1.0 | $1.34 \cdot 10^{-2}$ | $2.49 \cdot 10^{-3}$ | 5.38 x | $1.34 \cdot 10^{-2}$ | $2.46 \cdot 10^{-3}$ | 5.43 x | General |
| features.3.conv.2 | $1.04 \cdot 10^{-2}$ | $2.89 \cdot 10^{-3}$ | 3.61 x | $2.89 \cdot 10^{-3}$ | $2.85 \cdot 10^{-3}$ | 1.01 x | Dense |
| features.4.conv.1.0 | $7.46 \cdot 10^{-3}$ | $1.01 \cdot 10^{-3}$ | 7.41 x | $7.44 \cdot 10^{-3}$ | $9.84 \cdot 10^{-4}$ | 7.56 x | General |
| features.4.conv.2 | $3.00 \cdot 10^{-3}$ | $9.40 \cdot 10^{-4}$ | 3.19 x | $9.20 \cdot 10^{-4}$ | $8.94 \cdot 10^{-4}$ | 1.03 x | Dense |
| features.5.conv.0.0 | $1.49 \cdot 10^{-3}$ | $7.46 \cdot 10^{-4}$ | 2.00 x | $7.48 \cdot 10^{-4}$ | $7.46 \cdot 10^{-4}$ | 1.00 x | Dense |
| features.5.conv.1.0 | $4.77 \cdot 10^{-3}$ | $9.10 \cdot 10^{-4}$ | 5.24 x | $4.75 \cdot 10^{-3}$ | $8.88 \cdot 10^{-4}$ | 5.34 x | General |
| features.5.conv.2 | $3.62 \cdot 10^{-3}$ | $1.06 \cdot 10^{-3}$ | 3.41 x | $1.04 \cdot 10^{-3}$ | $1.01 \cdot 10^{-3}$ | 1.03 x | Dense |
| features.7.conv.1.0 | $2.61 \cdot 10^{-3}$ | $3.66 \cdot 10^{-4}$ | 7.13 x | $2.60 \cdot 10^{-3}$ | $3.65 \cdot 10^{-4}$ | 7.13 x | General |
| features.7.conv.2 | $1.43 \cdot 10^{-3}$ | $5.55 \cdot 10^{-4}$ | 2.58 x | $4.64 \cdot 10^{-4}$ | $5.55 \cdot 10^{-4}$ | **0.84 x** | Dense |
| features.8.conv.0.0 | $1.14 \cdot 10^{-3}$ | $5.36 \cdot 10^{-4}$ | 2.12 x | $5.22 \cdot 10^{-4}$ | $5.34 \cdot 10^{-4}$ | **0.98 x** | Dense |
| features.8.conv.1.0 | $2.44 \cdot 10^{-3}$ | $5.67 \cdot 10^{-4}$ | 4.31 x | $2.43 \cdot 10^{-3}$ | $5.68 \cdot 10^{-4}$ | 4.28 x | General |
| features.8.conv.2 | $2.23 \cdot 10^{-3}$ | $8.32 \cdot 10^{-4}$ | 2.68 x | $6.80 \cdot 10^{-4}$ | $8.82 \cdot 10^{-4}$ | **0.77 x** | Dense |
| features.11.conv.2 | $2.36 \cdot 10^{-3}$ | $8.68 \cdot 10^{-4}$ | 2.72 x | $7.89 \cdot 10^{-4}$ | $8.69 \cdot 10^{-4}$ | **0.91 x** | Dense |
| features.12.conv.0.0 | $1.55 \cdot 10^{-3}$ | $1.08 \cdot 10^{-3}$ | 1.44 x | $9.00 \cdot 10^{-4}$ | $1.03 \cdot 10^{-3}$ | **0.88 x** | Dense |
| features.12.conv.1.0 | $3.52 \cdot 10^{-3}$ | $8.20 \cdot 10^{-4}$ | 4.29 x | $3.50 \cdot 10^{-3}$ | $8.19 \cdot 10^{-4}$ | 4.27 x | General |
| features.12.conv.2 | $3.27 \cdot 10^{-3}$ | $1.26 \cdot 10^{-3}$ | 2.59 x | $1.10 \cdot 10^{-3}$ | $1.26 \cdot 10^{-3}$ | **0.87 x** | Dense |
| features.14.conv.1.0 | $2.07 \cdot 10^{-3}$ | $3.90 \cdot 10^{-4}$ | 5.31 x | $2.05 \cdot 10^{-3}$ | $3.90 \cdot 10^{-4}$ | 5.26 x | General |
| features.14.conv.2 | $1.06 \cdot 10^{-3}$ | $1.39 \cdot 10^{-3}$ | **0.76 x** | $5.10 \cdot 10^{-4}$ | $1.40 \cdot 10^{-3}$ | **0.36 x** | Dense |
| features.15.conv.0.0 | $1.12 \cdot 10^{-3}$ | $7.19 \cdot 10^{-4}$ | 1.56 x | $6.21 \cdot 10^{-4}$ | $7.10 \cdot 10^{-4}$ | **0.87 x** | Dense |
| features.15.conv.1.0 | $2.31 \cdot 10^{-3}$ | $5.96 \cdot 10^{-4}$ | 3.87 x | $2.28 \cdot 10^{-3}$ | $5.96 \cdot 10^{-4}$ | 3.83 x | General |
| features.15.conv.2 | $1.34 \cdot 10^{-3}$ | $1.41 \cdot 10^{-3}$ | **0.95 x** | $6.53 \cdot 10^{-4}$ | $1.40 \cdot 10^{-3}$ | **0.47 x** | Dense |
| features.17.conv.2 | $1.59 \cdot 10^{-3}$ | $1.67 \cdot 10^{-3}$ | **0.95 x** | $8.70 \cdot 10^{-4}$ | $1.62 \cdot 10^{-3}$ | **0.54 x** | Dense |
| features.18.0 | $1.53 \cdot 10^{-3}$ | $1.68 \cdot 10^{-3}$ | **0.91 x** | $1.04 \cdot 10^{-3}$ | $1.63 \cdot 10^{-3}$ | **0.64 x** | Dense |

### F.5 KFC FACTOR (KFAC-EXPAND)

We compare TN and TN+opt with a PyTorch implementation of the input-based KFC factor based on `torch.nn.functional.unfold`. Figure F21 visualizes the performance ratios for different convolution categories. Table F7 contains the detailed run times and performance factors.

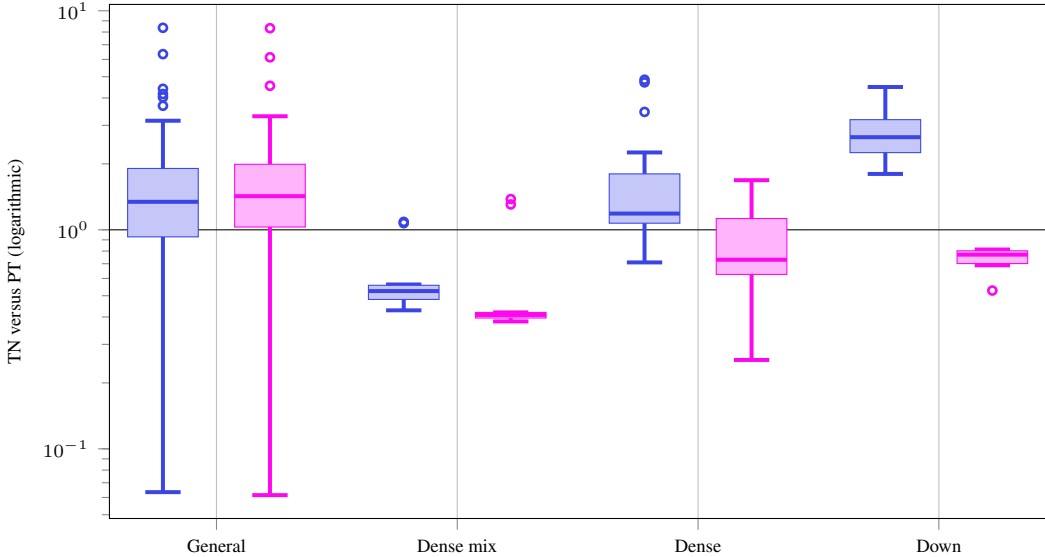

Figure F21: KFC/KFAC-expand factor performance ratios of TN versus PT and TN+opt versus PT for different convolution types on GPU.

Table F7: KFC (KFAC-expand) factor performance comparison on GPU.

(a) 3c3d, CIFAR-10, input shape (128, 3, 32, 32)

| Name | TN [s] | PT [s] | Factor | TN + opt [s] | PT [s] | Factor | Type |
|---|---|---|---|---|---|---|---|
| conv1.0 | $1.03 \cdot 10^{-3}$ | $2.42 \cdot 10^{-3}$ | **0.43 x** | $1.03 \cdot 10^{-3}$ | $2.52 \cdot 10^{-3}$ | **0.41 x** | General |
| conv2.0 | $6.69 \cdot 10^{-3}$ | $3.83 \cdot 10^{-3}$ | 1.75 x | $6.97 \cdot 10^{-3}$ | $4.52 \cdot 10^{-3}$ | 1.54 x | General |
| conv3.1 | $3.27 \cdot 10^{-3}$ | $2.38 \cdot 10^{-3}$ | 1.37 x | $3.53 \cdot 10^{-3}$ | $2.54 \cdot 10^{-3}$ | 1.39 x | General |

(b) F-MNIST 2c2d, input shape (128, 1, 28, 28)

| Name | TN [s] | PT [s] | Factor | TN + opt [s] | PT [s] | Factor | Type |
|---|---|---|---|---|---|---|---|
| conv1.1 | $1.22 \cdot 10^{-3}$ | $2.01 \cdot 10^{-3}$ | **0.61 x** | $9.30 \cdot 10^{-4}$ | $1.72 \cdot 10^{-3}$ | **0.54 x** | General |
| conv2.1 | $1.03 \cdot 10^{-2}$ | $9.54 \cdot 10^{-3}$ | 1.08 x | $1.02 \cdot 10^{-2}$ | $9.47 \cdot 10^{-3}$ | 1.08 x | General |

(c) CIFAR-100 All-CNN-C, input shape (128, 3, 32, 32)

| Name | TN [s] | PT [s] | Factor | TN + opt [s] | PT [s] | Factor | Type |
|---|---|---|---|---|---|---|---|
| conv1.1 | $1.37 \cdot 10^{-3}$ | $1.48 \cdot 10^{-3}$ | **0.93 x** | $2.72 \cdot 10^{-3}$ | $2.11 \cdot 10^{-3}$ | 1.29 x | General |
| conv2.1 | $1.48 \cdot 10^{-1}$ | $6.95 \cdot 10^{-2}$ | 2.13 x | $1.53 \cdot 10^{-1}$ | $7.14 \cdot 10^{-2}$ | 2.15 x | General |
| conv3.1 | $4.77 \cdot 10^{-2}$ | $1.15 \cdot 10^{-2}$ | 4.17 x | $4.56 \cdot 10^{-2}$ | $1.38 \cdot 10^{-2}$ | 3.30 x | General |
| conv4.1 | $2.32 \cdot 10^{-2}$ | $1.14 \cdot 10^{-2}$ | 2.03 x | $2.25 \cdot 10^{-2}$ | $1.14 \cdot 10^{-2}$ | 1.98 x | General |
| conv5.1 | $7.03 \cdot 10^{-2}$ | $5.82 \cdot 10^{-2}$ | 1.21 x | $1.01 \cdot 10^{-1}$ | $6.19 \cdot 10^{-2}$ | 1.63 x | General |
| conv6.1 | $2.84 \cdot 10^{-2}$ | $1.33 \cdot 10^{-2}$ | 2.14 x | $2.83 \cdot 10^{-2}$ | $9.77 \cdot 10^{-3}$ | 2.90 x | General |
| conv7.0 | $8.68 \cdot 10^{-3}$ | $5.95 \cdot 10^{-3}$ | 1.46 x | $9.30 \cdot 10^{-3}$ | $6.01 \cdot 10^{-3}$ | 1.55 x | General |
| conv8.1 | $1.03 \cdot 10^{-3}$ | $9.97 \cdot 10^{-4}$ | 1.03 x | $3.67 \cdot 10^{-4}$ | $1.44 \cdot 10^{-3}$ | **0.25 x** | Dense |
| conv9.1 | $1.06 \cdot 10^{-3}$ | $1.49 \cdot 10^{-3}$ | **0.71 x** | $4.61 \cdot 10^{-4}$ | $1.56 \cdot 10^{-3}$ | **0.30 x** | Dense |

(d) Alexnet, input shape (32, 3, 256, 256)

| Name | TN [s] | PT [s] | Factor | TN + opt [s] | PT [s] | Factor | Type |
|---|---|---|---|---|---|---|---|
| features.0 | $5.45 \cdot 10^{-2}$ | $1.35 \cdot 10^{-2}$ | 4.03 x | $6.09 \cdot 10^{-2}$ | $1.34 \cdot 10^{-2}$ | 4.55 x | General |
| features.3 | $4.57 \cdot 10^{-2}$ | $4.14 \cdot 10^{-2}$ | 1.10 x | $5.31 \cdot 10^{-2}$ | $3.73 \cdot 10^{-2}$ | 1.42 x | General |
| features.6 | $8.63 \cdot 10^{-3}$ | $7.86 \cdot 10^{-3}$ | 1.10 x | $1.12 \cdot 10^{-2}$ | $8.74 \cdot 10^{-3}$ | 1.28 x | General |
| features.8 | $3.76 \cdot 10^{-2}$ | $4.10 \cdot 10^{-2}$ | **0.92 x** | $4.57 \cdot 10^{-2}$ | $4.33 \cdot 10^{-2}$ | 1.06 x | General |
| features.10 | $1.52 \cdot 10^{-2}$ | $1.38 \cdot 10^{-2}$ | 1.10 x | $1.91 \cdot 10^{-2}$ | $1.47 \cdot 10^{-2}$ | 1.30 x | General |

(e) ResNet18, input shape (32, 3, 256, 256)

| Name | TN [s] | PT [s] | Factor | TN + opt [s] | PT [s] | Factor | Type |
|---|---|---|---|---|---|---|---|
| conv1 | $5.25 \cdot 10^{-2}$ | $2.07 \cdot 10^{-2}$ | 2.54 x | $5.22 \cdot 10^{-2}$ | $2.06 \cdot 10^{-2}$ | 2.54 x | General |
| layer1.0.conv1 | $4.36 \cdot 10^{-2}$ | $2.98 \cdot 10^{-2}$ | 1.46 x | $5.57 \cdot 10^{-2}$ | $3.02 \cdot 10^{-2}$ | 1.84 x | General |
| layer2.0.conv1 | $2.81 \cdot 10^{-2}$ | $6.38 \cdot 10^{-3}$ | 4.41 x | $2.78 \cdot 10^{-2}$ | $1.23 \cdot 10^{-2}$ | 2.25 x | General |
| layer2.0.conv2 | $2.56 \cdot 10^{-2}$ | $1.90 \cdot 10^{-2}$ | 1.34 x | $3.09 \cdot 10^{-2}$ | $2.01 \cdot 10^{-2}$ | 1.53 x | General |
| layer2.0.downsample.0 | $3.66 \cdot 10^{-3}$ | $8.14 \cdot 10^{-4}$ | 4.49 x | $6.45 \cdot 10^{-4}$ | $7.94 \cdot 10^{-4}$ | **0.81 x** | Down |
| layer3.0.conv1 | $1.34 \cdot 10^{-2}$ | $9.19 \cdot 10^{-3}$ | 1.46 x | $1.40 \cdot 10^{-2}$ | $9.17 \cdot 10^{-3}$ | 1.53 x | General |
| layer3.0.conv2 | $1.90 \cdot 10^{-2}$ | $1.84 \cdot 10^{-2}$ | 1.03 x | $2.25 \cdot 10^{-2}$ | $1.95 \cdot 10^{-2}$ | 1.16 x | General |
| layer3.0.downsample.0 | $1.98 \cdot 10^{-3}$ | $7.00 \cdot 10^{-4}$ | 2.83 x | $4.59 \cdot 10^{-4}$ | $5.72 \cdot 10^{-4}$ | **0.80 x** | Down |
| layer4.0.conv1 | $8.65 \cdot 10^{-3}$ | $4.79 \cdot 10^{-3}$ | 1.81 x | $9.12 \cdot 10^{-3}$ | $4.60 \cdot 10^{-3}$ | 1.98 x | General |
| layer4.0.conv2 | $2.48 \cdot 10^{-2}$ | $1.63 \cdot 10^{-2}$ | 1.52 x | $2.49 \cdot 10^{-2}$ | $1.88 \cdot 10^{-2}$ | 1.32 x | General |
| layer4.0.downsample.0 | $1.19 \cdot 10^{-3}$ | $5.45 \cdot 10^{-4}$ | 2.18 x | $2.88 \cdot 10^{-4}$ | $5.45 \cdot 10^{-4}$ | **0.53 x** | Down |

(f) ResNext101, input shape (32, 3, 256, 256)

| Name | TN [s] | PT [s] | Factor | TN + opt [s] | PT [s] | Factor | Type |
|---|---|---|---|---|---|---|---|
| conv1 | $5.13 \cdot 10^{-2}$ | $2.05 \cdot 10^{-2}$ | 2.50 x | $5.06 \cdot 10^{-2}$ | $2.05 \cdot 10^{-2}$ | 2.46 x | General |
| layer1.0.conv1 | $3.33 \cdot 10^{-3}$ | $1.85 \cdot 10^{-3}$ | 1.80 x | $1.70 \cdot 10^{-3}$ | $2.08 \cdot 10^{-3}$ | **0.82 x** | Dense |
| layer1.0.conv2 | $1.09 \cdot 10^{-1}$ | $6.60 \cdot 10^{-2}$ | 1.66 x | $1.11 \cdot 10^{-1}$ | $8.12 \cdot 10^{-2}$ | 1.37 x | General |
| layer1.0.conv3 | $1.60 \cdot 10^{-2}$ | $7.49 \cdot 10^{-3}$ | 2.14 x | $1.04 \cdot 10^{-2}$ | $7.52 \cdot 10^{-3}$ | 1.39 x | Dense |
| layer2.0.conv1 | $1.60 \cdot 10^{-2}$ | $1.53 \cdot 10^{-2}$ | 1.05 x | $1.04 \cdot 10^{-2}$ | $1.53 \cdot 10^{-2}$ | **0.68 x** | Dense |
| layer2.0.conv2 | $1.40 \cdot 10^{-1}$ | $4.44 \cdot 10^{-2}$ | 3.15 x | $1.44 \cdot 10^{-1}$ | $4.54 \cdot 10^{-2}$ | 3.18 x | General |
| layer2.0.conv3 | $1.14 \cdot 10^{-2}$ | $5.19 \cdot 10^{-3}$ | 2.20 x | $8.41 \cdot 10^{-3}$ | $5.20 \cdot 10^{-3}$ | 1.62 x | Dense |
| layer2.0.downsample.0 | $1.40 \cdot 10^{-2}$ | $4.22 \cdot 10^{-3}$ | 3.30 x | $2.92 \cdot 10^{-3}$ | $4.24 \cdot 10^{-3}$ | **0.69 x** | Down |
| layer2.1.conv2 | $5.07 \cdot 10^{-2}$ | $4.23 \cdot 10^{-2}$ | 1.20 x | $5.07 \cdot 10^{-2}$ | $4.24 \cdot 10^{-2}$ | 1.19 x | General |
| layer3.0.conv1 | $1.14 \cdot 10^{-2}$ | $5.21 \cdot 10^{-3}$ | 2.19 x | $8.42 \cdot 10^{-3}$ | $5.27 \cdot 10^{-3}$ | 1.60 x | Dense |
| layer3.0.conv2 | $6.11 \cdot 10^{-2}$ | $3.21 \cdot 10^{-2}$ | 1.90 x | $6.23 \cdot 10^{-2}$ | $2.92 \cdot 10^{-2}$ | 2.14 x | General |
| layer3.0.conv3 | $8.77 \cdot 10^{-3}$ | $4.30 \cdot 10^{-3}$ | 2.04 x | $7.17 \cdot 10^{-3}$ | $4.33 \cdot 10^{-3}$ | 1.66 x | Dense |
| layer3.0.downsample.0 | $7.59 \cdot 10^{-3}$ | $3.08 \cdot 10^{-3}$ | 2.47 x | $2.28 \cdot 10^{-3}$ | $3.08 \cdot 10^{-3}$ | **0.74 x** | Down |
| layer3.1.conv2 | $2.12 \cdot 10^{-2}$ | $1.95 \cdot 10^{-2}$ | 1.09 x | $2.05 \cdot 10^{-2}$ | $1.99 \cdot 10^{-2}$ | 1.03 x | General |
| layer4.0.conv1 | $8.75 \cdot 10^{-3}$ | $4.15 \cdot 10^{-3}$ | 2.11 x | $7.18 \cdot 10^{-3}$ | $4.26 \cdot 10^{-3}$ | 1.68 x | Dense |
| layer4.0.conv2 | $4.70 \cdot 10^{-2}$ | $2.47 \cdot 10^{-2}$ | 1.91 x | $4.71 \cdot 10^{-2}$ | $2.47 \cdot 10^{-2}$ | 1.91 x | General |
| layer4.0.conv3 | $7.88 \cdot 10^{-3}$ | $7.66 \cdot 10^{-3}$ | 1.03 x | $6.74 \cdot 10^{-3}$ | $7.67 \cdot 10^{-3}$ | **0.88 x** | Dense |
| layer4.0.downsample.0 | $4.54 \cdot 10^{-3}$ | $2.52 \cdot 10^{-3}$ | 1.80 x | $2.03 \cdot 10^{-3}$ | $2.54 \cdot 10^{-3}$ | **0.80 x** | Down |
| layer4.1.conv2 | $1.36 \cdot 10^{-2}$ | $1.16 \cdot 10^{-2}$ | 1.16 x | $1.36 \cdot 10^{-2}$ | $1.17 \cdot 10^{-2}$ | 1.16 x | General |

(g) ConvNeXt-base, input shape (32, 3, 256, 256)

| Name | TN [s] | PT [s] | Factor | TN + opt [s] | PT [s] | Factor | Type |
|---|---|---|---|---|---|---|---|
| features.0.0 | $9.94 \cdot 10^{-3}$ | $2.11 \cdot 10^{-3}$ | 4.71 x | $1.18 \cdot 10^{-3}$ | $2.11 \cdot 10^{-3}$ | **0.56 x** | Dense |
| features.1.0.block.0 | $4.09 \cdot 10^{-2}$ | $1.37 \cdot 10^{-1}$ | **0.30 x** | $5.25 \cdot 10^{-2}$ | $1.42 \cdot 10^{-1}$ | **0.37 x** | General |
| features.2.1 | $2.37 \cdot 10^{-2}$ | $4.90 \cdot 10^{-3}$ | 4.85 x | $7.81 \cdot 10^{-3}$ | $4.93 \cdot 10^{-3}$ | 1.59 x | Dense |
| features.3.0.block.0 | $1.61 \cdot 10^{-2}$ | $6.99 \cdot 10^{-2}$ | **0.23 x** | $1.57 \cdot 10^{-2}$ | $7.12 \cdot 10^{-2}$ | **0.22 x** | General |
| features.4.1 | $1.41 \cdot 10^{-2}$ | $4.08 \cdot 10^{-3}$ | 3.45 x | $6.88 \cdot 10^{-3}$ | $4.15 \cdot 10^{-3}$ | 1.66 x | Dense |
| features.5.0.block.0 | $3.98 \cdot 10^{-3}$ | $3.35 \cdot 10^{-2}$ | **0.12 x** | $3.96 \cdot 10^{-3}$ | $3.43 \cdot 10^{-2}$ | **0.12 x** | General |
| features.6.1 | $6.82 \cdot 10^{-3}$ | $3.30 \cdot 10^{-3}$ | 2.06 x | $4.77 \cdot 10^{-3}$ | $3.31 \cdot 10^{-3}$ | 1.44 x | Dense |
| features.7.0.block.0 | $1.02 \cdot 10^{-3}$ | $1.61 \cdot 10^{-2}$ | **0.06 x** | $1.00 \cdot 10^{-3}$ | $1.63 \cdot 10^{-2}$ | **0.06 x** | General |

(h) InceptionV3, input shape (32, 3, 299, 299)

| Name | TN [s] | PT [s] | Factor | TN + opt [s] | PT [s] | Factor | Type |
|---|---|---|---|---|---|---|---|
| Conv2d_1a_3x3.conv | $3.42 \cdot 10^{-2}$ | $4.09 \cdot 10^{-3}$ | 8.36 x | $3.43 \cdot 10^{-2}$ | $4.11 \cdot 10^{-3}$ | 8.33 x | General |
| Conv2d_2a_3x3.conv | $1.58 \cdot 10^{-1}$ | $9.29 \cdot 10^{-2}$ | 1.70 x | $1.91 \cdot 10^{-1}$ | $9.18 \cdot 10^{-2}$ | 2.08 x | General |
| Conv2d_2b_3x3.conv | $1.56 \cdot 10^{-1}$ | $9.66 \cdot 10^{-2}$ | 1.61 x | $1.88 \cdot 10^{-1}$ | $9.44 \cdot 10^{-2}$ | 1.99 x | General |
| Conv2d_3b_1x1.conv | $5.26 \cdot 10^{-3}$ | $2.33 \cdot 10^{-3}$ | 2.26 x | $1.74 \cdot 10^{-3}$ | $2.41 \cdot 10^{-3}$ | **0.72 x** | Dense |
| Conv2d_4a_3x3.conv | $1.01 \cdot 10^{-1}$ | $6.34 \cdot 10^{-2}$ | 1.58 x | $1.08 \cdot 10^{-1}$ | $6.16 \cdot 10^{-2}$ | 1.76 x | General |
| Mixed_5b.branch1x1.conv | $4.13 \cdot 10^{-3}$ | $2.02 \cdot 10^{-3}$ | 2.05 x | $2.40 \cdot 10^{-3}$ | $2.13 \cdot 10^{-3}$ | 1.13 x | Dense |
| Mixed_5b.branch5x5_1.conv | $4.12 \cdot 10^{-3}$ | $3.66 \cdot 10^{-3}$ | 1.13 x | $2.41 \cdot 10^{-3}$ | $3.65 \cdot 10^{-3}$ | **0.66 x** | Dense |
| Mixed_5b.branch5x5_2.conv | $4.29 \cdot 10^{-2}$ | $3.27 \cdot 10^{-2}$ | 1.31 x | $4.39 \cdot 10^{-2}$ | $3.29 \cdot 10^{-2}$ | 1.33 x | General |
| Mixed_5b.branch3x3dbl_2.conv | $8.57 \cdot 10^{-3}$ | $7.27 \cdot 10^{-3}$ | 1.18 x | $1.38 \cdot 10^{-2}$ | $7.31 \cdot 10^{-3}$ | 1.89 x | General |
| Mixed_5b.branch3x3dbl_3.conv | $1.72 \cdot 10^{-2}$ | $1.42 \cdot 10^{-2}$ | 1.21 x | $2.46 \cdot 10^{-2}$ | $1.42 \cdot 10^{-2}$ | 1.73 x | General |
| Mixed_5b.branch_pool.conv | $4.12 \cdot 10^{-3}$ | $2.05 \cdot 10^{-3}$ | 2.01 x | $2.40 \cdot 10^{-3}$ | $2.02 \cdot 10^{-3}$ | 1.19 x | Dense |
| Mixed_5c.branch1x1.conv | $5.43 \cdot 10^{-3}$ | $4.89 \cdot 10^{-3}$ | 1.11 x | $3.27 \cdot 10^{-3}$ | $4.89 \cdot 10^{-3}$ | **0.67 x** | Dense |
| Mixed_5c.branch5x5_1.conv | $5.40 \cdot 10^{-3}$ | $4.87 \cdot 10^{-3}$ | 1.11 x | $3.27 \cdot 10^{-3}$ | $4.88 \cdot 10^{-3}$ | **0.67 x** | Dense |
| Mixed_5d.branch1x1.conv | $7.24 \cdot 10^{-3}$ | $6.66 \cdot 10^{-3}$ | 1.09 x | $4.88 \cdot 10^{-3}$ | $6.68 \cdot 10^{-3}$ | **0.73 x** | Dense |
| Mixed_5d.branch5x5_1.conv | $7.25 \cdot 10^{-3}$ | $6.67 \cdot 10^{-3}$ | 1.09 x | $4.88 \cdot 10^{-3}$ | $6.69 \cdot 10^{-3}$ | **0.73 x** | Dense |
| Mixed_6a.branch3x3.conv | $7.76 \cdot 10^{-2}$ | $3.28 \cdot 10^{-2}$ | 2.37 x | $7.78 \cdot 10^{-2}$ | $3.23 \cdot 10^{-2}$ | 2.41 x | General |
| Mixed_6a.branch3x3dbl_3.conv | $1.29 \cdot 10^{-2}$ | $3.50 \cdot 10^{-3}$ | 3.69 x | $1.41 \cdot 10^{-2}$ | $7.15 \cdot 10^{-3}$ | 1.97 x | General |
| Mixed_6b.branch1x1.conv | $6.56 \cdot 10^{-3}$ | $5.66 \cdot 10^{-3}$ | 1.16 x | $4.80 \cdot 10^{-3}$ | $4.22 \cdot 10^{-3}$ | 1.14 x | Dense |
| Mixed_6b.branch7x7_1.conv | $6.55 \cdot 10^{-3}$ | $6.02 \cdot 10^{-3}$ | 1.09 x | $4.80 \cdot 10^{-3}$ | $6.03 \cdot 10^{-3}$ | **0.80 x** | Dense |
| Mixed_6b.branch7x7_2.conv | $2.01 \cdot 10^{-3}$ | $3.60 \cdot 10^{-3}$ | **0.56 x** | $1.50 \cdot 10^{-3}$ | $3.58 \cdot 10^{-3}$ | **0.42 x** | Dense mix |
| Mixed_6b.branch7x7_3.conv | $1.92 \cdot 10^{-3}$ | $3.50 \cdot 10^{-3}$ | **0.55 x** | $1.46 \cdot 10^{-3}$ | $3.58 \cdot 10^{-3}$ | **0.41 x** | Dense mix |
| Mixed_6b.branch7x7dbl_2.conv | $1.94 \cdot 10^{-3}$ | $3.54 \cdot 10^{-3}$ | **0.55 x** | $1.45 \cdot 10^{-3}$ | $3.56 \cdot 10^{-3}$ | **0.41 x** | Dense mix |
| Mixed_6b.branch7x7dbl_5.conv | $1.97 \cdot 10^{-3}$ | $3.49 \cdot 10^{-3}$ | **0.56 x** | $1.46 \cdot 10^{-3}$ | $3.49 \cdot 10^{-3}$ | **0.42 x** | Dense mix |
| Mixed_6c.branch7x7_1.conv | $6.59 \cdot 10^{-3}$ | $4.60 \cdot 10^{-3}$ | 1.43 x | $4.80 \cdot 10^{-3}$ | $4.90 \cdot 10^{-3}$ | **0.98 x** | Dense |
| Mixed_6c.branch7x7_2.conv | $2.59 \cdot 10^{-3}$ | $5.14 \cdot 10^{-3}$ | **0.50 x** | $2.08 \cdot 10^{-3}$ | $5.08 \cdot 10^{-3}$ | **0.41 x** | Dense mix |
| Mixed_6c.branch7x7_3.conv | $2.58 \cdot 10^{-3}$ | $5.32 \cdot 10^{-3}$ | **0.48 x** | $2.04 \cdot 10^{-3}$ | $5.23 \cdot 10^{-3}$ | **0.39 x** | Dense mix |
| Mixed_6c.branch7x7dbl_2.conv | $2.51 \cdot 10^{-3}$ | $5.32 \cdot 10^{-3}$ | **0.47 x** | $2.05 \cdot 10^{-3}$ | $5.25 \cdot 10^{-3}$ | **0.39 x** | Dense mix |
| Mixed_6c.branch7x7dbl_5.conv | $2.53 \cdot 10^{-3}$ | $5.21 \cdot 10^{-3}$ | **0.49 x** | $2.04 \cdot 10^{-3}$ | $5.12 \cdot 10^{-3}$ | **0.40 x** | Dense mix |
| Mixed_6e.branch7x7_2.conv | $3.35 \cdot 10^{-3}$ | $7.81 \cdot 10^{-3}$ | **0.43 x** | $2.90 \cdot 10^{-3}$ | $7.61 \cdot 10^{-3}$ | **0.38 x** | Dense mix |
| Mixed_6e.branch7x7_3.conv | $3.35 \cdot 10^{-3}$ | $7.52 \cdot 10^{-3}$ | **0.45 x** | $2.91 \cdot 10^{-3}$ | $7.31 \cdot 10^{-3}$ | **0.40 x** | Dense mix |
| AuxLogits.conv0.conv | $1.09 \cdot 10^{-3}$ | $6.14 \cdot 10^{-4}$ | 1.77 x | $3.82 \cdot 10^{-4}$ | $6.10 \cdot 10^{-4}$ | **0.63 x** | Dense |
| AuxLogits.conv1.conv | $8.95 \cdot 10^{-4}$ | $1.07 \cdot 10^{-3}$ | **0.84 x** | $8.52 \cdot 10^{-4}$ | $1.09 \cdot 10^{-3}$ | **0.78 x** | General |
| Mixed_7a.branch3x3_2.conv | $6.56 \cdot 10^{-3}$ | $2.67 \cdot 10^{-3}$ | 2.45 x | $6.98 \cdot 10^{-3}$ | $2.68 \cdot 10^{-3}$ | 2.60 x | General |
| Mixed_7a.branch7x7x3_4.conv | $6.93 \cdot 10^{-3}$ | $2.93 \cdot 10^{-3}$ | 2.36 x | $7.04 \cdot 10^{-3}$ | $2.94 \cdot 10^{-3}$ | 2.39 x | General |
| Mixed_7b.branch1x1.conv | $3.27 \cdot 10^{-3}$ | $1.82 \cdot 10^{-3}$ | 1.80 x | $2.39 \cdot 10^{-3}$ | $1.76 \cdot 10^{-3}$ | 1.36 x | Dense |
| Mixed_7b.branch3x3_1.conv | $3.66 \cdot 10^{-3}$ | $3.34 \cdot 10^{-3}$ | 1.10 x | $2.83 \cdot 10^{-3}$ | $3.36 \cdot 10^{-3}$ | **0.84 x** | Dense |
| Mixed_7b.branch3x3_2a.conv | $2.51 \cdot 10^{-3}$ | $2.34 \cdot 10^{-3}$ | 1.07 x | $3.03 \cdot 10^{-3}$ | $2.32 \cdot 10^{-3}$ | 1.31 x | Dense mix |
| Mixed_7b.branch3x3_2b.conv | $2.43 \cdot 10^{-3}$ | $2.24 \cdot 10^{-3}$ | 1.09 x | $2.98 \cdot 10^{-3}$ | $2.16 \cdot 10^{-3}$ | 1.38 x | Dense mix |
| Mixed_7b.branch3x3dbl_1.conv | $3.70 \cdot 10^{-3}$ | $2.57 \cdot 10^{-3}$ | 1.44 x | $2.83 \cdot 10^{-3}$ | $2.43 \cdot 10^{-3}$ | 1.17 x | Dense |
| Mixed_7b.branch3x3dbl_2.conv | $2.03 \cdot 10^{-2}$ | $1.45 \cdot 10^{-2}$ | 1.40 x | $1.94 \cdot 10^{-2}$ | $1.40 \cdot 10^{-2}$ | 1.39 x | General |
| Mixed_7b.branch_pool.conv | $2.89 \cdot 10^{-3}$ | $1.57 \cdot 10^{-3}$ | 1.84 x | $2.26 \cdot 10^{-3}$ | $1.57 \cdot 10^{-3}$ | 1.44 x | Dense |
| Mixed_7c.branch1x1.conv | $7.88 \cdot 10^{-3}$ | $7.66 \cdot 10^{-3}$ | 1.03 x | $6.73 \cdot 10^{-3}$ | $7.66 \cdot 10^{-3}$ | **0.88 x** | Dense |
| Mixed_7c.branch3x3_1.conv | $7.88 \cdot 10^{-3}$ | $7.66 \cdot 10^{-3}$ | 1.03 x | $6.73 \cdot 10^{-3}$ | $7.66 \cdot 10^{-3}$ | **0.88 x** | Dense |
| Mixed_7c.branch3x3dbl_1.conv | $7.92 \cdot 10^{-3}$ | $7.67 \cdot 10^{-3}$ | 1.03 x | $6.74 \cdot 10^{-3}$ | $7.67 \cdot 10^{-3}$ | **0.88 x** | Dense |
| Mixed_7c.branch_pool.conv | $7.92 \cdot 10^{-3}$ | $7.67 \cdot 10^{-3}$ | 1.03 x | $6.74 \cdot 10^{-3}$ | $7.67 \cdot 10^{-3}$ | **0.88 x** | Dense |

(i) MobileNetV2, input shape (32, 3, 256, 256)

| Name | TN [s] | PT [s] | Factor | TN + opt [s] | PT [s] | Factor | Type |
|---|---|---|---|---|---|---|---|
| features.0.0 | $2.30 \cdot 10^{-2}$ | $3.63 \cdot 10^{-3}$ | 6.34 x | $2.24 \cdot 10^{-2}$ | $3.65 \cdot 10^{-3}$ | 6.13 x | General |
| features.1.conv.0.0 | $2.84 \cdot 10^{-2}$ | $3.71 \cdot 10^{-2}$ | **0.76 x** | $2.84 \cdot 10^{-2}$ | $3.72 \cdot 10^{-2}$ | **0.76 x** | General |
| features.1.conv.1 | $7.35 \cdot 10^{-3}$ | $5.48 \cdot 10^{-3}$ | 1.34 x | $2.95 \cdot 10^{-3}$ | $5.50 \cdot 10^{-3}$ | **0.54 x** | Dense |
| features.2.conv.0.0 | $4.28 \cdot 10^{-3}$ | $2.92 \cdot 10^{-3}$ | 1.47 x | $1.50 \cdot 10^{-3}$ | $2.91 \cdot 10^{-3}$ | **0.51 x** | Dense |
| features.2.conv.1.0 | $3.98 \cdot 10^{-2}$ | $2.51 \cdot 10^{-2}$ | 1.59 x | $3.98 \cdot 10^{-2}$ | $2.51 \cdot 10^{-2}$ | 1.59 x | General |
| features.2.conv.2 | $5.33 \cdot 10^{-3}$ | $4.67 \cdot 10^{-3}$ | 1.14 x | $3.06 \cdot 10^{-3}$ | $5.04 \cdot 10^{-3}$ | **0.61 x** | Dense |
| features.3.conv.0.0 | $1.63 \cdot 10^{-3}$ | $1.37 \cdot 10^{-3}$ | 1.19 x | $7.02 \cdot 10^{-4}$ | $1.39 \cdot 10^{-3}$ | **0.50 x** | Dense |
| features.3.conv.1.0 | $2.07 \cdot 10^{-2}$ | $3.67 \cdot 10^{-2}$ | **0.56 x** | $2.06 \cdot 10^{-2}$ | $3.68 \cdot 10^{-2}$ | **0.56 x** | General |
| features.3.conv.2 | $9.72 \cdot 10^{-3}$ | $9.38 \cdot 10^{-3}$ | 1.04 x | $6.47 \cdot 10^{-3}$ | $9.36 \cdot 10^{-3}$ | **0.69 x** | Dense |
| features.4.conv.1.0 | $1.15 \cdot 10^{-2}$ | $1.02 \cdot 10^{-2}$ | 1.13 x | $1.15 \cdot 10^{-2}$ | $1.02 \cdot 10^{-2}$ | 1.13 x | General |
| features.4.conv.2 | $2.82 \cdot 10^{-3}$ | $2.65 \cdot 10^{-3}$ | 1.06 x | $1.77 \cdot 10^{-3}$ | $2.64 \cdot 10^{-3}$ | **0.67 x** | Dense |
| features.5.conv.0.0 | $1.05 \cdot 10^{-3}$ | $7.05 \cdot 10^{-4}$ | 1.49 x | $3.84 \cdot 10^{-4}$ | $7.08 \cdot 10^{-4}$ | **0.54 x** | Dense |
| features.5.conv.1.0 | $6.38 \cdot 10^{-3}$ | $1.19 \cdot 10^{-2}$ | **0.54 x** | $6.36 \cdot 10^{-3}$ | $1.19 \cdot 10^{-2}$ | **0.53 x** | General |
| features.5.conv.2 | $3.39 \cdot 10^{-3}$ | $3.16 \cdot 10^{-3}$ | 1.07 x | $2.10 \cdot 10^{-3}$ | $3.18 \cdot 10^{-3}$ | **0.66 x** | Dense |
| features.7.conv.1.0 | $3.66 \cdot 10^{-3}$ | $3.66 \cdot 10^{-3}$ | 1.00 x | $3.69 \cdot 10^{-3}$ | $3.67 \cdot 10^{-3}$ | 1.01 x | General |
| features.7.conv.2 | $1.41 \cdot 10^{-3}$ | $1.28 \cdot 10^{-3}$ | 1.10 x | $7.93 \cdot 10^{-4}$ | $1.28 \cdot 10^{-3}$ | **0.62 x** | Dense |
| features.8.conv.0.0 | $9.96 \cdot 10^{-4}$ | $6.18 \cdot 10^{-4}$ | 1.61 x | $3.37 \cdot 10^{-4}$ | $6.26 \cdot 10^{-4}$ | **0.54 x** | Dense |
| features.8.conv.1.0 | $2.88 \cdot 10^{-3}$ | $6.25 \cdot 10^{-3}$ | **0.46 x** | $2.87 \cdot 10^{-3}$ | $6.26 \cdot 10^{-3}$ | **0.46 x** | General |
| features.8.conv.2 | $2.36 \cdot 10^{-3}$ | $2.24 \cdot 10^{-3}$ | 1.06 x | $1.55 \cdot 10^{-3}$ | $2.24 \cdot 10^{-3}$ | **0.69 x** | Dense |
| features.11.conv.2 | $2.33 \cdot 10^{-3}$ | $2.22 \cdot 10^{-3}$ | 1.05 x | $1.55 \cdot 10^{-3}$ | $2.24 \cdot 10^{-3}$ | **0.69 x** | Dense |
| features.12.conv.0.0 | $9.43 \cdot 10^{-4}$ | $7.06 \cdot 10^{-4}$ | 1.34 x | $3.87 \cdot 10^{-4}$ | $7.07 \cdot 10^{-4}$ | **0.55 x** | Dense |
| features.12.conv.1.0 | $4.07 \cdot 10^{-3}$ | $8.89 \cdot 10^{-3}$ | **0.46 x** | $4.04 \cdot 10^{-3}$ | $8.90 \cdot 10^{-3}$ | **0.45 x** | General |
| features.12.conv.2 | $3.97 \cdot 10^{-3}$ | $3.84 \cdot 10^{-3}$ | 1.03 x | $2.97 \cdot 10^{-3}$ | $3.85 \cdot 10^{-3}$ | **0.77 x** | Dense |
| features.14.conv.1.0 | $2.41 \cdot 10^{-3}$ | $2.66 \cdot 10^{-3}$ | **0.91 x** | $2.39 \cdot 10^{-3}$ | $2.66 \cdot 10^{-3}$ | **0.90 x** | General |
| features.14.conv.2 | $1.50 \cdot 10^{-3}$ | $1.23 \cdot 10^{-3}$ | 1.22 x | $9.00 \cdot 10^{-4}$ | $1.23 \cdot 10^{-3}$ | **0.73 x** | Dense |
| features.15.conv.0.0 | $9.14 \cdot 10^{-4}$ | $6.34 \cdot 10^{-4}$ | 1.44 x | $3.38 \cdot 10^{-4}$ | $6.23 \cdot 10^{-4}$ | **0.54 x** | Dense |
| features.15.conv.1.0 | $9.60 \cdot 10^{-4}$ | $4.01 \cdot 10^{-3}$ | **0.24 x** | $9.83 \cdot 10^{-4}$ | $4.03 \cdot 10^{-3}$ | **0.24 x** | General |
| features.15.conv.2 | $2.57 \cdot 10^{-3}$ | $2.35 \cdot 10^{-3}$ | 1.10 x | $1.85 \cdot 10^{-3}$ | $2.35 \cdot 10^{-3}$ | **0.79 x** | Dense |
| features.17.conv.2 | $2.57 \cdot 10^{-3}$ | $2.35 \cdot 10^{-3}$ | 1.10 x | $1.85 \cdot 10^{-3}$ | $2.35 \cdot 10^{-3}$ | **0.79 x** | Dense |
| features.18.0 | $1.15 \cdot 10^{-3}$ | $7.91 \cdot 10^{-4}$ | 1.46 x | $4.79 \cdot 10^{-4}$ | $7.91 \cdot 10^{-4}$ | **0.61 x** | Dense |

## F.6   KFAC-REDUCE FACTOR

We compare TN and TN+opt with a PyTorch implementation of the input-based KFAC-reduce factor based on `torch.nn.functional.unfold`. Figure F22 visualizes the performance ratios for different convolution categories. Table F8 contains the detailed run times and performance factors.

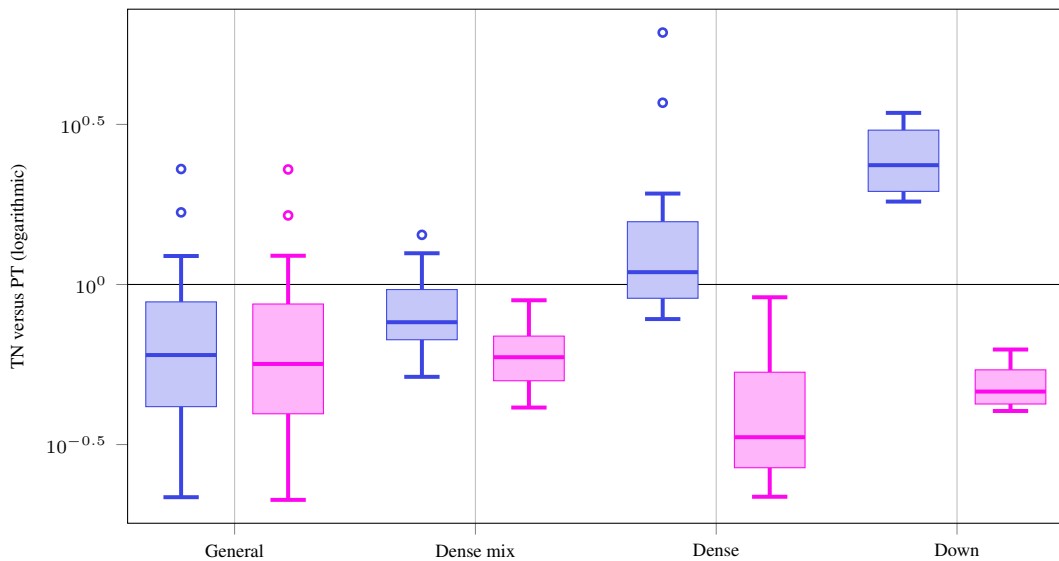

Figure F22: KFAC-reduce factor performance ratios of TN versus PT and TN+opt versus PT for different convolution types on GPU.

Table F8: KFAC-reduce factor performance comparison on GPU.

(a) 3c3d, CIFAR-10, input shape (128, 3, 32, 32)

| Name | TN [s] | PT [s] | Factor | TN + opt [s] | PT [s] | Factor | Type |
|---|---|---|---|---|---|---|---|
| conv1.0 | $8.88 \cdot 10^{-4}$ | $2.26 \cdot 10^{-3}$ | **0.39 x** | $8.59 \cdot 10^{-4}$ | $2.41 \cdot 10^{-3}$ | **0.36 x** | General |
| conv2.0 | $1.41 \cdot 10^{-3}$ | $1.79 \cdot 10^{-3}$ | **0.79 x** | $1.29 \cdot 10^{-3}$ | $1.75 \cdot 10^{-3}$ | **0.74 x** | General |
| conv3.1 | $1.33 \cdot 10^{-3}$ | $2.31 \cdot 10^{-3}$ | **0.57 x** | $1.46 \cdot 10^{-3}$ | $2.37 \cdot 10^{-3}$ | **0.61 x** | General |

(b) F-MNIST 2c2d, input shape (128, 1, 28, 28)

| Name | TN [s] | PT [s] | Factor | TN + opt [s] | PT [s] | Factor | Type |
|---|---|---|---|---|---|---|---|
| conv1.1 | $1.10 \cdot 10^{-3}$ | $1.67 \cdot 10^{-3}$ | **0.66 x** | $1.01 \cdot 10^{-3}$ | $1.83 \cdot 10^{-3}$ | **0.55 x** | General |
| conv2.1 | $1.58 \cdot 10^{-3}$ | $2.57 \cdot 10^{-3}$ | **0.62 x** | $1.54 \cdot 10^{-3}$ | $2.76 \cdot 10^{-3}$ | **0.56 x** | General |

(c) CIFAR-100 All-CNN-C, input shape (128, 3, 32, 32)

| Name | TN [s] | PT [s] | Factor | TN + opt [s] | PT [s] | Factor | Type |
|---|---|---|---|---|---|---|---|
| conv1.1 | $1.11 \cdot 10^{-3}$ | $1.84 \cdot 10^{-3}$ | **0.60 x** | $1.07 \cdot 10^{-3}$ | $1.89 \cdot 10^{-3}$ | **0.56 x** | General |
| conv2.1 | $3.26 \cdot 10^{-3}$ | $8.32 \cdot 10^{-3}$ | **0.39 x** | $3.19 \cdot 10^{-3}$ | $8.24 \cdot 10^{-3}$ | **0.39 x** | General |
| conv3.1 | $3.09 \cdot 10^{-3}$ | $3.56 \cdot 10^{-3}$ | **0.87 x** | $3.08 \cdot 10^{-3}$ | $3.56 \cdot 10^{-3}$ | **0.87 x** | General |
| conv4.1 | $1.45 \cdot 10^{-3}$ | $3.09 \cdot 10^{-3}$ | **0.47 x** | $1.44 \cdot 10^{-3}$ | $3.09 \cdot 10^{-3}$ | **0.47 x** | General |
| conv5.1 | $2.59 \cdot 10^{-3}$ | $5.63 \cdot 10^{-3}$ | **0.46 x** | $2.55 \cdot 10^{-3}$ | $5.62 \cdot 10^{-3}$ | **0.45 x** | General |
| conv6.1 | $2.46 \cdot 10^{-3}$ | $3.49 \cdot 10^{-3}$ | **0.71 x** | $2.43 \cdot 10^{-3}$ | $3.48 \cdot 10^{-3}$ | **0.70 x** | General |
| conv7.0 | $1.55 \cdot 10^{-3}$ | $3.03 \cdot 10^{-3}$ | **0.51 x** | $1.53 \cdot 10^{-3}$ | $3.02 \cdot 10^{-3}$ | **0.51 x** | General |
| conv8.1 | $1.14 \cdot 10^{-3}$ | $1.46 \cdot 10^{-3}$ | **0.78 x** | $3.59 \cdot 10^{-4}$ | $1.36 \cdot 10^{-3}$ | **0.26 x** | Dense |
| conv9.1 | $1.14 \cdot 10^{-3}$ | $1.46 \cdot 10^{-3}$ | **0.79 x** | $3.59 \cdot 10^{-4}$ | $1.36 \cdot 10^{-3}$ | **0.26 x** | Dense |

(d) Alexnet, input shape (32, 3, 256, 256)

| Name | TN [s] | PT [s] | Factor | TN + opt [s] | PT [s] | Factor | Type |
|---|---|---|---|---|---|---|---|
| features.0 | $1.86 \cdot 10^{-3}$ | $4.22 \cdot 10^{-3}$ | **0.44 x** | $1.84 \cdot 10^{-3}$ | $4.22 \cdot 10^{-3}$ | **0.44 x** | General |
| features.3 | $1.60 \cdot 10^{-3}$ | $3.88 \cdot 10^{-3}$ | **0.41 x** | $1.53 \cdot 10^{-3}$ | $3.89 \cdot 10^{-3}$ | **0.39 x** | General |
| features.6 | $1.51 \cdot 10^{-3}$ | $1.64 \cdot 10^{-3}$ | **0.92 x** | $1.43 \cdot 10^{-3}$ | $1.63 \cdot 10^{-3}$ | **0.88 x** | General |
| features.8 | $1.77 \cdot 10^{-3}$ | $3.02 \cdot 10^{-3}$ | **0.59 x** | $1.73 \cdot 10^{-3}$ | $3.02 \cdot 10^{-3}$ | **0.57 x** | General |
| features.10 | $1.56 \cdot 10^{-3}$ | $1.96 \cdot 10^{-3}$ | **0.79 x** | $1.51 \cdot 10^{-3}$ | $1.96 \cdot 10^{-3}$ | **0.77 x** | General |

(e) ResNet18, input shape (32, 3, 256, 256)

| Name | TN [s] | PT [s] | Factor | TN + opt [s] | PT [s] | Factor | Type |
|---|---|---|---|---|---|---|---|
| conv1 | $1.79 \cdot 10^{-3}$ | $5.41 \cdot 10^{-3}$ | **0.33 x** | $1.78 \cdot 10^{-3}$ | $5.40 \cdot 10^{-3}$ | **0.33 x** | General |
| layer1.0.conv1 | $2.24 \cdot 10^{-3}$ | $5.35 \cdot 10^{-3}$ | **0.42 x** | $2.20 \cdot 10^{-3}$ | $5.32 \cdot 10^{-3}$ | **0.41 x** | General |
| layer2.0.conv1 | $2.23 \cdot 10^{-3}$ | $1.99 \cdot 10^{-3}$ | 1.12 x | $2.16 \cdot 10^{-3}$ | $1.96 \cdot 10^{-3}$ | 1.10 x | General |
| layer2.0.conv2 | $1.47 \cdot 10^{-3}$ | $3.02 \cdot 10^{-3}$ | **0.49 x** | $1.47 \cdot 10^{-3}$ | $3.04 \cdot 10^{-3}$ | **0.49 x** | General |
| layer2.0.downsample.0 | $1.85 \cdot 10^{-3}$ | $7.70 \cdot 10^{-4}$ | 2.40 x | $3.24 \cdot 10^{-4}$ | $7.49 \cdot 10^{-4}$ | **0.43 x** | Down |
| layer3.0.conv1 | $1.46 \cdot 10^{-3}$ | $1.21 \cdot 10^{-3}$ | 1.21 x | $1.45 \cdot 10^{-3}$ | $1.21 \cdot 10^{-3}$ | 1.20 x | General |
| layer3.0.conv2 | $1.49 \cdot 10^{-3}$ | $1.96 \cdot 10^{-3}$ | **0.76 x** | $1.36 \cdot 10^{-3}$ | $1.95 \cdot 10^{-3}$ | **0.70 x** | General |
| layer3.0.downsample.0 | $1.26 \cdot 10^{-3}$ | $5.44 \cdot 10^{-4}$ | 2.31 x | $2.68 \cdot 10^{-4}$ | $5.44 \cdot 10^{-4}$ | **0.49 x** | Down |
| layer4.0.conv1 | $1.49 \cdot 10^{-3}$ | $1.33 \cdot 10^{-3}$ | 1.12 x | $1.44 \cdot 10^{-3}$ | $1.33 \cdot 10^{-3}$ | 1.08 x | General |
| layer4.0.conv2 | $1.60 \cdot 10^{-3}$ | $1.86 \cdot 10^{-3}$ | **0.86 x** | $1.62 \cdot 10^{-3}$ | $1.86 \cdot 10^{-3}$ | **0.87 x** | General |
| layer4.0.downsample.0 | $9.63 \cdot 10^{-4}$ | $5.25 \cdot 10^{-4}$ | 1.83 x | $2.57 \cdot 10^{-4}$ | $4.11 \cdot 10^{-4}$ | **0.63 x** | Down |

(f) ResNext101, input shape (32, 3, 256, 256)

| Name | TN [s] | PT [s] | Factor | TN + opt [s] | PT [s] | Factor | Type |
|---|---|---|---|---|---|---|---|
| conv1 | $1.78 \cdot 10^{-3}$ | $5.38 \cdot 10^{-3}$ | **0.33 x** | $1.77 \cdot 10^{-3}$ | $5.39 \cdot 10^{-3}$ | **0.33 x** | General |
| layer1.0.conv1 | $1.87 \cdot 10^{-3}$ | $1.73 \cdot 10^{-3}$ | 1.08 x | $4.47 \cdot 10^{-4}$ | $1.73 \cdot 10^{-3}$ | **0.26 x** | Dense |
| layer1.0.conv2 | $4.44 \cdot 10^{-2}$ | $1.94 \cdot 10^{-2}$ | 2.29 x | $4.43 \cdot 10^{-2}$ | $1.94 \cdot 10^{-2}$ | 2.29 x | General |
| layer1.0.conv3 | $6.09 \cdot 10^{-3}$ | $5.57 \cdot 10^{-3}$ | 1.09 x | $1.21 \cdot 10^{-3}$ | $5.57 \cdot 10^{-3}$ | **0.22 x** | Dense |
| layer2.0.conv1 | $6.09 \cdot 10^{-3}$ | $5.57 \cdot 10^{-3}$ | 1.09 x | $1.21 \cdot 10^{-3}$ | $5.58 \cdot 10^{-3}$ | **0.22 x** | Dense |
| layer2.0.conv2 | $1.37 \cdot 10^{-2}$ | $1.18 \cdot 10^{-2}$ | 1.16 x | $1.37 \cdot 10^{-2}$ | $1.18 \cdot 10^{-2}$ | 1.16 x | General |
| layer2.0.conv3 | $3.81 \cdot 10^{-3}$ | $3.02 \cdot 10^{-3}$ | 1.26 x | $7.44 \cdot 10^{-4}$ | $3.02 \cdot 10^{-3}$ | **0.25 x** | Dense |
| layer2.0.downsample.0 | $6.08 \cdot 10^{-3}$ | $1.77 \cdot 10^{-3}$ | 3.44 x | $7.12 \cdot 10^{-4}$ | $1.77 \cdot 10^{-3}$ | **0.40 x** | Down |
| layer2.1.conv2 | $4.16 \cdot 10^{-3}$ | $9.91 \cdot 10^{-3}$ | **0.42 x** | $4.16 \cdot 10^{-3}$ | $9.90 \cdot 10^{-3}$ | **0.42 x** | General |
| layer3.0.conv1 | $3.81 \cdot 10^{-3}$ | $3.02 \cdot 10^{-3}$ | 1.26 x | $7.32 \cdot 10^{-4}$ | $3.02 \cdot 10^{-3}$ | **0.24 x** | Dense |
| layer3.0.conv2 | $7.88 \cdot 10^{-3}$ | $6.42 \cdot 10^{-3}$ | 1.23 x | $7.90 \cdot 10^{-3}$ | $6.43 \cdot 10^{-3}$ | 1.23 x | General |
| layer3.0.conv3 | $1.61 \cdot 10^{-3}$ | $1.78 \cdot 10^{-3}$ | **0.91 x** | $5.42 \cdot 10^{-4}$ | $1.80 \cdot 10^{-3}$ | **0.30 x** | Dense |
| layer3.0.downsample.0 | $3.80 \cdot 10^{-3}$ | $1.17 \cdot 10^{-3}$ | 3.24 x | $5.10 \cdot 10^{-4}$ | $1.21 \cdot 10^{-3}$ | **0.42 x** | Down |
| layer3.1.conv2 | $2.25 \cdot 10^{-3}$ | $5.41 \cdot 10^{-3}$ | **0.42 x** | $2.26 \cdot 10^{-3}$ | $5.41 \cdot 10^{-3}$ | **0.42 x** | General |
| layer4.0.conv1 | $1.61 \cdot 10^{-3}$ | $1.77 \cdot 10^{-3}$ | **0.91 x** | $5.44 \cdot 10^{-4}$ | $1.80 \cdot 10^{-3}$ | **0.30 x** | Dense |
| layer4.0.conv2 | $4.21 \cdot 10^{-3}$ | $5.29 \cdot 10^{-3}$ | **0.80 x** | $4.21 \cdot 10^{-3}$ | $5.29 \cdot 10^{-3}$ | **0.80 x** | General |
| layer4.0.conv3 | $1.23 \cdot 10^{-3}$ | $1.45 \cdot 10^{-3}$ | **0.85 x** | $7.68 \cdot 10^{-4}$ | $1.44 \cdot 10^{-3}$ | **0.53 x** | Dense |
| layer4.0.downsample.0 | $1.62 \cdot 10^{-3}$ | $8.91 \cdot 10^{-4}$ | 1.82 x | $4.97 \cdot 10^{-4}$ | $8.93 \cdot 10^{-4}$ | **0.56 x** | Down |
| layer4.1.conv2 | $2.18 \cdot 10^{-3}$ | $4.73 \cdot 10^{-3}$ | **0.46 x** | $2.16 \cdot 10^{-3}$ | $4.72 \cdot 10^{-3}$ | **0.46 x** | General |

(g) ConvNeXt-base, input shape (32, 3, 256, 256)

| Name | TN [s] | PT [s] | Factor | TN + opt [s] | PT [s] | Factor | Type |
|---|---|---|---|---|---|---|---|
| features.0.0 | $1.72 \cdot 10^{-3}$ | $1.02 \cdot 10^{-3}$ | 1.69 x | $7.52 \cdot 10^{-4}$ | $1.01 \cdot 10^{-3}$ | **0.74 x** | Dense |
| features.1.0.block.0 | $1.53 \cdot 10^{-2}$ | $4.41 \cdot 10^{-2}$ | **0.35 x** | $1.53 \cdot 10^{-2}$ | $4.40 \cdot 10^{-2}$ | **0.35 x** | General |
| features.2.1 | $3.80 \cdot 10^{-3}$ | $1.99 \cdot 10^{-3}$ | 1.91 x | $8.44 \cdot 10^{-4}$ | $1.99 \cdot 10^{-3}$ | **0.43 x** | Dense |
| features.3.0.block.0 | $8.21 \cdot 10^{-3}$ | $2.22 \cdot 10^{-2}$ | **0.37 x** | $8.19 \cdot 10^{-3}$ | $2.22 \cdot 10^{-2}$ | **0.37 x** | General |
| features.4.1 | $2.32 \cdot 10^{-3}$ | $1.21 \cdot 10^{-3}$ | 1.92 x | $6.85 \cdot 10^{-4}$ | $1.18 \cdot 10^{-3}$ | **0.58 x** | Dense |
| features.5.0.block.0 | $4.62 \cdot 10^{-3}$ | $1.18 \cdot 10^{-2}$ | **0.39 x** | $4.57 \cdot 10^{-3}$ | $1.16 \cdot 10^{-2}$ | **0.40 x** | General |
| features.6.1 | $1.40 \cdot 10^{-3}$ | $1.10 \cdot 10^{-3}$ | 1.27 x | $9.28 \cdot 10^{-4}$ | $1.02 \cdot 10^{-3}$ | **0.91 x** | Dense |
| features.7.0.block.0 | $1.38 \cdot 10^{-3}$ | $6.35 \cdot 10^{-3}$ | **0.22 x** | $1.35 \cdot 10^{-3}$ | $6.34 \cdot 10^{-3}$ | **0.21 x** | General |

(h) InceptionV3, input shape (32, 3, 299, 299)

| Name | TN [s] | PT [s] | Factor | TN + opt [s] | PT [s] | Factor | Type |
|---|---|---|---|---|---|---|---|
| Conv2d_1a_3x3.conv | $2.39 \cdot 10^{-3}$ | $2.03 \cdot 10^{-3}$ | 1.18 x | $2.36 \cdot 10^{-3}$ | $2.00 \cdot 10^{-3}$ | 1.18 x | General |
| Conv2d_2a_3x3.conv | $4.42 \cdot 10^{-3}$ | $1.30 \cdot 10^{-2}$ | **0.34 x** | $4.38 \cdot 10^{-3}$ | $1.30 \cdot 10^{-2}$ | **0.34 x** | General |
| Conv2d_2b_3x3.conv | $4.33 \cdot 10^{-3}$ | $1.30 \cdot 10^{-2}$ | **0.33 x** | $4.32 \cdot 10^{-3}$ | $1.30 \cdot 10^{-2}$ | **0.33 x** | General |
| Conv2d_3b_1x1.conv | $1.32 \cdot 10^{-2}$ | $2.16 \cdot 10^{-3}$ | 6.12 x | $5.53 \cdot 10^{-4}$ | $2.16 \cdot 10^{-3}$ | **0.26 x** | Dense |
| Conv2d_4a_3x3.conv | $2.72 \cdot 10^{-3}$ | $8.00 \cdot 10^{-3}$ | **0.34 x** | $2.74 \cdot 10^{-3}$ | $8.02 \cdot 10^{-3}$ | **0.34 x** | General |
| Mixed_5b.branch1x1.conv | $1.43 \cdot 10^{-3}$ | $1.57 \cdot 10^{-3}$ | **0.91 x** | $4.52 \cdot 10^{-4}$ | $1.57 \cdot 10^{-3}$ | **0.29 x** | Dense |
| Mixed_5b.branch5x5_1.conv | $1.43 \cdot 10^{-3}$ | $1.57 \cdot 10^{-3}$ | **0.91 x** | $4.53 \cdot 10^{-4}$ | $1.57 \cdot 10^{-3}$ | **0.29 x** | Dense |
| Mixed_5b.branch5x5_2.conv | $1.56 \cdot 10^{-3}$ | $3.73 \cdot 10^{-3}$ | **0.42 x** | $1.35 \cdot 10^{-3}$ | $3.72 \cdot 10^{-3}$ | **0.36 x** | General |
| Mixed_5b.branch3x3dbl_2.conv | $1.57 \cdot 10^{-3}$ | $1.91 \cdot 10^{-3}$ | **0.82 x** | $1.44 \cdot 10^{-3}$ | $1.91 \cdot 10^{-3}$ | **0.76 x** | General |
| Mixed_5b.branch3x3dbl_3.conv | $1.48 \cdot 10^{-3}$ | $2.66 \cdot 10^{-3}$ | **0.56 x** | $1.41 \cdot 10^{-3}$ | $2.66 \cdot 10^{-3}$ | **0.53 x** | General |
| Mixed_5b.branch_pool.conv | $1.45 \cdot 10^{-3}$ | $1.59 \cdot 10^{-3}$ | **0.91 x** | $4.62 \cdot 10^{-4}$ | $1.59 \cdot 10^{-3}$ | **0.29 x** | Dense |
| Mixed_5c.branch1x1.conv | $1.79 \cdot 10^{-3}$ | $2.00 \cdot 10^{-3}$ | **0.90 x** | $5.48 \cdot 10^{-4}$ | $2.00 \cdot 10^{-3}$ | **0.27 x** | Dense |
| Mixed_5c.branch5x5_1.conv | $1.79 \cdot 10^{-3}$ | $2.00 \cdot 10^{-3}$ | **0.90 x** | $5.46 \cdot 10^{-4}$ | $2.00 \cdot 10^{-3}$ | **0.27 x** | Dense |
| Mixed_5d.branch1x1.conv | $1.99 \cdot 10^{-3}$ | $2.22 \cdot 10^{-3}$ | **0.90 x** | $5.90 \cdot 10^{-4}$ | $2.21 \cdot 10^{-3}$ | **0.27 x** | Dense |
| Mixed_5d.branch5x5_1.conv | $1.97 \cdot 10^{-3}$ | $2.20 \cdot 10^{-3}$ | **0.89 x** | $5.83 \cdot 10^{-4}$ | $2.18 \cdot 10^{-3}$ | **0.27 x** | Dense |
| Mixed_6a.branch3x3.conv | $2.81 \cdot 10^{-3}$ | $3.18 \cdot 10^{-3}$ | **0.88 x** | $2.76 \cdot 10^{-3}$ | $3.18 \cdot 10^{-3}$ | **0.87 x** | General |
| Mixed_6a.branch3x3dbl_3.conv | $1.30 \cdot 10^{-3}$ | $1.22 \cdot 10^{-3}$ | 1.07 x | $1.44 \cdot 10^{-3}$ | $1.29 \cdot 10^{-3}$ | 1.11 x | General |
| Mixed_6b.branch1x1.conv | $1.49 \cdot 10^{-3}$ | $1.64 \cdot 10^{-3}$ | **0.91 x** | $5.47 \cdot 10^{-4}$ | $1.62 \cdot 10^{-3}$ | **0.34 x** | Dense |
| Mixed_6b.branch7x7_1.conv | $1.46 \cdot 10^{-3}$ | $1.64 \cdot 10^{-3}$ | **0.89 x** | $5.60 \cdot 10^{-4}$ | $1.64 \cdot 10^{-3}$ | **0.34 x** | Dense |
| Mixed_6b.branch7x7_2.conv | $1.01 \cdot 10^{-3}$ | $1.07 \cdot 10^{-3}$ | **0.94 x** | $7.56 \cdot 10^{-4}$ | $1.10 \cdot 10^{-3}$ | **0.69 x** | Dense mix |
| Mixed_6b.branch7x7_3.conv | $9.45 \cdot 10^{-4}$ | $1.23 \cdot 10^{-3}$ | **0.77 x** | $7.61 \cdot 10^{-4}$ | $1.23 \cdot 10^{-3}$ | **0.62 x** | Dense mix |
| Mixed_6b.branch7x7dbl_2.conv | $1.06 \cdot 10^{-3}$ | $1.23 \cdot 10^{-3}$ | **0.86 x** | $7.62 \cdot 10^{-4}$ | $1.23 \cdot 10^{-3}$ | **0.62 x** | Dense mix |
| Mixed_6b.branch7x7dbl_5.conv | $1.14 \cdot 10^{-3}$ | $1.10 \cdot 10^{-3}$ | 1.04 x | $7.56 \cdot 10^{-4}$ | $1.10 \cdot 10^{-3}$ | **0.69 x** | Dense mix |
| Mixed_6c.branch7x7_1.conv | $1.43 \cdot 10^{-3}$ | $1.62 \cdot 10^{-3}$ | **0.88 x** | $5.47 \cdot 10^{-4}$ | $1.64 \cdot 10^{-3}$ | **0.33 x** | Dense |
| Mixed_6c.branch7x7_2.conv | $1.01 \cdot 10^{-3}$ | $1.34 \cdot 10^{-3}$ | **0.75 x** | $7.64 \cdot 10^{-4}$ | $1.35 \cdot 10^{-3}$ | **0.56 x** | Dense mix |
| Mixed_6c.branch7x7_3.conv | $1.07 \cdot 10^{-3}$ | $1.69 \cdot 10^{-3}$ | **0.63 x** | $7.69 \cdot 10^{-4}$ | $1.68 \cdot 10^{-3}$ | **0.46 x** | Dense mix |
| Mixed_6c.branch7x7dbl_2.conv | $8.59 \cdot 10^{-4}$ | $1.67 \cdot 10^{-3}$ | **0.51 x** | $7.67 \cdot 10^{-4}$ | $1.69 \cdot 10^{-3}$ | **0.45 x** | Dense mix |
| Mixed_6c.branch7x7dbl_5.conv | $1.01 \cdot 10^{-3}$ | $1.33 \cdot 10^{-3}$ | **0.76 x** | $7.64 \cdot 10^{-4}$ | $1.35 \cdot 10^{-3}$ | **0.57 x** | Dense mix |
| Mixed_6e.branch7x7_2.conv | $1.01 \cdot 10^{-3}$ | $1.48 \cdot 10^{-3}$ | **0.69 x** | $7.68 \cdot 10^{-4}$ | $1.49 \cdot 10^{-3}$ | **0.51 x** | Dense mix |
| Mixed_6e.branch7x7_3.conv | $9.53 \cdot 10^{-4}$ | $1.77 \cdot 10^{-3}$ | **0.54 x** | $7.38 \cdot 10^{-4}$ | $1.79 \cdot 10^{-3}$ | **0.41 x** | Dense mix |
| AuxLogits.conv0.conv | $9.97 \cdot 10^{-4}$ | $6.04 \cdot 10^{-4}$ | 1.65 x | $3.66 \cdot 10^{-4}$ | $6.58 \cdot 10^{-4}$ | **0.56 x** | Dense |
| AuxLogits.conv1.conv | $1.05 \cdot 10^{-3}$ | $1.09 \cdot 10^{-3}$ | **0.97 x** | $9.31 \cdot 10^{-4}$ | $1.09 \cdot 10^{-3}$ | **0.85 x** | General |
| Mixed_7a.branch3x3_2.conv | $1.30 \cdot 10^{-3}$ | $7.75 \cdot 10^{-4}$ | 1.68 x | $1.26 \cdot 10^{-3}$ | $7.67 \cdot 10^{-4}$ | 1.64 x | General |
| Mixed_7a.branch7x7x3_4.conv | $1.34 \cdot 10^{-3}$ | $1.14 \cdot 10^{-3}$ | 1.18 x | $1.36 \cdot 10^{-3}$ | $1.14 \cdot 10^{-3}$ | 1.19 x | General |
| Mixed_7b.branch1x1.conv | $1.03 \cdot 10^{-3}$ | $9.07 \cdot 10^{-4}$ | 1.13 x | $5.08 \cdot 10^{-4}$ | $9.07 \cdot 10^{-4}$ | **0.56 x** | Dense |
| Mixed_7b.branch3x3_1.conv | $1.16 \cdot 10^{-3}$ | $9.30 \cdot 10^{-4}$ | 1.25 x | $5.20 \cdot 10^{-4}$ | $9.10 \cdot 10^{-4}$ | **0.57 x** | Dense |
| Mixed_7b.branch3x3_2a.conv | $1.13 \cdot 10^{-3}$ | $7.94 \cdot 10^{-4}$ | 1.43 x | $6.89 \cdot 10^{-4}$ | $7.93 \cdot 10^{-4}$ | **0.87 x** | Dense mix |
| Mixed_7b.branch3x3_2b.conv | $1.07 \cdot 10^{-3}$ | $8.53 \cdot 10^{-4}$ | 1.25 x | $7.60 \cdot 10^{-4}$ | $8.51 \cdot 10^{-4}$ | **0.89 x** | Dense mix |
| Mixed_7b.branch3x3dbl_1.conv | $1.16 \cdot 10^{-3}$ | $9.32 \cdot 10^{-4}$ | 1.25 x | $5.38 \cdot 10^{-4}$ | $9.32 \cdot 10^{-4}$ | **0.58 x** | Dense |
| Mixed_7b.branch3x3dbl_2.conv | $1.67 \cdot 10^{-3}$ | $1.55 \cdot 10^{-3}$ | 1.08 x | $1.59 \cdot 10^{-3}$ | $1.55 \cdot 10^{-3}$ | 1.02 x | General |
| Mixed_7b.branch_pool.conv | $1.16 \cdot 10^{-3}$ | $6.79 \cdot 10^{-4}$ | 1.71 x | $5.19 \cdot 10^{-4}$ | $6.85 \cdot 10^{-4}$ | **0.76 x** | Dense |
| Mixed_7c.branch1x1.conv | $1.23 \cdot 10^{-3}$ | $1.45 \cdot 10^{-3}$ | **0.85 x** | $7.69 \cdot 10^{-4}$ | $1.44 \cdot 10^{-3}$ | **0.53 x** | Dense |
| Mixed_7c.branch3x3_1.conv | $1.21 \cdot 10^{-3}$ | $1.44 \cdot 10^{-3}$ | **0.84 x** | $7.80 \cdot 10^{-4}$ | $1.45 \cdot 10^{-3}$ | **0.54 x** | Dense |
| Mixed_7c.branch3x3dbl_1.conv | $1.21 \cdot 10^{-3}$ | $1.43 \cdot 10^{-3}$ | **0.84 x** | $7.66 \cdot 10^{-4}$ | $1.44 \cdot 10^{-3}$ | **0.53 x** | Dense |
| Mixed_7c.branch_pool.conv | $1.21 \cdot 10^{-3}$ | $1.44 \cdot 10^{-3}$ | **0.84 x** | $7.80 \cdot 10^{-4}$ | $1.45 \cdot 10^{-3}$ | **0.54 x** | Dense |

(i) MobileNetV2, input shape (32, 3, 256, 256)

| Name | TN [s] | PT [s] | Factor | TN + opt [s] | PT [s] | Factor | Type |
|---|---|---|---|---|---|---|---|
| features.0.0 | $1.90 \cdot 10^{-3}$ | $1.66 \cdot 10^{-3}$ | 1.15 x | $1.91 \cdot 10^{-3}$ | $1.68 \cdot 10^{-3}$ | 1.14 x | General |
| features.1.conv.0.0 | $2.69 \cdot 10^{-3}$ | $9.87 \cdot 10^{-3}$ | **0.27 x** | $2.70 \cdot 10^{-3}$ | $9.89 \cdot 10^{-3}$ | **0.27 x** | General |
| features.1.conv.1 | $1.12 \cdot 10^{-2}$ | $3.03 \cdot 10^{-3}$ | **0.24 x** | $7.12 \cdot 10^{-4}$ | $3.00 \cdot 10^{-3}$ | **0.24 x** | Dense |
| features.2.conv.0.0 | $1.80 \cdot 10^{-3}$ | $1.77 \cdot 10^{-3}$ | 1.02 x | $4.64 \cdot 10^{-4}$ | $1.76 \cdot 10^{-3}$ | **0.26 x** | Dense |
| features.2.conv.1.0 | $7.01 \cdot 10^{-3}$ | $9.06 \cdot 10^{-3}$ | **0.77 x** | $6.99 \cdot 10^{-3}$ | $9.06 \cdot 10^{-3}$ | **0.77 x** | General |
| features.2.conv.2 | $2.59 \cdot 10^{-3}$ | $2.38 \cdot 10^{-3}$ | 1.09 x | $6.08 \cdot 10^{-4}$ | $2.40 \cdot 10^{-3}$ | **0.25 x** | Dense |
| features.3.conv.0.0 | $1.44 \cdot 10^{-3}$ | $9.19 \cdot 10^{-4}$ | 1.57 x | $2.96 \cdot 10^{-4}$ | $9.40 \cdot 10^{-4}$ | **0.31 x** | Dense |
| features.3.conv.1.0 | $2.99 \cdot 10^{-3}$ | $1.12 \cdot 10^{-2}$ | **0.27 x** | $2.99 \cdot 10^{-3}$ | $1.12 \cdot 10^{-2}$ | **0.27 x** | General |
| features.3.conv.2 | $3.65 \cdot 10^{-3}$ | $3.38 \cdot 10^{-3}$ | 1.08 x | $7.92 \cdot 10^{-4}$ | $3.40 \cdot 10^{-3}$ | **0.23 x** | Dense |
| features.4.conv.1.0 | $3.01 \cdot 10^{-3}$ | $3.76 \cdot 10^{-3}$ | **0.80 x** | $2.99 \cdot 10^{-3}$ | $3.77 \cdot 10^{-3}$ | **0.79 x** | General |
| features.4.conv.2 | $1.38 \cdot 10^{-3}$ | $1.16 \cdot 10^{-3}$ | 1.19 x | $3.53 \cdot 10^{-4}$ | $1.16 \cdot 10^{-3}$ | **0.30 x** | Dense |
| features.5.conv.0.0 | $8.51 \cdot 10^{-4}$ | $5.17 \cdot 10^{-4}$ | 1.65 x | $2.77 \cdot 10^{-4}$ | $5.34 \cdot 10^{-4}$ | **0.52 x** | Dense |
| features.5.conv.1.0 | $1.38 \cdot 10^{-3}$ | $3.99 \cdot 10^{-3}$ | **0.34 x** | $1.36 \cdot 10^{-3}$ | $3.99 \cdot 10^{-3}$ | **0.34 x** | General |
| features.5.conv.2 | $1.68 \cdot 10^{-3}$ | $1.36 \cdot 10^{-3}$ | 1.24 x | $3.94 \cdot 10^{-4}$ | $1.35 \cdot 10^{-3}$ | **0.29 x** | Dense |
| features.7.conv.1.0 | $1.37 \cdot 10^{-3}$ | $1.69 \cdot 10^{-3}$ | **0.81 x** | $1.35 \cdot 10^{-3}$ | $1.69 \cdot 10^{-3}$ | **0.80 x** | General |
| features.7.conv.2 | $8.59 \cdot 10^{-4}$ | $7.05 \cdot 10^{-4}$ | 1.22 x | $2.52 \cdot 10^{-4}$ | $7.00 \cdot 10^{-4}$ | **0.36 x** | Dense |
| features.8.conv.0.0 | $8.45 \cdot 10^{-4}$ | $4.92 \cdot 10^{-4}$ | 1.72 x | $2.49 \cdot 10^{-4}$ | $4.93 \cdot 10^{-4}$ | **0.51 x** | Dense |
| features.8.conv.1.0 | $1.16 \cdot 10^{-3}$ | $2.36 \cdot 10^{-3}$ | **0.49 x** | $1.12 \cdot 10^{-3}$ | $2.35 \cdot 10^{-3}$ | **0.47 x** | General |
| features.8.conv.2 | $8.73 \cdot 10^{-4}$ | $9.30 \cdot 10^{-4}$ | **0.94 x** | $3.06 \cdot 10^{-4}$ | $9.29 \cdot 10^{-4}$ | **0.33 x** | Dense |
| features.11.conv.2 | $9.89 \cdot 10^{-4}$ | $9.49 \cdot 10^{-4}$ | 1.04 x | $3.06 \cdot 10^{-4}$ | $9.25 \cdot 10^{-4}$ | **0.33 x** | Dense |
| features.12.conv.0.0 | $9.55 \cdot 10^{-4}$ | $5.32 \cdot 10^{-4}$ | 1.80 x | $2.50 \cdot 10^{-4}$ | $5.14 \cdot 10^{-4}$ | **0.49 x** | Dense |
| features.12.conv.1.0 | $1.51 \cdot 10^{-3}$ | $3.23 \cdot 10^{-3}$ | **0.47 x** | $1.27 \cdot 10^{-3}$ | $3.22 \cdot 10^{-3}$ | **0.39 x** | General |
| features.12.conv.2 | $1.14 \cdot 10^{-3}$ | $1.24 \cdot 10^{-3}$ | **0.92 x** | $3.94 \cdot 10^{-4}$ | $1.17 \cdot 10^{-3}$ | **0.34 x** | Dense |
| features.14.conv.1.0 | $1.51 \cdot 10^{-3}$ | $1.61 \cdot 10^{-3}$ | **0.94 x** | $1.45 \cdot 10^{-3}$ | $1.61 \cdot 10^{-3}$ | **0.90 x** | General |
| features.14.conv.2 | $1.14 \cdot 10^{-3}$ | $6.83 \cdot 10^{-4}$ | 1.67 x | $3.67 \cdot 10^{-4}$ | $6.80 \cdot 10^{-4}$ | **0.54 x** | Dense |
| features.15.conv.0.0 | $9.53 \cdot 10^{-4}$ | $5.23 \cdot 10^{-4}$ | 1.82 x | $2.74 \cdot 10^{-4}$ | $5.23 \cdot 10^{-4}$ | **0.52 x** | Dense |
| features.15.conv.1.0 | $1.41 \cdot 10^{-3}$ | $2.25 \cdot 10^{-3}$ | **0.63 x** | $1.37 \cdot 10^{-3}$ | $2.25 \cdot 10^{-3}$ | **0.61 x** | General |
| features.15.conv.2 | $1.15 \cdot 10^{-3}$ | $8.81 \cdot 10^{-4}$ | 1.31 x | $4.46 \cdot 10^{-4}$ | $8.83 \cdot 10^{-4}$ | **0.51 x** | Dense |
| features.17.conv.2 | $1.16 \cdot 10^{-3}$ | $8.80 \cdot 10^{-4}$ | 1.31 x | $4.36 \cdot 10^{-4}$ | $8.58 \cdot 10^{-4}$ | **0.51 x** | Dense |
| features.18.0 | $9.51 \cdot 10^{-4}$ | $5.40 \cdot 10^{-4}$ | 1.76 x | $2.50 \cdot 10^{-4}$ | $5.22 \cdot 10^{-4}$ | **0.48 x** | Dense |

# G  MEMORY EVALUATION DETAILS (CPU)

Here, we investigate the peak memory consumption of our proposed TN implementations.

## G.1  THEORETICAL & EMPIRICAL ANALYSIS FOR KFAC-REDUCE FACTOR

We assume a two-dimensional convolution with input $\mathbf{X}$ of shape $(C_{\text{in}}, I_1, I_2)$, output of shape $(C_{\text{out}}, O_1, O_2)$ and kernel of shape $(C_{\text{out}}, C_{\text{in}}, K_1, K_2)$. The analysis with a batch dimension is analogous; hence we suppress it here to de-clutter the notation.

The main difference between the default and our proposed TN implementation of $\hat{\mathbf{\Omega}}$ from §3.3 lies in the computation of the averaged unfolded input $[\![\mathbf{X}]\!]^{(\text{avg})} := {}^{1}\!/_{(O_1 O_2)} \mathbf{1}_{O_1 O_2}^{\top} [\![\mathbf{X}]\!]$ which consists of $C_{\text{in}} K_1 K_2$ numbers. In the following, we will look at the extra memory on top of storing the input $\mathbf{X}$, the averaged unfolded input $[\![\mathbf{X}]\!]^{(\text{avg})}$, and the result $\hat{\mathbf{\Omega}}$.

**Default implementation:**  The standard implementation computes $[\![\mathbf{X}]\!]^{(\text{avg})}$ via the unfolded input $[\![X]\!]$ and thus requires extra storage of $C_{\text{in}} K_1 K_2 O_1 O_2$ numbers.

**TN implementation (general case):**  The TN implementation requires storing the averaged index patterns $\mathbf{\Pi}^{(i,\text{avg})} := {}^{1}\!/_{O_i} \sum_{o=1}^{O_i} [\mathbf{\Pi}^{(i)}]_{:,o,:}$ for $i = 1, 2$. These are directly computed via a slight modification of Algorithm D1 and require storing $I_1 K_1 + I_2 K_2$ numbers. In contrast to the default implementation, spatial dimensions are de-coupled and there is no dependency on $C_{\text{in}}$.

**TN implementation (structured case):**  For structured convolutions (Figure 6) we can describe the action of the index pattern tensor through reshape and narrowing operations. ML libraries usually perform these without allocating additional memory. Hence, our symbolic simplifications completely eliminate the allocation of temporary intermediates to compute $[\![\mathbf{X}]\!]^{(\text{avg})}$.

**Empirical results:**  To demonstrate the memory reduction inside the computation of $\hat{\mathbf{\Omega}}$ we measure its peak memory with the `memory-profiler` library and subtract the memory required to store $\mathbf{X}$ and $\hat{\mathbf{\Omega}}$. This approximates the extra internal memory requirement of an implementation. With the setup of §F we report the minimum additional memory over 50 independent runs in Table G9. We consistently observe that the TN implementation has lower peak memory, which is further reduced by our symbolic simplifications (see for example the effect on ResNext101's dense and down-sampling convolutions in Table G9f).

Our theoretical analysis from above suggests that the peak memory difference becomes most visible for many channels with large kernel and output sizes. One example are ConxNeXt-base's `features.1.0.block.0` convolutions with $K_1 = K_2 = 7$, $O_1 = O_2 = 64$, and $C_{\text{in}} = 128$ (Table E3g). For those convolutions, we observe that the default implementation requires an additional $3{,}140\,\text{MiB}$ ($\approx 3\,\text{GiB!}$) of memory, whereas the TN implementation has zero extra memory demand (Table G9g). This is consistent with our theoretical analysis in that the overhead is storing the unfolded input, which has $(N = 32) \cdot (C_{\text{in}} = 128) \cdot (O_1 = 64) \cdot (O_2 = 64) \cdot (K_1 = 7) \cdot (K_2 = 7) = 822{,}083{,}584$ `float32` entries, corresponding to $3{,}136\,\text{MiB}$.

Table G9: Additional internally required memory to compute the KFAC-reduce factor (measured on CPU). The value 0 indicates that an implementation's peak memory matches the memory consumption of its input $\mathbf{X}$ and result $\hat{\Omega}$.

(a) 3c3d, CIFAR-10, input shape (128, 3, 32, 32)

| Name | TN [MiB] | TN + opt [MiB] | PT [MiB] | Type |
|------|----------|----------------|----------|------|
| conv1.0 | 0.0 | 0.0 | 0.0 | General |
| conv2.0 | 0.0 | 0.0 | 0.0 | General |
| conv3.1 | 0.0 | 0.0 | 0.0 | General |

(b) F-MNIST 2c2d, input shape (128, 1, 28, 28)

| Name | TN [MiB] | TN + opt [MiB] | PT [MiB] | Type |
|------|----------|----------------|----------|------|
| conv1.1 | 0.0 | 0.0 | 0.0 | General |
| conv2.1 | 0.0 | 0.0 | 0.0 | General |

(c) CIFAR-100 All-CNN-C, input shape (128, 3, 32, 32)

| Name | TN [MiB] | TN + opt [MiB] | PT [MiB] | Type |
|------|----------|----------------|----------|------|
| conv1.1 | 0.0 | 0.0 | 0.0 | General |
| conv2.1 | 0.0 | 0.0 | 431 | General |
| conv3.1 | 0.0 | 0.0 | 0.0 | General |
| conv4.1 | 0.0 | 0.0 | 0.0 | General |
| conv5.1 | 0.0 | 0.0 | 215 | General |
| conv6.1 | 0.0 | 0.0 | 0.0 | General |
| conv7.0 | 0.0 | 0.0 | 0.0 | General |
| conv8.1 | 0.0 | 0.0 | 0.0 | Dense |
| conv9.1 | 0.0 | 0.0 | 0.0 | Dense |

(d) Alexnet, input shape (32, 3, 256, 256)

| Name | TN [MiB] | TN + opt [MiB] | PT [MiB] | Type |
|------|----------|----------------|----------|------|
| features.0 | 0.0 | 0.0156 | 175 | General |
| features.3 | 0.0 | 0.0 | 186 | General |
| features.6 | 0.0 | 0.0156 | 0.0 | General |
| features.8 | 0.0 | 0.0156 | 93.8 | General |
| features.10 | 0.0 | 0.0195 | 0.0 | General |

(e) ResNet18, input shape (32, 3, 256, 256)

| Name | TN [MiB] | TN + opt [MiB] | PT [MiB] | Type |
|------|----------|----------------|----------|------|
| conv1 | 0.0 | 0.0 | 293 | General |
| layer1.0.conv1 | 0.0 | 0.0 | 287 | General |
| layer2.0.conv1 | 31.7 | 0.0 | 71.1 | General |
| layer2.0.conv2 | 0.0 | 0.0 | 143 | General |
| layer2.0.downsample.0 | 0.0 | 0.0 | 0.0 | Down |
| layer3.0.conv1 | 0.0 | 0.0 | 0.0 | General |
| layer3.0.conv2 | 0.0 | 0.0 | 70.8 | General |
| layer3.0.downsample.0 | 0.0 | 0.0 | 0.0 | Down |
| layer4.0.conv1 | 0.0 | 0.0 | 0.0 | General |
| layer4.0.conv2 | 0.0 | 80.3 | 0.0 | General |
| layer4.0.downsample.0 | 0.0 | 0.0 | 0.0 | Down |

(f) ResNext101, input shape (32, 3, 256, 256)

| Name | TN [MiB] | TN + opt [MiB] | PT [MiB] | Type |
|------|----------|----------------|----------|------|
| conv1 | 0.0 | 0.0 | 293 | General |
| layer1.0.conv1 | 0.0 | 0.0 | 0.0 | Dense |
| layer1.0.conv2 | 576 | 576 | 1150 | General |
| layer1.0.conv3 | 128 | 0.0 | 127 | Dense |
| layer2.0.conv1 | 128 | 0.0 | 127 | Dense |
| layer2.0.conv2 | 256 | 256 | 575 | General |
| layer2.0.conv3 | 0.0 | 0.0 | 0.0 | Dense |
| layer2.0.downsample.0 | 128 | 0.0 | 19.3 | Down |
| layer2.1.conv2 | 0.0 | 0.0 | 575 | General |
| layer3.0.conv1 | 0.0 | 0.0 | 0.0 | Dense |
| layer3.0.conv2 | 128 | 128 | 288 | General |
| layer3.0.conv3 | 0.0 | 0.0 | 0.0 | Dense |
| layer3.0.downsample.0 | 0.0 | 0.0 | 0.0 | Down |
| layer3.1.conv2 | 0.0 | 0.0 | 288 | General |
| layer4.0.conv1 | 0.0 | 0.0 | 0.0 | Dense |
| layer4.0.conv2 | 0.0 | 0.0 | 144 | General |
| layer4.0.conv3 | 0.0 | 0.0 | 0.0 | Dense |
| layer4.0.downsample.0 | 0.0 | 0.0 | 0.0 | Down |
| layer4.1.conv2 | 0.0 | 0.0 | 144 | General |

(g) ConvNeXt-base, input shape (32, 3, 256, 256)

| Name | TN [MiB] | TN + opt [MiB] | PT [MiB] | Type |
|------|----------|----------------|----------|------|
| features.0.0 | 0.0 | 0.0 | 0.0 | Dense |
| features.1.0.block.0 | 0.0 | 0.0 | 3140 | General |
| features.2.1 | 0.0 | 0.0 | 0.0 | Dense |
| features.3.0.block.0 | 0.0 | 0.0 | 1570 | General |
| features.4.1 | 0.0 | 0.0 | 0.0 | Dense |
| features.5.0.block.0 | 0.0 | 0.0 | 784 | General |
| features.6.1 | 0.0 | 0.0 | 0.0 | Dense |
| features.7.0.block.0 | 0.0 | 0.0 | 392 | General |

(h) InceptionV3, input shape (32, 3, 299, 299)

| Name | TN [MiB] | TN + opt [MiB] | PT [MiB] | Type |
|---|---|---|---|---|
| Conv2d_1a_3x3.conv | 54.6 | 0.0 | 73.0 | General |
| Conv2d_2a_3x3.conv | 86.7 | 86.7 | 759 | General |
| Conv2d_2b_3x3.conv | 84.4 | 84.4 | 758 | General |
| Conv2d_3b_1x1.conv | 166 | 0.0 | 0.0 | Dense |
| Conv2d_4a_3x3.conv | 52.0 | 0.0 | 442 | General |
| Mixed_5b.branch1x1.conv | 0.0 | 0.0 | 0.0 | Dense |
| Mixed_5b.branch5x5_1.conv | 0.0 | 0.0 | 0.0 | Dense |
| Mixed_5b.branch5x5_2.conv | 0.0 | 0.0 | 178 | General |
| Mixed_5b.branch3x3dbl_2.conv | 0.0 | 0.0 | 84.8 | General |
| Mixed_5b.branch3x3dbl_3.conv | 0.0 | 0.0 | 128 | General |
| Mixed_5b.branch_pool.conv | 0.0 | 0.0 | 0.0 | Dense |
| Mixed_5c.branch1x1.conv | 0.0 | 0.0 | 0.0 | Dense |
| Mixed_5c.branch5x5_1.conv | 0.0 | 0.0 | 0.0 | Dense |
| Mixed_5d.branch1x1.conv | 42.7 | 0.0 | 0.0 | Dense |
| Mixed_5d.branch5x5_1.conv | 42.8 | 0.0 | 0.0 | Dense |
| Mixed_6a.branch3x3.conv | 0.0 | 0.0 | 0.0 | General |
| Mixed_6a.branch3x3dbl_3.conv | 0.0 | 0.0 | 0.0 | General |
| Mixed_6b.branch1x1.conv | 0.0 | 0.0 | 0.0 | Dense |
| Mixed_6b.branch7x7_1.conv | 0.0 | 0.0 | 0.0 | Dense |
| Mixed_6b.branch7x7_2.conv | 0.0 | 0.0 | 0.0 | Dense mix |
| Mixed_6b.branch7x7_3.conv | 0.0 | 0.0 | 0.0 | Dense mix |
| Mixed_6b.branch7x7dbl_2.conv | 0.0 | 0.0 | 0.0 | Dense mix |
| Mixed_6b.branch7x7dbl_5.conv | 0.0 | 0.0 | 0.0 | Dense mix |
| Mixed_6c.branch7x7_1.conv | 0.0195 | 0.0 | 0.0 | Dense |
| Mixed_6c.branch7x7_2.conv | 0.0156 | 0.0 | 0.0 | Dense mix |
| Mixed_6c.branch7x7_3.conv | 0.0 | 0.0 | 0.0 | Dense mix |
| Mixed_6c.branch7x7dbl_2.conv | 0.0 | 0.0 | 0.0 | Dense mix |
| Mixed_6c.branch7x7dbl_5.conv | 0.0 | 0.0 | 0.0 | Dense mix |
| Mixed_6e.branch7x7_2.conv | 0.0 | 0.0 | 0.0 | Dense mix |
| Mixed_6e.branch7x7_3.conv | 0.0 | 0.0 | 0.0 | Dense mix |
| AuxLogits.conv0.conv | 0.0 | 0.0 | 0.0 | Dense |
| AuxLogits.conv1.conv | 0.0 | 0.0 | 0.0 | General |
| Mixed_7a.branch3x3_2.conv | 0.0 | 0.0 | 0.0 | General |
| Mixed_7a.branch7x7x3_4.conv | 0.0 | 0.0 | 0.0 | General |
| Mixed_7b.branch1x1.conv | 0.0 | 0.0 | 0.0 | Dense |
| Mixed_7b.branch3x3_1.conv | 0.0 | 0.0 | 0.0 | Dense |
| Mixed_7b.branch3x3_2a.conv | 0.0 | 0.0 | 0.0 | Dense mix |
| Mixed_7b.branch3x3_2b.conv | 0.0 | 0.0 | 0.0 | Dense mix |
| Mixed_7b.branch3x3dbl_1.conv | 0.0 | 0.0 | 0.0 | Dense |
| Mixed_7b.branch3x3dbl_2.conv | 0.0 | 0.0 | 0.0 | General |
| Mixed_7b.branch_pool.conv | 0.0 | 0.0 | 0.0 | Dense |
| Mixed_7c.branch1x1.conv | 0.0 | 0.0 | 0.0 | Dense |
| Mixed_7c.branch3x3_1.conv | 0.0 | 0.0 | 0.0 | Dense |
| Mixed_7c.branch3x3dbl_1.conv | 0.0 | 0.0 | 0.0 | Dense |
| Mixed_7c.branch_pool.conv | 0.0 | 0.0 | 0.0 | Dense |

(i) MobileNetV2, input shape (32, 3, 256, 256)

| Name | TN [MiB] | TN + opt [MiB] | PT [MiB] | Type |
|---|---|---|---|---|
| features.0.0 | 0.0 | 0.0 | 53.8 | General |
| features.1.conv.0.0 | 26.1 | 26.1 | 576 | General |
| features.1.conv.1 | 128 | 0.0 | 63.8 | Dense |
| features.2.conv.0.0 | 0.0 | 0.0 | 0.0 | Dense |
| features.2.conv.1.0 | 192 | 192 | 432 | General |
| features.2.conv.2 | 0.0 | 0.0 | 0.0 | Dense |
| features.3.conv.0.0 | 0.0 | 0.0 | 0.0 | Dense |
| features.3.conv.1.0 | 34.1 | 70.4 | 648 | General |
| features.3.conv.2 | 71.7 | 0.0 | 71.4 | Dense |
| features.4.conv.1.0 | 59.5 | 55.7 | 162 | General |
| features.4.conv.2 | 0.0 | 0.0 | 0.0 | Dense |
| features.5.conv.0.0 | 0.0 | 0.0 | 0.0 | Dense |
| features.5.conv.1.0 | 0.0 | 0.0 | 215 | General |
| features.5.conv.2 | 0.0 | 0.0 | 0.0 | Dense |
| features.7.conv.1.0 | 0.0 | 0.0 | 53.3 | General |
| features.7.conv.2 | 0.0 | 0.0 | 0.0 | Dense |
| features.8.conv.0.0 | 0.0 | 0.0 | 0.0 | Dense |
| features.8.conv.1.0 | 0.0 | 0.0 | 107 | General |
| features.8.conv.2 | 0.0 | 0.0 | 0.0 | Dense |
| features.11.conv.2 | 0.0 | 0.0 | 0.0 | Dense |
| features.12.conv.0.0 | 0.0 | 0.0 | 0.0 | Dense |
| features.12.conv.1.0 | 0.0 | 0.0 | 161 | General |
| features.12.conv.2 | 0.0 | 0.0 | 0.0 | Dense |
| features.14.conv.1.0 | 0.0 | 0.0 | 39.7 | General |
| features.14.conv.2 | 0.0 | 0.0 | 0.0 | Dense |
| features.15.conv.0.0 | 0.0 | 0.0 | 0.0 | Dense |
| features.15.conv.1.0 | 0.0 | 0.0 | 63.8 | General |
| features.15.conv.2 | 0.0 | 0.0 | 0.0 | Dense |
| features.17.conv.2 | 0.0 | 0.0 | 0.0 | Dense |
| features.18.0 | 0.0 | 0.0 | 0.0 | Dense |

# H   MISCELLANEOUS

## H.1   EXAMPLE: ASSOCIATIVITY OF TENSOR MULTIPLICATION

Here, we demonstrate associativity of tensor multiplication through an example. The technical challenge is that an index can only be summed once there are no remaining tensors sharing it. Therefore, we must carry indices that are summed in later multiplications in the intermediate results, which requires some set arithmetic on the index sets.

Let $S_1, S_2, S_3$ be index tuples of the input tensors $\mathbf{A}, \mathbf{B}, \mathbf{C}$, and $S_4 \subseteq (S_1 \cup S_2 \cup S_3)$ a valid output index tuple of their tensor multiplication $\mathbf{D} = *_{(S_1, S_2, S_3, S_4)}(\mathbf{A}, \mathbf{B}, \mathbf{C})$. We can either first multiply $\mathbf{A}$ with $\mathbf{B}$ to obtain an intermediate tensor of index structure $S_{1,2}$, or $\mathbf{B}$ with $\mathbf{C}$ to obtain an intermediate tensor of index structure $S_{2,3}$, before carrying out the remaining multiplications. To construct the intermediate index structures, we divide the indices $\tilde{S} = (S_1 \cup S_2 \cup S_3) \setminus S_4$ that are summed over into those only shared between $\mathbf{A}, \mathbf{B}$ given by $\tilde{S}_{1,2} = (S_1 \cup S_2) \setminus (S_4 \cup S_3)$, and those only shared among $\mathbf{B}, \mathbf{C}$ given by $\tilde{S}_{2,3} = (S_2 \cup S_3) \setminus (S_4 \cup S_1)$. This yields the intermediate indices $S_{1,2} = (S_1 \cup S_2) \setminus \tilde{S}_{1,2}$ and $S_{2,3} = (S_2 \cup S_3) \setminus \tilde{S}_{2,3}$, and the parenthesizations

$$
\begin{aligned}
[\mathbf{D}]_{S_4} &= \left( \sum_{\tilde{S} \setminus \tilde{S}_{1,2}} \left( \sum_{\tilde{S}_{1,2}} [\mathbf{A}]_{S_1} [\mathbf{B}]_{S_2} \right) [\mathbf{C}]_{S_3} \right) = \left( \sum_{\tilde{S} \setminus \tilde{S}_{2,3}} [\mathbf{A}]_{S_1} \left( \sum_{\tilde{S}_{2,3}} [\mathbf{B}]_{S_2} [\mathbf{C}]_{S_3} \right) \right) \\
&\Leftrightarrow \mathbf{D} = *_{(S_{1,2}, S_3, S_4)} \left( *_{(S_2, S_3, S_{2,3})}(\mathbf{A}, \mathbf{B}), \mathbf{C} \right) = *_{(S_1, S_{2,3}, S_4)} \left( \mathbf{A}, *_{(S_1, S_2, S_{1,2})}(\mathbf{B}, \mathbf{C}) \right) .
\end{aligned}
\tag{H15}
$$

This generalizes to $n$-ary multiplication, allowing to break it down into smaller multiplications. However, the index notation and set arithmetic from Equation (H15) quickly becomes impractical.

## H.2   EXAMPLE: MATRIX-MATRIX MULTIPLICATION AS TENSOR MULTIPLICATION

Here we provide a small self-contained example that demonstrates Equation (3) for matrix-matrix multiplication.

Consider two matrices $A, B$ which are compatible for multiplication and let $C = AB$. In index notation, we have

$$
[C]_{i,k} = \sum_j [A]_{i,j} [B]_{j,k} .
$$

The index tuples are $S_A = (i, j)$, $S_B = (j, k)$, and $S_C = (i, k)$. Next, we evaluate which indices are summed over. Since the order of those indices does not matter, we can interpret the tuples as sets and use set arithmetic:

$$
(S_A \cup S_B) \setminus S_C = ((i, j) \cup (j, k)) \setminus (i, k) = (j) \setminus (i, k) = (j) .
$$

Now we see that matrix-matrix multiplication is a case of tensor multiplication (Equation (3)),

$$
[C]_{S_C} = \sum_{(S_A \cup S_B) \setminus S_C} [A]_{S_A} [B]_{S_B} = *_{(S_A, S_B, S_C)}(A, B) .
$$