# OpenReview forum: "Convolutions Through the Lens of Tensor Networks"
_ICLR.cc/2024/Conference — Submitted to ICLR 2024_

### Official Review · Reviewer_4fmq · 2023-10-19

**Soundness:** 2 fair
**Presentation:** 2 fair
**Contribution:** 2 fair
**Rating:** 5
**Confidence:** 3

**Summary:**

This paper proposes a simplifying perspective onto convolutions through tensor networks (TNs). The authors first demonstrate the expressive power of TN by deriving the diagrams of various auto-differentiation operations and popular approximations of second-order information with different hyper-parameters. Using TN also allows re-wiring and simplifying diagrams for faster computation. Based on established machinery for efficient TN contraction, experimental results demonstrate that using TN speeds up a recently-proposed KFAC variant and enables new hardware-efficient tensor dropout for approximate backpropagation.

**Strengths:**

The authors propose a novel perspective on convolution operations from tensor network that can leads to faster computation

**Weaknesses:**

- The novelty of this work is a bit limited, specifically compared with (Hayashi et al. 2019).
- This paper is a bit difficult to understand without enough prior knowledge on tensor networks
- Empirical results may be further improved to better support the claim, details can be found in Questions part.

**Questions:**

- From current draft, I am a bit confused on the difference between this work and (Hayashi et al. 2019). I suppose this paper proposes to compute some first-order and second-order information from tensor network as well. Then what is the difference between using standard auto-differentiation packages and the proposed method based on tensor network? It seems that in experiments, the authors only compare the proposed method with standard PyTorch implementations, but not tensor network combined with auto-differentiation packages. Some explanations may be needed here.
- Based on the above concern, I also wonder if we need to store these computation patterns derived in this paper in implementation. If that is the case, then given new types of convolutions or differentiation operations, will we need to derive some formula again? That sounds not so flexible compared with standard auto-differentiation packages.
- I also wonder how is the index tensor \Pi stored in real applications. Since it should be a very sparse tensor, do we have to use some sparse formats? How will it affect the computation time? The authors may need to add more details here.
- Experimental results are a bit limited from my perspective. While the proposed method based on tensor network really offers some speedup in computation, I suppose there are many other works on speed up inference time (e.g., [1]). Without such comparison, it is hard to see how the proposed method outperforms other works.
- I also note that most experiments are performed with simple convolution operations, while there are also many different types of convolutions (e.g., separate convolution). It would be better if the authors can also compare with these operations to demonstrate the flexibility of tensor network.
- Given that the authors have conducted many experiments on using tensor networks to compute higher-order information, it would be better if the authors can provide some more applications with such information to better demonstrate the applicability of proposed method.

Minor: formatting issues. Some captions in the appendix seems to be overlap with the page head.

Reference:
[1] Fast algorithms for convolutional neural networks. CVPR 2016

---

> ### Author Response · Authors · 2023-11-15
> **Response to your questions (1/2)**
>
> Dear Reviewer 4fmq,
>
> thanks for your detailed review, especially for taking the time to formulate all of your questions. We would like to address them in the following. Please let us know if you have follow-up questions or think we did not address all of them. We are happy to discuss and provide more details.
>
> **Difference to using auto-diff & Hayashi et al.:** By deriving the Jacobians, our goal was not to replace standard auto-diff. Rather, we wanted to demonstrate that using the TN formulation of convolutions yields white-box implementations, not only of many standard operations like vector-Jacobian products, but also less traditional operations used to approximate second-order information. We argue that this improves flexibility and performance:
>
> - KFAC-reduce is a great example for a non-standard operation that is interesting for second-order methods, but whose implementation in existing ML frameworks suffers from limitations. We showed that a TN implementation's performance improves over the default implementation (**New:** We have added memory benchmarks to address a request from Reviewer D3NB and showed that our TN version also has consistently lower peak memory with savings up to 3 GiB).
>
> - Our second main application, randomized autodiff, demonstrates the flexibility of a TN implementation: While ML frameworks support efficient multiplication with the Jacobian, randomizing the vector-Jacobian product is hard because its implementation is a black box. Our TN VJP is a white box that can be randomized more easily.
>
> The applications we investigate in the main text are representatives of operations that are challenging to realize with existing frameworks. There are many others, e.g. approximating Hessian diagonals (Appendix B4), or diagonals and mini-block diagonals of GGN/Fisher matrices (Appendix B2 & C). Not only do we mention those in the introduction, but also do we provide their tensor diagrams in the appendix (see Appendix A and Table B2 for visual and pseudo-code overviews). By doing so, we make them accessible to the community, and allow other works to benefit from our proposed TN simplifications.
>
> In summary, we believe that the focus of our work is different from Hayashi et al. in that it is concerned with investigating and improving computational performance, as well as significantly extending the amount of operations beyond the forward pass.
>
> **Support for different convolution types:** We believe that our framework is fully-compatible with structured convolutions like those mentioned in [1]. You mentioned separable convolutions which we believe are referred to as 'convolutions with channel groups' in our paper. Our formulation supports channel groups, and therefore separable convolutions. To improve clarity, we have added a half sentence in the paper which mentions the alternative name.
>
> Our derivations also carry over to other structured convolutions mentioned in Hayashi et al: From their Figure 1, we support
> - depthwise separable
> - bottleneck/Tucker-2
> - inverted bottleneck
> - flattened
> - CP
> - low-rank filter
>
> The only difference is that the kernel used by those convolutions is factorized, i.e. a tensor network itself. We can simply substitute this tensor network into our derived expressions and proceed as usual, e.g. apply our symbolic simplifications.
> We tried to explain this compatibility in Section 6. Let us know if you are satisfied with this clarification. In case you would like us to experiment with other factorized convolutions, it would be great if you could narrow down the scope of experiments we could provide to convince you further.

---

> > ### Author Response · Authors · 2023-11-15
> > **Response to your questions (2/2)**
> >
> > **Leveraging sparsity of the index pattern:** You are right that the index pattern tensor is sparse. Right now, we store this tensor in dense (boolean) format because (1) it is relatively small and (2) currently we cannot leverage its sparsity to speed up the contraction:
> >
> > (1) Take for example the first layer of ConvNeXt-base which has large input sizes ($I_{1,2} = 256$), output sizes ($O_{1,2} = 64$), and kernel sizes ($K_{1,2} = 4$). Storing that layer's index pattern as dense boolean tensor requires $256 \cdot 64 \cdot 4 = 65,536$ bits, or 8 kiB.
> >
> > (2) Ideally, we would like to not only store the index pattern in sparse format, but also leverage its sparsity in the contraction with other tensors. However, the `einsum` implementation in PyTorch that we rely on requires all tensors to be dense (to the best of our knowledge, this limitation applies to all other popular ML frameworks).
> >
> > We tried experimenting with TACO [2], a tensor algebra compiler which is capable to compile tensor expressions involving sparse and dense tensors. The Python front-end only supports CPU code generation and we eventually had to discard it because we kept encountering memory leaks that seem to be caused by its Python front-end.
> >
> > For convolutions with special structure, our symbolic simplifications from Section 4 leverage the sparsity of $\mathbf{\mathsf{\Pi}}$ in that they re-express its contraction through cheap operations such as `reshape`s and `narrow`s. In the general case, however, we are unaware of a framework that can leverage the sparsity. Such a framework would certainly further improve our approach.
> >
> > **Minor:** We have uploaded a version that fixes the formatting issues of the tables in the appendix. Thanks for pointing them out.
> >
> > **References:**
> >
> > [1] Hayashi, K., Yamaguchi, T., Sugawara, Y., & Maeda, S. (NeurIPS 2019). Exploring unexplored tensor network decompositions for convolutional neural networks.
> >
> > [2] Kjolstad, F., Chou, S., Lugato, D., Kamil, S., & Amarasinghe, S. (IEEE/ACM International Conference on ASE 2017). Taco: a tool to generate tensor algebra kernels.

---

### Official Review · Reviewer_Z3Mj · 2023-10-29

**Soundness:** 4 excellent
**Presentation:** 4 excellent
**Contribution:** 4 excellent
**Rating:** 8
**Confidence:** 5

**Summary:**

The work studies how to represent the CNN layers efficiently using tensor networks. With this framework, the authors further studied efficient automatic differentiation, focusing on the  KFC and KFAC-reduce, two types of approximation of second-order information.

**Strengths:**

1. The paper is quite well-written. I’d like to highlight it because the papers of tensor networks (TNs) are typically mathematically complicated but this paper makes it very clear.
2. Although this work is not the first to model CNN layers with TNs (see. Hayashi’s work in Neurips’19), it highlights the usefulness of tensor modeling for computationally efficient automatic differentiation, which is very important in the computation of deep learning.
3. The work connects TNs with several critical techniques in ML like KFAC and randomized autodiff. I think these ideas are very helpful to boost the activity of the tensor community to put more effort in machine learning.

**Weaknesses:**

The novelty is relatively weak. For example,  Section 4.2 introduced not too much interesting tricks. It would be better to put this part in the Supp. and instead to illustrate more numerical results.

**Questions:**

In Section 2.2, I cannot fully follow how to use the set operation with the index tuples to model the tensor contractions. Could you give a more intuitive explanation or examples?

---

> ### Author Response · Authors · 2023-11-15
> **Response to your questions**
>
> Dear Reviewer Z3Mj,
>
> thanks for your strong support! We agree that tensor networks have a great potential for machine learning and hope we can contribute to making them even more popular with this work.
>
> **Update on results:** In response to Reviewer D3NB, we have extended the run time evaluation by a theoretical and empirical analysis of KFAC-reduce's memory consumption (added as Appendix G). We showed that our TN implementation is not only consistently faster (up to 4.5x), but also has much lower peak memory (with memory savings up to 3GiB for the CNNs we benchmarked in the main text).
>
> **Clarification on Section 2.2:** You are right that the notation in Section 2.2 is complex and may be better explained by an example. For a matrix multiplication `C = einsum("ij,jk->ik", A, B)` you can think of the index tuples as `S_A = ("i", "j")`, `S_B = ("j", "k")` and `S_C = ("i", "k")`. The set operations in Equations 3, 4 serve to identify which indices are summed out, i.e. are not part of the output index tuple: `(S_A ∪ S_B) \ S_C = ("j")`.
>
> Our goal was to rigorously write down what `einsum` does and to convince the reader that drawing a tensor diagram is more intuitive and requires less cognitive load than using Equations 3, 4. We have added the above example to Appendix H2 and referenced it in the main text. Let us know if this improves your understanding of the presentation.
>
> Please also let us know if you have any follow-up questions. We would be happy to discuss further.

---

> > ### Comment · Reviewer_Z3Mj · 2023-11-23
> > **Thank you for the response.**
> >
> > The part of "Clarification on Section 2.2:" is very clear. Thank you for the explanation. I found that many reviewers mentioned their concerns about the novelty of this work: it seems to be similar to the work (Hayashi et al., 2019). On this point, I'd like to highlight again from my review that "Although this work is not the first to model CNN layers with TNs (see. Hayashi’s work in Neurips’19), it highlights the *usefulness of tensor modeling for computationally efficient automatic differentiation*, which is very important in the computation of deep learning."

---

### Official Review · Reviewer_XFQf · 2023-10-31

**Soundness:** 3 good
**Presentation:** 3 good
**Contribution:** 2 fair
**Rating:** 5
**Confidence:** 2

**Summary:**

This paper proposes a perspective to simplify convolutions through tensor networks (TNs) which allow reasoning about the underlying tensor multiplications by drawing diagrams. To demonstrate its expressiveness, the diagrams of various autodiff operations and popular approximations of second-order information are derived. Finally, the computational performance improvement is proved under the proposed perspective.

**Strengths:**

1. The proposed perspective is significant to the development of convolution neural networks since it opens up potential research prospects.
2. Based on the proposed perspective of the tensor network, the authors derive the Jacobians of convolution and automatic differentiation. These efforts are quite meaningful since both derivatives and automatic differentiation mechanisms always play an important role in ML research.
3. Both implementation results relying on established machinery for efficient TN contraction and experimental results show the advantage of this perspective.

**Weaknesses:**

The main concern is contribution. The authors point out the advantages of this perspective rather than developing a framework in a novel way. From this point of view, the contribution seems limited. Therefore, the authors' central contribution only lies in some derivation based on this perspective, such as automatic differentiation.

**Questions:**

I'm not sure whether the proposal of perspective is a contribution and means much to the community or not. It would be helpful to provide some explanation about this point.

---

> ### Author Response · Authors · 2023-11-15
> **Response to your review**
>
> Dear Reviewer XFQf,
>
> Thank you for your feedback.
>
> **Contribution beyond new perspective:** We would like to tackle your point that the paper's only contribution is presenting a new perspective.
> Specifically, we believe that our paper's experiments elaborate on concrete applications and advance the state of the art:
>
> We can accelerate the pre-conditioner computation of a popular second-order method for deep learning by up to 4.5x. The run time improvements enable more frequent pre-conditioner updates which are so far not very popular due to the increased cost. As requested by Reviewer D3NB, we extended the run time results and measured the memory consumption, showing that our TN approach has consistently smaller peak memory with savings up to 3 GiB for KFAC-reduce. The authors of [1] mention KFAC-reduce's memory overhead caused by `im2col` as important limitation. By reducing the peak memory, our approach enables operating the algorithm at larger batch sizes. This is an important improvement because second-order methods seem to require larger batch sizes than first-order methods (see for instance [2, 3]).
>
> Our faster and more memory-efficient KFAC implementation for convolutions is not only relevant to the optimization community. KFAC is also a popular curvature approximation for Bayesian neural networks to construct Laplace approximations [4].
>
> Please let us know if you have any follow-up questions. We would be happy to discuss them!
>
> **References:**
>
> [1] Eschenhagen, R., Immer, A., Turner, R. E., Schneider, F., & Hennig, P. (NeurIPS 2023): Kronecker-factored approximate curvature for modern neural network architectures.
>
> [2] Martens, J., & Grosse, R. (ICML 2015): Optimizing neural networks with Kronecker-factored approximate curvature.
>
> [3] Grosse, R., & Martens, J. (ICML 2016). A kronecker-factored approximate Fisher matrix for convolution layers.
>
> [4] Daxberger, E., Kristiadi, A., Immer, A., Eschenhagen, R., Bauer, M., & Hennig, P. (NeurIPS 2021). Laplace redux - effortless bayesian deep learning.

---

### Official Review · Reviewer_D3NB · 2023-11-01

**Soundness:** 3 good
**Presentation:** 3 good
**Contribution:** 3 good
**Rating:** 6
**Confidence:** 4

**Summary:**

The paper discusses the analysis and simplification of convolutions in neural networks using tensor networks (TNs). Convolutional layers are found to be more challenging to analyze than other layers in deep learning architectures. The authors propose a new perspective using TNs, which allow for reasoning about tensor multiplications through diagrams. They demonstrate the expressive power of TNs by deriving diagrams for various automatic differentiation operations and approximations of second-order information. The document also introduces convolution-specific transformations based on connectivity patterns to simplify TN diagrams. The authors compare the computational performance of default implementations and TN implementations, showing potential speed-ups. They also mention the potential for hardware-efficient tensor dropout for approximate backpropagation.

**Strengths:**

1. StrengthsRepresenting convolution operation as multiple tensor contractions, which is quite interesting and novel.
2. By giving TN representation of convolution operation, the authors find that some memory-cost operations can be improved, e.g., KFC and its variants.
3. This paper is easy to understand and presents many graphical operations to illustrate the operations under the TNs framework.

**Weaknesses:**

1. This method is relatively straightforward and intuitive. The primary innovation of the paper lies in the use of tensor networks to represent CNN operations. However, when it comes to accelerating the KFC process, the paper lacks theoretical analysis on how much memory consumption is reduced. Furthermore, in the experiments, its effectiveness is only demonstrated based on the proportion of experimental runtime. Whether in theory or practice, the paper's description of the improvements in KFC is insufficient.
2. The advantages of using tensor networks to represent CNNs are not thoroughly discussed in this article. The paper primarily focuses on the advantages in the context of KFC, leading me to believe that it is primarily aimed at addressing memory consumption issues within KFC. Therefore, it might be more appropriate to modify the paper's topic and title to "Accelerating KFC with Tensor Network (TN) Methods.”

**Questions:**

1. The main improvement of this paper is that it avoids to unfolding the input tensor [[X]] using  memory cost methods, e.g., im2col. However, in both theoretical and practical experiments, what amount of memory savings can be achieved by using tensor networks for KFC training?
2. The paper provides a comprehensive guide on how to use Tensor Networks to represent CNNs, and offers detailed operations for various CNNs. However, in terms of the advantages of using Tensor Networks to represent CNNs, the paper lacks further analysis and discussion beyond a brief analysis in the context of KFC. For instance, once CNNs are represented in the form of TN, could this representation also be benefit to other second-order analysis and optimization methods, such as the Approximate Hessian diagonal, KBFGS, and Hessian rank mentioned in the Introduction? If this is possible, I would prefer to see the authors provide a more in-depth discussion.

---

> ### Author Response · Authors · 2023-11-15
> **Response to your questions**
>
> Dear Reviewer D3NB,
>
> thanks for your thorough review and detailed questions. We are happy to add more details to our answers below and discuss follow-up questions; please let us know if you have any.
>
> **Memory consumption (Q1):** You have a point that we do not support our claim of reduced peak memory for KFAC-reduce with evidence. To alleviate this concern, we added a theoretical analysis and empirical evaluation in Appendix G:
>
> For KFAC-reduce, the main difference between default and TN implementation is the computation of the averaged unfolded input $[[\mathbf{\mathsf{X}} ]]^{(\text{avg})} := \frac{1}{(O_1 O_2)} 1_{O_1 O_2}^{\top} [[ \mathbf{\mathsf{X}} ]]$ which consists of $C_{\text{in}} K_1 K_2$ numbers.
>
> - The default implementation needs to build up the unfolded input. This requires extra storage of $C_{\text{in}} K_1 K_2 O_1 O_2$ numbers.
>
> - Our proposed TN implementation uses the averaged index patterns, directly computed via a modification of Algorithm D1. They consist of $I_1 K_1 + I_2 K_2$ numbers. In contrast to the default implementation, spatial dimensions are de-coupled and there is no dependency on $C_{\text{in}}$.
>
> - For structured convolutions, we can use our proposed simplifications to express the pattern tensor action through `reshape`s and `narrow`s that often do not require additional memory. This completely eliminates the need to store additional tensors.
>
> To empirically demonstrate this memory reduction inside the computation of the KFAC-reduce factor, we measured its peak memory and subtracted the memory of storing the input $\mathbf{\mathsf{X}}$ and the result $\hat{\mathbf{\Omega}}$. This serves as a proxy for the extra memory that is temporarily required. We consistently observe that the default implementation requires more extra memory than the TN implementation, whose demand is further reduced for structured convolutions if we enable our simplifications.
>
> **For example, our TN implementation uses 3 GiB less memory on ConvNeXt-base's `features.1.0.block.0` convolutions.** This is a considerable fraction of the 16-32 GiB available on most contemporary GPUs.
>
> These results further substantiate our claims as they complement our demonstrated run time improvements. Our work leads to direct improvements of KFC in that it not only enables more frequent pre-conditioner updates (currently, updating every $10-100$ steps is popular), but also extends the regime of feasible batch sizes such an optimizer can operate in. This is important as second-order methods seem to require larger batch sizes than first-order methods (e.g. [1, 2]).
>
> **Applications other than KFC (Q2):** We take your point that our performance evaluation focuses on KFC. This is because it makes a great example for a non-standard operation that is interesting for the design of new algorithms, but whose implementation in existing ML frameworks suffers from limitations. Our second application, randomized autodiff, is another---yet orthogonal---instance that demonstrates the flexibility of a TN implementation: While ML frameworks support efficient multiplication with the Jacobian, randomizing the vector-Jacobian product is hard because its implementation is a black box. By deriving the VJP as a TN, we provide a white box implementation which can be randomized more easily.
>
> The applications we provide in the main text are representatives of operations that are challenging to realize with existing frameworks, e.g. approximating Hessian diagonals (Appendix B4), or diagonals and mini-block diagonals of GGN/Fisher matrices (Appendix B2 & C). Note that we not only mention them in the introduction, but also provide their tensor diagrams in the appendix (see Appendix A and Table B2 for visual and pseudo-code overviews). By doing so, we make them accessible to the community, and allow other works to benefit from our proposed TN simplifications. Please let us know if you would like us to provide more details on one of these operations in the appendix, or add the discussion of a new quantity that is currently not addressed.
>
> Our faster and more memory-efficient KFAC implementation for convolutions is not only relevant for optimization. KFAC is also a popular curvature approximation for Bayesian applications with neural networks, such as constructing Laplace approximations (e.g. [3] and references within).
>
> **References:**
>
> [1] Martens, J., & Grosse, R. (ICML 2015): Optimizing neural networks with Kronecker-factored approximate curvature.
>
> [2] Grosse, R., & Martens, J. (ICML 2016). A kronecker-factored approximate Fisher matrix for convolution layers.
>
> [3] Daxberger, E., Kristiadi, A., Immer, A., Eschenhagen, R., Bauer, M., & Hennig, P. (NeurIPS 2021). Laplace redux - effortless bayesian deep learning.

---

### Author Response · Authors · 2023-11-21

We once again thank all reviewers for their thoughtful and constructive feedback!

We would like to briefly summarize the updates in the revised version (changes highlighted in orange):

- In response to Reviewer D3NB, we have extended the run time evaluation by a theoretical and empirical *analysis of KFAC-reduce's memory consumption* (Appendix G). We showed that our TN implementation is not only consistently faster (up to 4.5x), but also has much lower peak memory (with memory savings up to 3GiB).

- In response to Reviewer Z3Mj, we have added a *concrete example for matrix-matrix multiplication* in Appendix H2 to explain the tuple notation from the definition of tensor multiplication (Equations 3, 4).

- In response to Reviewer 4fmq, we have clarified that our *derivations are compatible with non-standard convolutions*, e.g. separable convolutions (group convolutions) and factorized convolutions (e.g. those in [1]). For the latter, the main difference is that the kernel is a tensor network itself that can simply be substituted into our derived expressions.

We would be happy to answer any final questions before the end of the discussion period, if there are any.

**References:**

[1] Hayashi, K., Yamaguchi, T., Sugawara, Y., & Maeda, S. (NeurIPS 2019). Exploring unexplored tensor network decompositions for convolutional neural networks.

---

### Meta-Review · Area_Chair_nEau · 2023-12-06

**Metareview:**

The paper examines convolutions under tensor networks (TNs), which enable a diagrammatic representation of tensor multiplications. It focuses on enhancing automatic differentiation operations and approximations of second-order information through TN transformations. The authors also demonstrate potential speedup improvements.

Strengths
- Introduces an innovative way of representing convolution operations as multiple tensor contractions.
- Demonstrates improvements in computational efficiency in the context of KFAC and its variants.

Weaknesses
- Limited novelty and overall contribution.
- Empirical results and experimental comparisons are lacking to truly validate the approach as a practical speedup of KFAC (see reviewer concerns).

The paper is borderline. Overall, there does not seem to be a strong case for the work, where its limited novelty and lack of empirical or theoretical work make it difficult to recommend the work.

**Justification For Why Not Higher Score:**

See weaknesses.

**Justification For Why Not Lower Score:**

N/A

---

### Decision · Program_Chairs · 2024-01-16

Reject